# Taming the Loss Landscape of PINNs with Noisy Feynman–Kac Supervision: Operator Preconditioning and Non-Asymptotic Error Bounds

**Nathanael Tepakbong** [1]   **Hanyu Hu** [2]   **Chengyu Liu** [1]   **Xiang Zhou** [2]

## Abstract

Physics-Informed Neural Networks (PINNs) often train slowly or fail to converge on challenging partial differential equations (PDEs), a behavior recently linked to severely ill-conditioned loss landscapes inherited from the underlying differential operator. We study PINNs augmented with a pointwise data-fidelity term, added at a few points in the domain to the standard residual and boundary losses. We show that this supervision term acts as an operator-level preconditioner: for suitable weights, our comparison bounds guarantee a substantially smaller condition number than under the standard PINN loss, independently of how the pointwise labels are obtained. For a broad class of PDEs admitting a Feynman-Kac (FK) representation, we generate such labels by Monte Carlo averages of the FK functional, resulting in what we call "FK-PINNs", and using the excess risk decomposition approach, we derive non-asymptotic $L^2(\Omega)$-error bounds for FK-PINNs with $\mathtt{tanh}$ activation trained by finitely many steps of gradient descent. Along the way, we establish pseudo-dimension bounds for first- and second-order derivatives of $\mathtt{tanh}$ neural networks, which are of independent interest and, to the best of our knowledge, new. Numerical experiments on Poisson, Schrödinger, mean exit time, and committor problems corroborate the theory, and show that FK-PINNs can successfully solve PDEs for which standard PINNs exhibit severe failure modes.

[1]Department of Data Science, City University of Hong Kong, 83 Tat Chee Ave, Hong Kong [2]Department of Mathematics, City University of Hong Kong, 83 Tat Chee Ave, Hong Kong. Correspondence to: Nathanael Tepakbong <ntepakboc@my.cityu.edu.hk>.

*Proceedings of the 43rd International Conference on Machine Learning*, Seoul, South Korea. PMLR 306, 2026. Copyright 2026 by the author(s).

## 1. Introduction

Partial differential equations (PDEs) underpin much of computational science and engineering. While classical numerical methods, such as finite elements, are mature and accurate, their cost grows quickly with dimension and geometric complexity, making large-scale or high-resolution simulations prohibitively difficult in numerous practical scenarios. Physics-informed neural networks (PINNs) offer a mesh-free alternative: a neural network $u_\theta$ approximates the solution, and training is driven by a loss that penalizes PDE residuals and boundary-condition violations at sampled collocation points (Raissi et al., 2019). PINNs have been investigated in a variety of settings, including forward and inverse problems, parameter estimation, and coupled multi-physics models (Cuomo et al., 2022).

Although the idea of enforcing physical constraints within expressive neural networks is conceptually appealing, many studies report severe training pathologies. For moderately stiff or high-frequency problems, optimization can be extremely slow and may fail to reduce the residual even with expressive architectures (Krishnapriyan et al., 2021; Wang et al., 2021). Moreover, the training dynamics are highly sensitive to hyperparameters, sampling strategies, and loss weights, hence small changes in either of these can lead to vastly different outcomes. Various heuristics, such as adaptive sampling, curriculum training, residual/gradient-based re-weighting or domain decomposition have been proposed to alleviate these issues (Wu et al., 2023; Krishnapriyan et al., 2021). Although such approaches have been shown to help in specific cases, there is still no generally reliable way to obtain well-conditioned, robust PINN objectives.

### 1.1. An Operator-Conditioning Perspective on PINN Training

Recent work investigates this issue through an *operator-conditioning* and loss landscape lens. In particular, De Ryck et al. (2024) analyze gradient descent on PINN objectives and relate the dynamics to the spectrum of the Hermitian square $\mathcal{L}^*\mathcal{L}$ of the underlying PDE operator $\mathcal{L}$. Closely related analyses likewise tie PINN training difficulty to the conditioning of PDE-induced operators, their associated kernels, or PINN loss Hessians, and explain through theoretical

and numerical results the resulting optimization pathologies (Rathore et al., 2024; Wang et al., 2022; Chen et al., 2024; Wang et al., 2026). In this view, a "Hessian-like" operator governs training: when it is ill-conditioned—with widely spread eigenvalues and nearly null modes—some parameter directions evolve orders of magnitude more slowly than others, leading to stagnation and failure to fit the residual even when the target solution lies well within the hypothesis class.

This operator-centric view suggests that successful training of PINNs is not just a matter of expressive networks or dense collocation sampling, but also of constructing losses and optimization schemes whose associated operators are well conditioned. Several prior works have indeed proposed to modify the traditional gradient-based neural network training algorithms by schemes which take into account the natural function space geometry associated with PINNs (Müller & Zeinhofer, 2023; Schwencke & Furtlehner, 2025; Chen et al., 2025; Wang et al., 2022; 2026). In many cases, these modified training schemes can be interpreted as applying a preconditioning on the operator (De Ryck et al., 2024).

Complementary to these optimizer-based approaches, we instead propose to modify the training objective itself: we design *preconditioned* PINN losses that augment the standard residual and boundary terms with lightweight "mass" or data-anchoring penalties, which effectively precondition the operator governing the training dynamics. Such data could be obtained in a number of manners, through e.g. coarse FEM computations, experimental data or any kind of numerical simulation. In this work, we propose constructing these penalties using probabilistic representations of PDE solutions via the Feynman–Kac formula, yielding a simple, scalable, and mesh-free modification of the PINN loss that is completely agnostic to architecture and optimizer choice.

### 1.2. Feynman–Kac Supervision as Operator Preconditioning

For a broad class of linear second-order elliptic and parabolic PDEs, the Feynman–Kac (FK) formula represents the solution $u^\star(x)$ as the expectation of a functional of a stochastic process $(X_t)_{t\geq 0}$ solving a certain SDE (Øksendal, 2003; Karatzas & Shreve, 2014). For an appropriate choice of SDE and stopping time, $u^\star(x)$ is given by an expectation over trajectories of $X_t$ started at $x$, with the functional involving source terms, potentials, and boundary data along the path. Approximating this expectation by simulating finitely many time-discretized trajectories from $x$ and averaging the corresponding FK functional yields a natural Monte Carlo estimator of $u^\star(x)$.

Classical probabilistic numerical methods for PDEs exploit this idea by simulating many trajectories at spatial points of interest to approximate the solution field (Sabelfeld, 1991;

Billaud-Friess et al., 2018). In contrast, we use FK only as a *sparse* source of supervision: we incorporate a small number of Monte Carlo FK estimates into the PINN loss as anchors, augmenting the standard residual and boundary terms with a penalty on discrepancies between $u_\theta$ and these estimates at a few interior points. We refer to the resulting model as a *Feynman–Kac-Preconditioned Physics-Informed Neural Network* (FK-PINN), and show that the added term acts as an operator preconditioner, contributing a mass-like component that improves conditioning and stabilizes gradient descent even for relatively small FK budgets.

### 1.3. Contributions

This paper makes the following contributions:

1. **Feynman–Kac-Preconditioned PINN objective.** We introduce FK-PINNs, which augment the standard PINN loss with a data-fidelity term based on Monte Carlo FK estimates of the solution at a small set of interior domain points. The objective is mesh-free and acts as a simple drop-in replacement compatible with standard PINN architectures and training pipelines.

2. **Operator-preconditioning analysis.** Building on the operator-conditioning framework of De Ryck et al. (2024) and Rathore et al. (2024), we rigorously show in Theorem 5.4 that adding a suitable data-fidelity term to the standard PINN loss leads to a bounded and substantially improved condition number for the resulting loss. As a consequence, we deduce vastly improved convergence guarantees for gradient descent performed on this augmented objective. This regularizing effect is agnostic to the source of "supervised data" and is thus applicable to FK supervision as well as any other such method.

3. **Learning-theoretic error decomposition.** We develop in Theorem 6.2 a learning-theoretic analysis of FK-PINNs trained by gradient descent. Under standard regularity assumptions, we derive high-probability $L^2$-error bounds between the true solution $u^\star$ and an FK-PINN after $T$ steps of gradient descent. The bounds are non-asymptotic and depend explicitly on the problem parameters.

4. **Empirical illustrations on canonical PDE problems.** We complement our theory with experiments on Poisson, Schrödinger, mean exit time, and committor problems. FK-PINNs exhibit improved training dynamics, better-conditioned operator spectra, lower solution error, and remain stable on instances where baseline PINNs suffer catastrophic optimization failures, all while using relatively small FK budgets. Detailed numerical results are reported in Appendix E.

## 2. Background and Problem Setup

### 2.1. PDE Setting and Function Spaces

We consider linear second-order elliptic and parabolic PDEs on a bounded Lipschitz domain $\Omega \subset \mathbb{R}^d$ with Dirichlet boundary conditions. Let $\mathcal{L}$ be a linear second-order differential operator with bounded measurable coefficients, and consider

$$\mathcal{L}u(x) = f(x), \quad x \in \Omega, \tag{2.1}$$

with boundary condition

$$u(x) = g(x), \quad x \in \partial\Omega. \tag{2.2}$$

We assume that $\mathcal{L}$ is uniformly elliptic or parabolic in the usual sense, and that $g$ is sufficiently regular. Under these standard assumptions, there exists a unique weak solution $u^\star \in H^1(\Omega)$ with $u^\star|_{\partial\Omega} = g$ (Evans, 2022).

In later sections, we will restrict our attention to operators $\mathcal{L}$ for which the solution $u^\star$ admits a Feynman–Kac representation in terms of an associated stochastic differential equation (SDE) (Øksendal, 2003). This class covers many important PDE models used in applications (E et al., 2021). We also note that our construction has natural analogues for broader classes of nonlinear PDEs that admit similar stochastic representations (E et al., 2017). We defer a more detailed discussion to Section 3 and Appendix A.

### 2.2. Standard PINN Approximation of PDE Solutions

Let $\{u_\theta : \theta \in \Theta\}$ be a parametric class of functions implemented by a fully connected neural network with depth $L$, layer-wise widths $(m_\ell)_{\ell=0}^L$ and activation $\sigma \equiv \tanh$, mapping $x \in \Omega \subset \mathbb{R}^d$ to $u_\theta(x) \in \mathbb{R}$. Physics-informed neural networks (PINNs) approximate $u^\star$ by minimizing a loss that penalizes PDE residuals in the interior and violations of boundary conditions (Raissi et al., 2019).

Let $\{x_i^{\text{int}}\}_{i=1}^{N_{\text{int}}} \subset \Omega$ be interior collocation points and $\{x_j^{\partial\Omega}\}_{j=1}^{N_{\partial\Omega}} \subset \partial\Omega$ boundary collocation points, drawn i.i.d. from the uniform distribution on $\Omega$ and $\partial\Omega$. We define the empirical PDE and boundary losses

$$\mathcal{R}_{\text{PDE}}(\theta) := \frac{1}{N_{\text{int}}} \sum_{i=1}^{N_{\text{int}}} \left| \mathcal{L}u_\theta(x_i^{\text{int}}) - f(x_i^{\text{int}}) \right|^2, \tag{2.3}$$

$$\mathcal{R}_{\partial\Omega}(\theta) := \frac{1}{N_{\partial\Omega}} \sum_{j=1}^{N_{\partial\Omega}} \left| u_\theta(x_j^{\partial\Omega}) - g(x_j^{\partial\Omega}) \right|^2. \tag{2.4}$$

The standard PINN objective is

$$\mathcal{R}_{\text{PINN}}(\theta) := \mathcal{R}_{\text{PDE}}(\theta) + \lambda_{\partial\Omega} \, \mathcal{R}_{\partial\Omega}(\theta), \tag{2.5}$$

where $\lambda_{\partial\Omega} > 0$ balances PDE residuals and boundary accuracy. In practice, $\mathcal{L}_{\text{PINN}}$ is minimized using (stochastic) gradient-based optimizers (e.g., Adam), with automatic differentiation to compute $\mathcal{L}u_\theta(x)$.

### 2.3. Operator-Conditioning View of PINN Training

Recent work analyzes PINN training through the conditioning of the local quadratic model induced by the underlying differential operator (Rathore et al., 2024; De Ryck et al., 2024). We follow this viewpoint. Let $H$ denote a positive semidefinite curvature matrix for $\mathcal{R}_{\text{PINN}}$ at a local minimizer $\theta^\star$ (for instance the empirical Gauss–Newton matrix). In a neighborhood of $\theta^\star$, the loss is approximately quadratic and gradient descent is well approximated by a linear iteration governed by $H$, so components along eigendirections with small eigenvalues decay very slowly, leading to the optimization pathologies observed in practice (Wang et al., 2022; De Ryck et al., 2024; Rathore et al., 2024).

In this work, we augment the standard PINN objective with a (noisy) pointwise data-fidelity term. This adds an additional positive semidefinite curvature contribution $M$, so that the resulting curvature becomes

$$H_{\text{FK}} := H + \lambda_{\text{FK}} M, \tag{2.6}$$

with $\lambda_{\text{FK}} > 0$ a weight. Building on the operator-conditioning analyses of Rathore et al. (2024); De Ryck et al. (2024), we study in Section 5 how the spectrum and conditioning of $H_{\text{FK}}$ compare to those of $H$.

### 2.4. Related Work

**Mitigating the failure modes of PINNs.** Ever since their introduction by Raissi et al. (2019), a growing body of work documents a number of severe training difficulties when training PINNs, including spectral bias, failure to minimize residuals, and sensitivity to sampling and loss weights (Krishnapriyan et al., 2021; Cuomo et al., 2022). Finding methods to mitigate these issues is an active field of research, and proposed methods include adaptive collocation, residual-based reweighting, domain decomposition, and specialized architectures (Wang et al., 2021; Wu et al., 2023; Wang et al., 2024). While the augmentation of the PINN loss with supervised data has been shown empirically to help mitigate these failure modes (Heldmann et al., 2023; Bensoussan et al., 2023), the theoretical underpinnings of this phenomenon have remained largely unexplored.

**Operator- and loss-landscape-based analyses.** A complementary line of work analyzes PINN training through the operators induced by the PDE and the loss, relating gradient descent dynamics and convergence rates to the spectrum and conditioning of $\mathcal{L}^*\mathcal{L}$ or NTK-like operators (Wang et al., 2022; De Ryck et al., 2024). Recent such studies (Rathore et al., 2024; Chen et al., 2024; Wang et al., 2026) further show how ill-conditioned differential operators translate into ill-conditioned PINN objectives and motivate preconditioned or second-order optimization schemes. Our work adopts this operator-centric viewpoint but improves condi-

tioning by modifying the PINN objective itself.

**Feynman-Kac and Monte Carlo methods for PDEs.** Classical probabilistic numerical methods exploit Feynman–Kac representations to compute PDE solutions via Monte Carlo simulation of associated SDEs (Sabelfeld, 1991; Billaud-Friess et al., 2018; Yu & Mascagni, 2022). Recent works have combined these ideas with the expressive power of neural networks, resulting in FK-based neural PDE solvers (Han et al., 2020; Beck et al., 2021; Han et al., 2026). All of these approaches aim at directly computing pointwise estimates of $u^\star(x)$, with controllable bias and variance. This differs to the way we use the Feynman–Kac Monte Carlo data only as a *supervision term* for PINNs, and analyze its effect on both the conditioning of the loss landscape and the resulting generalization error.

## 3. Feynman–Kac Representation and Monte Carlo Supervision

### 3.1. Feynman–Kac Representation for Linear PDEs

We briefly recall the celebrated Feynman–Kac (FK) representation at the core of our Monte Carlo supervision. Consider the boundary value problem (2.1)–(2.2) for the operator $\mathcal{L}$ introduced in Section 2. Under conditions ensuring that (2.1)–(2.2) admits a Feynman–Kac representation (Øksendal, 2003; Karatzas & Shreve, 2014), there exists a diffusion process $(X_t)_{t\geq 0}$ with generator $\mathcal{L}$ and exit time $\tau := \inf\{t \geq 0 : X_t \notin \Omega\}$, such that for every $x \in \Omega$,

$$u^\star(x) = \mathbb{E}_x\bigg[ \int_0^\tau r(X_t)\, dt + h(X_\tau) \bigg], \qquad (3.1)$$

where $r$ is a running functional and $h$ is a terminal functional determined by the coefficients of $\mathcal{L}$ and the data $(f, g)$. Explicit expressions for $r$ and $h$, together with analogous stochastic representation formulas for more general PDE families are detailed in Appendix A.

This probabilistic representation is classical in the theory of diffusion processes and underlies many probabilistic numerical methods for elliptic and parabolic PDEs (Pavliotis, 2014). It naturally covers, for example, mean exit time problems and committor functions describing transition probabilities between metastable sets in stochastic dynamics (E & Vanden-Eijnden, 2006).

### 3.2. Monte Carlo Estimator of the FK Functional

To obtain pointwise supervision values for the PINN, we approximate the FK functional in (3.1) by Monte Carlo (MC) simulation of the associated diffusion. Starting from a point $x \in \Omega$, we simulate $N_{\mathrm{MC}}$ independent trajectories with a time step $\Delta t > 0$, using a standard Euler–Maruyama scheme (Gobet, 2016), and stop each trajectory when it exits

---

**Algorithm 1** Monte Carlo Feynman–Kac estimator at a point $x$

---

**Require:** Point $x \in \Omega$; number of trajectories $N_{\mathrm{MC}}$; time step $\Delta t$; truncation time $T_{\max}$; drift $b$ and diffusion $\sigma$; running and terminal functionals $r, h$.

1: **for** $m = 1, \ldots, N_{\mathrm{MC}}$ **do**
2:     Initialize $X_0^{(m)} \leftarrow x$, $S^{(m)} \leftarrow 0$, $n \leftarrow 0$.
3:     **while** $X_n^{(m)} \in \Omega$ **and** $n\Delta t < T_{\max}$ **do**
4:         Sample $\Delta W_n^{(m)} \sim \mathcal{N}(0, \Delta t\, \boldsymbol{I})$.
5:         Update $X_{n+1}^{(m)} \leftarrow X_n^{(m)} + b(X_n^{(m)})\, \Delta t + \sigma(X_n^{(m)})\, \Delta W_n^{(m)}$.
6:         Update $S^{(m)} \leftarrow S^{(m)} + r(X_n^{(m)})\, \Delta t$.
7:         Increment $n \leftarrow n + 1$.
8:     **end while**
9:     Set $\hat{\tau}^{(m)} \leftarrow n$ and update $S^{(m)} \leftarrow S^{(m)} + h(X_{\hat{\tau}^{(m)}}^{(m)})$.
10: **end for**

11: **return** $\widehat{u}^{\mathrm{MC}}(x) \leftarrow \frac{1}{N_{\mathrm{MC}}} \sum_{m=1}^{N_{\mathrm{MC}}} S^{(m)}$.

---

$\Omega$ or when a truncation time $T_{\max} > 0$ is reached.

Let $X_n^{(m)}$ denote the $n$-th point along the $m$-th simulated trajectory, and let $\hat{\tau}^{(m)}$ be the corresponding (discrete) stopping index. Our Monte Carlo approximation of $u^\star(x)$ is

$$\widehat{u}^{\mathrm{MC}}(x) = \frac{1}{N_{\mathrm{MC}}} \sum_{m=1}^{N_{\mathrm{MC}}} \bigg( \sum_{n=0}^{\hat{\tau}^{(m)}-1} r(X_n^{(m)})\, \Delta t + h(X_{\hat{\tau}^{(m)}}^{(m)}) \bigg). \tag{3.2}$$

In words, along each trajectory we approximate the FK integral by a simple time Riemann sum of $r$, add the terminal term $h$ at the (possibly truncated) exit point, and then average across all trajectories. Algorithm 1 summarizes the simulation procedure.

Under mild conditions on the diffusion and on $(\Delta t, T_{\max})$, the estimator $\widehat{u}^{\mathrm{MC}}(x)$ has variance $\mathrm{Var}[\widehat{u}^{\mathrm{MC}}(x)] = \mathcal{O}(1/N_{\mathrm{MC}})$ and a bias controlled by $\Delta t$ and $T_{\max}$. Precise bias–variance bounds and tail estimates for $\widehat{u}^{\mathrm{MC}}(x)$ are proved in Appendix C (see in particular Propositions C.11 and 6.1).

### 3.3. Selection of Supervision Points and Budget

We evaluate (3.2) at a relatively small set of *Feynman–Kac supervision points* $\{x_k^{\mathrm{FK}}\}_{k=1}^{N_{\mathrm{FK}}} \subset \Omega$. These points can be selected by simple generic strategies, such as uniform sampling over $\Omega$, low-discrepancy sequences, or an adaptive second pass that concentrates points in regions where an initial PINN exhibits large residuals.

Given a fixed computational budget, there is a natural trade-off between the number of supervision points $N_{\mathrm{FK}}$ and the

number of trajectories per point $N_{\mathrm{MC}}$: increasing $N_{\mathrm{FK}}$ improves spatial coverage, whereas increasing $N_{\mathrm{MC}}$ reduces the Monte Carlo variance at each point. Concrete parameter choices and additional implementation details for our numerical experiments are deferred to Appendix E.

## 4. Feynman–Kac-Preconditioned PINNs (FK-PINNs)

We now formulate explicitly how supervised data obtained from Algorithm 1 is used to train PINNs in our framework. Naturally, one could apply this same methodology with supervised data coming from any other source. Indeed, as we will see in Section 5 below, the improved conditioning due to supervised data is agnostic to the data source.

### 4.1. Augmented Loss with FK Supervision

Given FK supervision points $\{x_k^{\mathrm{FK}}\}_{k=1}^{N_{\mathrm{FK}}} \subset \Omega$ and corresponding Monte Carlo estimates $\widehat{u}_\Gamma^{\mathrm{MC}}(x_k^{\mathrm{FK}})$ of the Feynman–Kac functional (3.1), we define the FK supervision loss

$$\mathcal{R}_{\mathrm{FK}}(\theta) = \frac{1}{N_{\mathrm{FK}}} \sum_{k=1}^{N_{\mathrm{FK}}} \big(u_\theta(x_k^{\mathrm{FK}}) - \widehat{u}_\Gamma^{\mathrm{MC}}(x_k^{\mathrm{FK}})\big)^2. \quad (4.1)$$

Here $\Gamma$ collects the Monte Carlo parameters (number of trajectories, time step, truncation time, etc.), as described in Algorithm 1.

The full FK-PINN objective augments the standard PINN loss with (4.1):

$$\mathcal{R}_{\mathrm{FK\text{-}PINN}}(\theta, \phi) = \lambda_{\mathrm{PDE}}(\phi)\, \mathcal{R}_{\mathrm{PDE}}(\theta) + \lambda_{\partial\Omega}(\phi)\, \mathcal{R}_{\partial\Omega}(\theta)$$
$$+ \lambda_{\mathrm{FK}}(\phi)\, \mathcal{R}_{\mathrm{FK}}(\theta),$$
$$(4.2)$$

where $\mathcal{R}_{\mathrm{PDE}}$ and $\mathcal{R}_{\partial\Omega}$ are the empirical PDE residual and boundary losses defined in Section 2, and $\phi$ denotes the task-weighting parameters introduced below. When $\lambda_{\mathrm{FK}}(\phi) = 0$ and the remaining weights are fixed, (4.2) reduces to the standard PINN objective.

### 4.2. Adaptive Weighting of Loss Components

The objective (4.2) couples three losses with different noise levels: $\mathcal{R}_{\mathrm{PDE}}$, $\mathcal{R}_{\partial\Omega}$, and $\mathcal{R}_{\mathrm{FK}}$. We adopt the uncertainty-based weighting scheme of Kendall et al. (2018), in the same form used for PINNs by Niu et al. (2025). In their formulation, for task losses $\{\mathcal{R}_j\}$ with respective variances $\{\sigma_j^2\}$, the joint objective is

$$\mathcal{R}_{\mathrm{multi}} = \sum_j \left( \frac{1}{2\sigma_j^2}\, \mathcal{R}_j + \log \sigma_j \right). \quad (4.3)$$

Motivated by this, for our three tasks $j \in \{\mathrm{PDE}, \partial\Omega, \mathrm{FK}\}$ we introduce log-variances $s_j := \log \sigma_j^2$ and define train-

---

**Algorithm 2** Training a Feynman–Kac-Preconditioned PINN (FK-PINN)

**Require:** Domain $\Omega$ and boundary $\partial\Omega$; PDE operator $\mathcal{L}$ and data $(f, g)$; neural network $u_\theta$; task-weight parameters $\phi$; FK Monte Carlo parameters $\Gamma$; number of FK points $N_{\mathrm{FK}}$; numbers of interior and boundary collocation points.

1: **Stage 1: Offline FK supervision generation**
2: Choose FK supervision points $\{x_k^{\mathrm{FK}}\}_{k=1}^{N_{\mathrm{FK}}} \subset \Omega$ (e.g., uniform or low-discrepancy sampling).
3: **for** $k = 1, \ldots, N_{\mathrm{FK}}$ **do**
4:  Approximate $u^\star(x_k^{\mathrm{FK}})$ by a Monte Carlo FK estimate $\widehat{u}_\Gamma^{\mathrm{MC}}(x_k^{\mathrm{FK}})$ using Algorithm 1 with parameters $\Gamma$.
5: **end for**
6: Form the FK dataset $\mathcal{D}_{\mathrm{FK}} = \big\{(x_k^{\mathrm{FK}}, \widehat{u}_\Gamma^{\mathrm{MC}}(x_k^{\mathrm{FK}}))\big\}_{k=1}^{N_{\mathrm{FK}}}$.
7: **Stage 2: PINN training with FK supervision**
8: **while** stopping criterion not met **do**
9:  Sample interior collocation points in $\Omega$ and boundary points on $\partial\Omega$.
10:  Compute $\mathcal{R}_{\mathrm{PDE}}(\theta)$ and $\mathcal{R}_{\partial\Omega}(\theta)$ from the current collocation batches.
11:  Sample a minibatch from $\mathcal{D}_{\mathrm{FK}}$ and compute $\mathcal{R}_{\mathrm{FK}}(\theta)$ via (4.1).
12:  Evaluate the total loss $\mathcal{R}_{\mathrm{FK\text{-}PINN}}(\theta, \phi)$ in (4.2).
13:  Update $(\theta, \phi)$ using (stochastic) gradient descent.
14: **end while**
15: **return** trained parameters $(\theta, \phi)$.

---

able weights $\lambda_j(\phi) := \exp(-s_j)$ with $\phi := \{s_j\}_j$. The three scalars in (4.2), $\lambda_{\mathrm{PDE}}(\phi)$, $\lambda_{\partial\Omega}(\phi)$, and $\lambda_{\mathrm{FK}}(\phi)$, are obtained in this way and are optimized jointly with $\theta$. In practice, we initialize the $s_j$ so that the initial weights are of comparable magnitude and clip $s_j$ to a bounded interval to avoid degenerate solutions in which one loss (especially the noisy FK term) is effectively ignored.

### 4.3. Training Algorithm

FK-PINN training consists of an offline stage that generates FK supervision and an online stage that trains the network with the augmented loss (4.2). The full procedure is summarized in Algorithm 2.

## 5. Theoretical guarantees for supervision-based operator preconditioning

We now give an overview of our theoretical framework for supervision-based operator preconditioning. As we will see, the regularizing effect of supervised data on the loss landscape is completely agnostic to the source of such data, and our results are thus stated for generic data terms. Our presentation follows the PL* framework of Liu et al. (2022) and

its adaptation to finite-width PINNs by Rathore et al. (2024). Full assumptions and proofs are deferred to Appendix B.

## 5.1. Setup and PL* framework

We analyze gradient descent on the PINN loss augmented with a squared supervision term of the form (4.1), with fixed task weights, where the supervision targets may come from the Feynman–Kac construction or from any other data-labeling procedure. Throughout this section we fix constants $\lambda_{\mathrm{PDE}} > 0$, $\lambda_{\partial\Omega} > 0$, and $\lambda_{\mathrm{FK}} \geq 0$, and we consider the objectives

$$\mathcal{R}_{\mathrm{PINN}}(\theta) := \lambda_{\mathrm{PDE}}\,\mathcal{R}_{\mathrm{PDE}}(\theta) + \lambda_{\partial\Omega}\,\mathcal{R}_{\partial\Omega}(\theta),$$
$$\mathcal{R}_{\mathrm{FK\text{-}PINN}}(\theta) := \mathcal{R}_{\mathrm{PINN}}(\theta) + \lambda_{\mathrm{FK}}\,\mathcal{R}_{\mathrm{FK}}(\theta),$$

where $\mathcal{R}_{\mathrm{PDE}}$ and $\mathcal{R}_{\partial\Omega}$ are defined in Section 2.2, and, slightly abusing notation, $\mathcal{R}_{\mathrm{FK}}$ is a data supervision loss of the form (4.1), but with supervised data coming from any source. The standard PINN corresponds to $\lambda_{\mathrm{FK}} = 0$.

To carry out our analysis, we will assume that the following holds:

**Assumption 5.1** (Interpolation regime). Both $\mathcal{R}_{\mathrm{PINN}}$ and $\mathcal{R}_{\mathrm{FK}}$ are optimized in an interpolation regime: each loss has minimum value 0, and every global minimizer attains zero loss.

Assumption 5.1 is an expressivity assumption on the hypothesis class. It requires that each empirical loss can attain the value zero for at least one choice of network parameters, meaning that the training constraints are realizable. This is the standard interpolation setting used in (Liu et al., 2022; Rathore et al., 2024), and has been shown to hold even for moderately over-parametrized networks (Vardi et al., 2022; Madden, 2024).

**Definition 5.1** (PL* condition (Karimi et al., 2016)). Let $\mathcal{J}$ be a loss function and let $\theta^\star$ be any global minimizer of $\mathcal{J}$. We say that $\mathcal{J}$ satisfies the PL* condition in a neighborhood $\mathcal{S}$ of $\theta^\star$ with constant $\mu > 0$ if

$$\mathcal{J}(\theta) - \mathcal{J}(\theta^\star) \;\leq\; \frac{1}{2\mu}\,\|\nabla\mathcal{J}(\theta)\|^2 \quad \text{for all } \theta \in \mathcal{S}.$$

This is the Polyak–Łojasiewicz (PL) inequality (Polyak, 1963), adapted to possibly non-isolated minimizers as in Karimi et al. (2016); Liu et al. (2022). Under Assumption 5.1 we have $\mathcal{J}(\theta^\star) = 0$, so the PL* condition directly controls the loss in terms of the gradient norm.

**Definition 5.2** (Condition number for PL* functions). Let $\mathcal{J}$ be differentiable and $L$-smooth[1] on a set $\mathcal{S}$, and assume that $\mathcal{J}$ satisfies a PL* condition on $\mathcal{S}$ with constant $\mu > 0$.

---

[1]That is, $\nabla\mathcal{J}$ is $L$-Lipschitz on $\mathcal{S}$, meaning that $\|\nabla\mathcal{J}(\theta) - \nabla\mathcal{J}(\theta')\| \leq L\|\theta - \theta'\|$ holds for all $\theta, \theta' \in \mathcal{S}$.

We define the PL-based condition number of $\mathcal{J}$ on $\mathcal{S}$ as

$$\kappa_{\mathrm{PL}}(\mathcal{J}; \mathcal{S}) := \frac{L}{\mu}.$$

For the two objectives of interest, we fix a neighborhood $\mathcal{S}$ of the interpolation manifold and write $\kappa_{\mathrm{PINN}} := \kappa_{\mathrm{PL}}(\mathcal{R}_{\mathrm{PINN}}; \mathcal{S})$ and $\kappa_{\mathrm{FK}} := \kappa_{\mathrm{PL}}(\mathcal{R}_{\mathrm{FK\text{-}PINN}}; \mathcal{S})$, omitting the dependence on $\mathcal{S}$ when clear from context.

For the losses considered here, the constants $L$ and the PL* constant $\mu$ are controlled by the largest and smallest nonzero eigenvalues of the linearized Gauss–Newton matrix, so $\kappa_{\mathrm{PL}}$ is comparable to the usual spectral condition number. As in Liu et al. (2022); Rathore et al. (2024), we assume that the empirical PINN objectives $\mathcal{R}_{\mathrm{PINN}}$ and $\mathcal{R}_{\mathrm{FK}}$ are $L$-smooth and satisfy a PL* condition on a neighborhood $\mathcal{S}$ of the interpolation manifold.

**Theorem 5.3** (Rathore et al. (2024), Theorem 8.4). *Consider the standard PINN loss $\mathcal{R}_{\mathrm{PINN}}$ for a linear PDE and an interpolating minimizer $\theta^\star$ as in Assumption 5.1. Let $\mathcal{T}_K(\theta^\star)$ be the associated kernel integral operator with eigenvalues $(\lambda_j)_{j\geq 1}$ satisfying $\lambda_j \leq Cj^{-\beta}$ for some $C > 0$ and $\beta > 1$. Let $\kappa_{\mathrm{PINN}}$ denote the PL-based condition number of Definition 5.2 on a PL* neighborhood of $\theta^\star$. Then there exists a constant $c > 0$ such that, with high probability for all sufficiently large number $N$ of collocation points,*

$$\kappa_{\mathrm{PINN}} \;\geq\; c\,N^{\beta/2}.$$

Thus, in the PDE-only case the PL-based condition number of the PINN loss deteriorates polynomially as the residual (collocation) mesh is refined.

## 5.2. Effect of supervision on conditioning

Augmenting the loss with a data term adds curvature in parameter space, which can prevent the conditioning from deteriorating as the residual mesh is refined. The following theorem makes this effect precise by controlling the local PL* and smoothness constants of the data-augmented objective. Before stating our result, we also require the following compatibility condition ensuring that the supervised data term is not "blind" to parameter directions that are relevant for satisfying the PDE constraints.

**Assumption 5.2.** Consider small parameter perturbations around the parameters of interest. Split such perturbations into those that change the supervised loss term and those that do not. We assume that the physics part of the objective remains well conditioned on the second class of perturbations, and that the interaction between the two classes is not too large.

A precise formulation is given in Section B as Assumption B.1. We can now formulate our main result of this section:

**Theorem 5.4** (Informal). *Assume the setting of Theorem 5.3, the regularity assumptions on the supervision term $\mathcal{R}_{\text{FK}}$ in (4.1) stated in Section B, and the compatibility condition of Assumption B.1. Then there exist constants $\mu_0, L_0, C > 0$, which may depend on the task weights and on the supervised data distribution but are independent of the number $N$ of PDE collocation points, such that with high probability, the data-augmented objective $\mathcal{R}_{\text{FK-PINN}}$ is $L_0$-smooth and satisfies a PL\* condition with constant $\mu_0$ on a neighborhood $\mathcal{S}$ of the interpolation manifold. In particular,*

$$\kappa_{\text{FK}} = \kappa_{\text{PL}}(\mathcal{R}_{\text{FK-PINN}}; \mathcal{S}) \leq C$$

*uniformly in $N$.*

A detailed statement and proof are given in Section B (see in particular Theorem B.9).

Combining Theorems 5.3 and 5.4 yields a sharp contrast between the two objectives: while for the standard PINN loss, the PL-based condition number $\kappa_{\text{PINN}}$ deteriorates polynomially with the number of collocation points , the data-augmented loss' condition number $\kappa_{\text{FK}}$ remains uniformly bounded as the mesh is refined, provided the supervised data points satisfy the compatibility requirements stated in Assumption 5.2. This explains why the data-supervision term can effectively enable PINNs to overcome training pathologies due to large collocation sample size.

**Implications for gradient descent.** Under the PL\* framework, Liu et al. (2022) show that gradient descent on an $L$-smooth loss $\mathcal{J}$ satisfying a PL\* condition with constant $\mu$ converges linearly to a global minimizer, with iteration complexity

$$T_{\mathcal{J}}(\varepsilon) = \mathcal{O}\big(\kappa_{\text{PL}}(\mathcal{J}; \mathcal{S}) \log(1/\varepsilon)\big), \ \kappa_{\text{PL}}(\mathcal{J}; \mathcal{S}) = \frac{L(\mathcal{J}; \mathcal{S})}{\mu},$$

where $L(\mathcal{J}; \mathcal{S})$ is the Lipschitz constant of $\nabla \mathcal{J}$ on $\mathcal{S}$. For the standard PINN loss this gives $T_{\text{PINN}}(\varepsilon) = \mathcal{O}\big(\kappa_{\text{PINN}} \log(1/\varepsilon)\big)$, and Theorem 5.3 suggests that $T_{\text{PINN}}(\varepsilon)$ grows like $N^{\beta/2} \log(1/\varepsilon)$ as $N$ increases. In contrast, Theorem 5.4 shows that $\kappa_{\text{FK}}$ stays bounded in $N$, so $T_{\text{FK}}(\varepsilon) = \mathcal{O}\big(\log(1/\varepsilon)\big)$, i.e., reaching error $\varepsilon$ costs about a factor $N^{\beta/2}$ more iterations for the standard PINN than for the data-enhanced PINN.

# 6. Error Analysis of FK-PINNs

## 6.1. Setting and regime

We complement the operator-level PL\* analysis from Section 5 with learning-theoretic error bounds for FK-PINNs. The analysis, which relies in part on explicit control of the error and noise induced by the Monte Carlo approximation scheme, is specific to FK-PINNs, and an extension to other supervised data generation processes would require a separate case-specific study. As per Section 2, we consider the hypothesis class $\mathcal{F}$ of fully connected networks $u_{\boldsymbol{\theta}}$ with depth $L$, layer-wise widths $(m_\ell)_{\ell=0}^L$ and $\tanh$ activation in all layers except for a linear output neuron. As described in Algorithm 2, the neural networks in $\mathcal{F}$ are trained to minimize the loss $\mathcal{R}_{\text{FK-PINN}}$ (4.1), which serves as an empirical risk.

We sample $N_{\text{FK}}$ FK points $x_i^{\text{FK}}$, $N_{\text{int}}$ interior points $x_i^{\text{int}}$, and $N_{\partial\Omega}$ boundary points $x_i^{\partial\Omega}$ independently and uniformly on their respective domains. At each FK point $x_i^{\text{FK}}$, a Monte Carlo approximation of the FK functional provides a supervision label $\hat{u}^{\text{MC}}(x_i^{\text{FK}})$, as described in Algorithm 1. Under the assumptions collected in Appendix D, we establish non-asymptotic $L^2(\Omega)$ error bounds for trained FK-PINNs via an approximation–estimation–optimization decomposition. To our knowledge, these are the first *non-asymptotic* learning-error guarantees for PINNs with $\tanh$ activation—a widely used choice in practice—contrasting with, e.g., earlier *asymptotic* convergence results in similar settings (Doumèche et al., 2025), data-independent *a posteriori* error bounds (Hillebrecht & Unger, 2023), or non-asymptotic results restricted to piecewise polynomial activations (Jiao et al., 2022; Lu et al., 2022; Lei et al., 2025).

## 6.2. Non-asymptotic error bound

Our first result establishes that, with the Monte Carlo estimator described by Algorithm 1, each fidelity label in $\mathcal{R}_{\text{FK-PINN}}$ is a pointwise approximation of $u^\star$ with a deterministic bias of order $\sqrt{\Delta t}$ plus an exponentially small term in $T_{\max}$ and with sub-exponential Monte Carlo noise.

**Proposition 6.1** (Sub-exponential FK noise). *Suppose that the FK data in $\mathcal{R}_{\text{FK-PINN}}$ is generated by simulating the diffusion with the Euler–Maruyama scheme described in Algorithm 1, with time step $0 < \Delta t \leq \Delta t_0$, truncation horizon $T_{\max} \in (0, \infty]$ and a fixed number of Monte Carlo paths $N_{\text{MC}}$ per point, where $\Delta t_0 > 0$ is a constant depending only on the problem data (as in Appendix C). Then each FK datum admits a decomposition*

$$Y_i^{\text{FK}} = u^\star(X_i^{\text{FK}}) + b(X_i^{\text{FK}}) + \zeta_i,$$

*where $\{(X_i^{\text{FK}}, Y_i^{\text{FK}})\}_{i=1}^{N_{\text{FK}}}$ are i.i.d., the bias satisfies*

$$|b(x)| \leq C_{\text{bias}}\sqrt{\Delta t} + C_T e^{-\kappa T_{\max}} \quad \text{for all } x \in \Omega,$$

*for constants $C_{\text{bias}}, C_T, \kappa > 0$ independent of $\Delta t$, $N_{\text{FK}}$ and $N_{\text{MC}}$, with the convention $e^{-\kappa T_{\max}} = 0$ when $T_{\max} = \infty$, and, conditional on $X_i^{\text{FK}}$, the fluctuation $\zeta_i$ is mean-zero and sub-exponential with parameters that do not depend on $N_{\text{FK}}$ or $N_{\text{MC}}$. A proof is given in Appendix C.*

We can now state our main learning-theoretic estimates.

**Theorem 6.2** (Error analysis for depth-3 FK-PINNs). *Assume the setting of Section 6 and the PDE, regularity and PL\* conditions of Appendix D. Let $s \geq 3$ and suppose that $u^\star \in W^{s,\infty}(\Omega)$ and set $\beta := (s-2)/(2d+4(s-2))$. Consider FK-PINNs with depth $L = 3$ and width $m$, trained, according to Algorithm 2, by $T$ steps of gradient descent on $\mathcal{R}_{\text{FK-PINN}}$ with sufficiently small step size. Denote by $\varepsilon_{\text{bias}} := C_{\text{bias}}\sqrt{\Delta t} + C_T e^{-\kappa T_{\max}}$ the bias term from Proposition 6.1. Lastly, suppose that the max layer width $m \asymp N_{\text{FK}}^{d/(4d+8(s-2))}$, and that $N_{\text{int}}, N_{\partial\Omega} \gtrsim N_{\text{FK}}$. Then, for any $\delta \in (0,1)$, there exist constants $C, c_{\text{opt}} > 0$, independent of $(N_{\text{int}}, N_{\partial\Omega}, N_{\text{FK}}, T)$, such that with probability at least $1 - \delta$,*

$$\|u_{\boldsymbol{\theta}_T} - u^\star\|_{L^2(\Omega)} \leq C\big(N_{\text{FK}}^{-\beta} + e^{-c_{\text{opt}}T} + \varepsilon_{\text{bias}} + \varepsilon_{\text{bias}}^{1/2}\big), \tag{6.1}$$

*up to logarithmic factors in $N_{\text{FK}}$ and in the sample sizes.*

**Corollary 6.3** (Rate for the FK-PINN algorithm). *In the setting of Theorem 6.2, define $\varepsilon_{\text{bias}}$ and choose the width $m$ as in the theorem, and run gradient descent for*

$$T \gtrsim \log N_{\text{FK}}$$

*iterations. Let $\hat{u}$ denote the resulting depth-3 FK-PINN. Then, up to logarithmic factors in $N_{\text{FK}}$ and in the sample sizes, there exists a constant $C > 0$, independent of $N_{\text{FK}}$, such that with probability at least $1 - N_{\text{FK}}^{-10}$,*

$$\|\hat{u} - u^\star\|_{L^2(\Omega)} \leq C\big(N_{\text{FK}}^{-\beta} + \varepsilon_{\text{bias}} + \varepsilon_{\text{bias}}^{1/2}\big), \tag{6.2}$$

*where $\beta := (s-2)/(2d+4(s-2))$. See Appendix D, Corollary D.25 for proofs.*

This result shows that, when $N_{\text{FK}}$ grows and the width is tuned as in Theorem 6.2, FK-PINNs achieve the expected rate in terms of $N_{\text{FK}}$, in line with the convergence theory of PINNs with dense physics sampling (Doumèche et al., 2025). In contrast, in the genuinely sparse FK regime where $N_{\text{FK}}$ is fixed while $N_{\text{int}}$ and $N_{\partial\Omega}$ increase, the error bound is saturated by the FK term and cannot vanish uniformly, which quantifies the statistical "cost" paid for the improved conditioning brought by FK supervision.

# 7. Numerical experiments

## 7.1. An Illustrative Example

We now illustrate our theoretical results by comparing the FK-PINNs against the standard PINN baseline on a representative PDE: a steady-state *Schrödinger-type* equation. Further extensive numerical experiments on a wider range of problems, together with implementation details are provided in Appendix E.

For a stationary system defined on a bounded domain $\Omega \subset \mathbb{R}^d$, we consider the following non-homogeneous

*Schrödinger-type* equation:

$$-\frac{1}{2}\Delta\psi(x) + V(x)\psi(x) = f(x) \qquad x \in \Omega,$$
$$\psi(x) = 0 \qquad x \in \partial\Omega, \tag{7.1}$$

where $\psi(x)$ represents the response wavefunction, $V(x)$ denotes a prescribed potential function, and $f(x)$ constitutes a given source term. The multi-scale nature of this Schrödinger-type equation, notably for oscillatory, high-frequency potentials, poses a significant challenge for conventional PINNs.

We set the domain $\Omega$ as $\Omega = [-1, 1] \times [-1, 1]$, while the potential $V(x)$ is that of a periodic crystalline lattice, defined as

$$V(x) = -V_0[\cos(kx_1) + \cos(kx_2)], \tag{7.2}$$

where $V_0 = 5$ is the potential depth and $k = 2\pi/a$ with lattice constant $a = 0.5$. This periodic structure induces strong local confinement and sharp gradients in the wavefunction $\psi(x)$. W further introduce a high-frequency external forcing term:

$$f(x) = A\sin(Kx_1)\sin(Kx_2), \tag{7.3}$$

where $A = 50$ and $K = 5$.

Figure 1 illustrates the potential $V$, forcing term $f$ and corresponding wavefunction $\psi$. We report respectively in Figures 2 and 3 the wavefunctions learned by a standard PINN, and an FK-PINN, with both identical architecture training parameters and schedule, after $30000 + 15000$ steps of Adam + L-BFGS. As the results clearly illustrate, while the standard PINN completely fails at learning the solution structure, due to the high-frequency oscillations making the loss landscape intractable, the proposed FK-PINNs effectively mitigates the underlying stiffness of the PDE and succesfully recovers the intended solution profile.

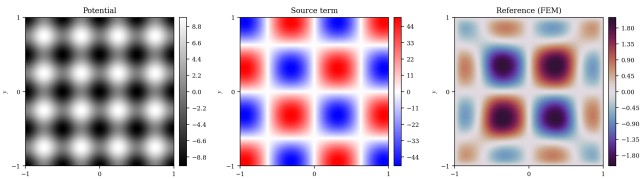

*Figure 1.* Data for the Schrödinger-type equation (7.1). **From left to right:** potential $V(\mathbf{x})$, source $f(\mathbf{x})$, and FEM reference $\psi$.

## 7.2. Summary table

We now showcase the performance of FK-PINNs when compared to standard PINNs on a number of canonical PDEs. As we can see, the ability of this supervised approach to overcome the failure modes of PINNs is clear. We refer the reader to Appendix E for further experimental details and background on the PDEs under consideration.

| Model | Metric | Poisson (E.2.2) | Schrödinger-type (7) | Mean Escape Time (E.2.3) | Committor (E.2.4) |
|---|---|---|---|---|---|
| PINNs | $L^2$ Abs Err | $1.333 \pm 0.616$ | $0.475 \pm 0.148$ | $16.56 \pm 0.055$ | $1.028 \pm 0.283$ |
| | $L^2$ Rel Err | $0.322 \pm 0.149$ | $0.624 \pm 0.195$ | $1.007 \pm 0.003$ | $0.839 \pm 0.661$ |
| | $H^1$ Abs Err | $12.42 \pm 4.345$ | $2.893 \pm 0.415$ | $43.55 \pm 0.039$ | $3.154 \pm 0.794$ |
| | $H^1$ Rel Err | $0.171 \pm 0.061$ | $0.512 \pm 0.073$ | $1.002 \pm 0.001$ | $0.547 \pm 0.074$ |
| FK-PINNs | $L^2$ Abs Err | $0.761 \pm 0.011$ | $0.073 \pm 0.008$ | $1.755 \pm 0.093$ | $0.791 \pm 0.039$ |
| | $L^2$ Rel Err | $0.1184 \pm 0.003$ | $0.096 \pm 0.010$ | $0.107 \pm 0.006$ | $0.030 \pm 0.008$ |
| | $H^1$ Abs Err | $7.822 \pm 0.412$ | $1.043 \pm 0.083$ | $1.755 \pm 0.093$ | $1.495 \pm 0.105$ |
| | $H^1$ Rel Err | $0.108 \pm 0.006$ | $0.185 \pm 0.015$ | $0.395 \pm 0.051$ | $0.153 \pm 0.049$ |

*Table 1.* Performance comparison between standard PINNs and FK-PINNs on various problems. All results are averaged over 5 independent runs with different random seeds.

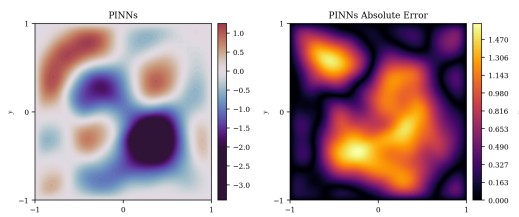

*Figure 2.* Numerical results for the standard PINN on the Schrödinger-type equation (7.1). **Left side:** PINN prediction **Right side:** absolute errors $|\psi - u_\theta|$

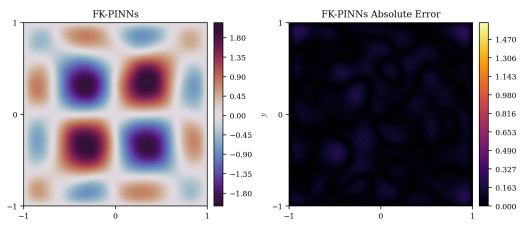

*Figure 3.* Numerical results for the FK-PINN on the Schrödinger-type equation (7.1). **Left side:** FK-PINN prediction **Right side:** absolute errors $|\psi - u_\theta|$

## 8. Conclusion

In this work, we've established that enriching the standard PINN loss with pointwise approximate solution data can provably regularize the loss landscape and enhance the convergence of gradient descent. Furthermore, we've proposed an easily implementable and flexible algorithm based on a Monte Carlo approximation of the Feynman–Kac representation of certain elliptic and parabolic PDEs to compute such approximate solution data, resulting in what we've referred to as FK-PINNs. Using a learning-theoretic framework together with tight error bounds on the Monte Carlo tails, we have derived non-asymptotic, end-to-end $L^2(\Omega)$ a priori error bounds for FK-PINNs using tanh activation trained with finitely many steps of gradient descent. Those estimates are based on pseudo-dimension bounds for first-

and second-order derivatives of tanh networks, which are of independent interest. Numerical experiments confirm our theoretical findings, showing more stable optimization, faster training, and lower solution error than baseline PINNs, even with modest Monte Carlo budgets.

## Acknowledgments

N.T. acknowledges support from the Hong Kong PhD Fellowship Scheme. X.Z. acknowledges support from the Hong Kong General Research Funds (Grants No. 11304525, No. 11318522, and No. 11308323).

## Impact Statement

This paper presents work whose goal is to advance the field of Machine Learning. There are many potential societal consequences of our work, none which we feel must be specifically highlighted here.

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

# A. More on Feynman–Kac Representations and Probabilistic PDE Solutions

This appendix collects standard probabilistic representations for linear and nonlinear parabolic and elliptic PDEs. By a Feynman–Kac (FK) representation we mean a formula of the form (3.1) expressing the solution $u$ as the expectation of a running and terminal functional of a suitable Markov process with generator $\mathcal{L}$. This structure underlies the Monte Carlo supervision term and FK-PINN loss introduced in Section 3, but the discussion here is self-contained. The following subsections present representative examples—diffusion and jump–diffusion generators, semilinear and fully nonlinear equations, branching diffusions, and stochastic control/HJB problems—to illustrate that PDEs admitting such stochastic (Feynman–Kac-type) representations form a large and mathematically rich class. We refer the interested readers to (Dynkin, 1965; Ethier & Kurtz, 2009; Karatzas & Shreve, 2014) for background on Markov processes and classical FK formulas.

## A.1. Linear parabolic and elliptic equations with diffusion generators

Let $B$ be a $d$-dimensional Brownian motion and consider the Itô SDE

$$dX_s = b(X_s)\,ds + \sigma(X_s)\,dB_s, \qquad X_t = x \in \mathbb{R}^d, \tag{A.1}$$

where $b : \mathbb{R}^d \to \mathbb{R}^d$ and $\sigma : \mathbb{R}^d \to \mathbb{R}^{d \times d}$ satisfy standard Lipschitz and linear growth conditions (Karatzas & Shreve, 2014). The associated generator $\mathcal{L}$ acts on smooth test functions $\varphi$ as

$$(\mathcal{L}\varphi)(x) = b(x) \cdot \nabla\varphi(x) + \tfrac{1}{2}\operatorname{Tr}\big(a(x)D^2\varphi(x)\big), \qquad a(x) := \sigma(x)\sigma(x)^\top. \tag{A.2}$$

A prototypical linear parabolic Cauchy problem on $[0, T] \times \mathbb{R}^d$ is

$$\begin{cases} \partial_t u(t,x) + (\mathcal{L}u(t,\cdot))(x) - V(t,x)u(t,x) + f(t,x) = 0, & (t,x) \in [0,T) \times \mathbb{R}^d, \\ u(T,x) = g(x), & x \in \mathbb{R}^d, \end{cases} \tag{A.3}$$

with bounded (or suitably integrable) data $V, f, g$. Under standard regularity assumptions on $b, \sigma, V, f, g$ (e.g. Dynkin, 1965; Karatzas & Shreve, 2014), the classical Feynman–Kac formula yields

$$u(t,x) = \mathbb{E}^{t,x}\Big[e^{-\int_t^T V(s,X_s)\,ds}\,g(X_T) + \int_t^T e^{-\int_t^s V(r,X_r)\,dr} f(s,X_s)\,ds\Big], \tag{A.4}$$

where $X$ solves (A.1) with initial condition $X_t = x$.

Elliptic boundary value problems such as Poisson, mean exit time, or committor equations arise by dropping the time derivative in (A.3). For a bounded $C^2$ domain $D \subset \mathbb{R}^d$, define the exit time $\tau_D := \inf\{s \geq 0 : X_{t+s} \notin D\}$. Under standard conditions, the Dirichlet problem

$$\begin{cases} (\mathcal{L}u)(x) - V(x)u(x) + f(x) = 0, & x \in D, \\ u(x) = g(x), & x \in \partial D, \end{cases} \tag{A.5}$$

admits the representation

$$u(x) = \mathbb{E}^x\Big[\int_0^{\tau_D} e^{-\int_0^s V(X_r)\,dr} f(X_s)\,ds + e^{-\int_0^{\tau_D} V(X_r)\,dr} g(X_{\tau_D})\Big], \tag{A.6}$$

and analogous formulas hold for Neumann and Robin boundary conditions in terms of reflected diffusions and boundary local time (cf. Dynkin, 1965; Fleming & Soner, 2006).

For the boundary value problem (2.1)–(2.2) considered in Section 3, the operator $\mathcal{L}$ and data $(f, g)$ fit into the elliptic template (A.5) (with $V$ encoding any zeroth-order term). In particular, in the Poisson, mean exit time, and committor benchmarks of the main text one typically has $V \equiv 0$, so that (A.6) reduces to

$$u(x) = \mathbb{E}^x\Big[\int_0^{\tau_D} f(X_s)\,ds + g(X_{\tau_D})\Big],$$

which is of the form (3.1), with running and terminal functionals $r$ and $h$ determined (up to the sign convention for $\mathcal{L}$) by the source term $f$ and boundary data $g$. In the algorithms of the main text, expectations such as (A.4) and (A.6) are approximated by simulating time-discretized trajectories of (A.1) (e.g. Euler–Maruyama) and averaging over samples. Weak and strong error estimates for such schemes can be found in (Kloeden & Platen, 1992; Glasserman, 2004).

## A.2. Linear equations with nonlocal generators

Let $X$ be a Lévy process in $\mathbb{R}^d$ with Lévy triplet $(b, a, \nu)$, where $b \in \mathbb{R}^d$, $a \in \mathbb{R}^{d \times d}$ is symmetric nonnegative definite, and $\nu$ is a Lévy measure. Its generator $\mathcal{L}$ acts on $C_c^2(\mathbb{R}^d)$ as

$$
\begin{aligned}
(\mathcal{L}\varphi)(x) = {} & b \cdot \nabla\varphi(x) + \tfrac{1}{2}\operatorname{Tr}\big(aD^2\varphi(x)\big) \\
& + \int_{\mathbb{R}^d \setminus \{0\}} \Big[\varphi(x+y) - \varphi(x) - \mathbf{1}_{\{|y| \leq 1\}}\, y \cdot \nabla\varphi(x)\Big]\nu(dy),
\end{aligned}
\tag{A.7}
$$

see, for instance, (Applebaum, 2009).

Linear parabolic and elliptic equations with the nonlocal operator $\mathcal{L}$ in (A.7) have the same abstract form as (A.3) and (A.5). Under suitable conditions on $(b, a, \nu)$ and the data, the solution $u$ admits a Feynman–Kac representation identical to (A.4), with $X$ now a Lévy (or more generally a Lévy-type) process (see, e.g., Dynkin, 1965; Applebaum, 2009). Monte Carlo approximation then proceeds by simulating jump–diffusion paths (using standard jump-adapted schemes) and averaging the corresponding functionals (Glasserman, 2004). In principle, FK-PINNs could be extended to this setting by supervising the network with such Lévy paths instead of pure diffusions.

## A.3. Semilinear parabolic equations and BSDE representations

Consider the semilinear parabolic PDE

$$
\begin{cases}
\partial_t u(t, x) + (\mathcal{L}u(t, \cdot))(x) + F\big(t, x, u(t, x), \sigma(x)^\top \nabla u(t, x)\big) = 0, & (t, x) \in [0, T) \times \mathbb{R}^d, \\
u(T, x) = g(x), & x \in \mathbb{R}^d,
\end{cases}
\tag{A.8}
$$

where $\mathcal{L}$ is the diffusion generator (A.2), $F$ is Lipschitz in $(y, z)$ with linear growth, and $g$ is bounded and Lipschitz. Let $X$ solve (A.1) and consider the forward–backward system

$$
dX_s = b(X_s)\, ds + \sigma(X_s)\, dB_s, \qquad X_t = x,
\tag{A.9}
$$

$$
Y_s = g(X_T) + \int_s^T F\big(r, X_r, Y_r, Z_r\big)\, dr - \int_s^T Z_r\, dB_r, \qquad s \in [t, T].
\tag{A.10}
$$

Under the above assumptions, there exists a unique adapted solution $(Y, Z)$ to (A.10) (Pardoux & Peng, 1990; El Karoui et al., 1997), and the nonlinear Feynman–Kac formula identifies

$$
u(t, x) = Y_t^{t, x},
\tag{A.11}
$$

where $(X^{t,x}, Y^{t,x}, Z^{t,x})$ denotes the solution of (A.9)–(A.10) with $X_t^{t,x} = x$. Conversely, if $u$ is a sufficiently regular solution of (A.8), then $(Y_s, Z_s) = (u(s, X_s), \sigma(X_s)^\top \nabla u(s, X_s))$ solves (A.10) (Pardoux & Peng, 1990; El Karoui et al., 1997).

Monte Carlo approximations for (A.8) discretize the forward SDE and the BSDE and estimate the conditional expectations in the backward recursion by regression-based methods (Bouchard & Touzi, 2004). In the present paper we only need the linear case where $F(t, x, y, z) = -V(t, x)\, y + f(t, x)$, which reduces to the classical FK form in Section 3.1, but the nonlinear Feynman–Kac formula suggests a natural path to extending FK-PINNs to general semilinear problems by coupling PINNs with Monte Carlo BSDE solvers.

## A.4. Semilinear equations with polynomial nonlinearities and branching diffusions

Branching diffusion processes provide representations for semilinear equations of reaction–diffusion type. A prototype is

$$
\partial_t u(t, x) = (\mathcal{L}u(t, \cdot))(x) + \lambda\big(u(t, x)^p - u(t, x)\big), \qquad u(0, x) = u_0(x),
\tag{A.12}
$$

with $\lambda > 0$, integer $p \geq 2$, and $\mathcal{L}$ the diffusion generator (A.2). Under appropriate integrability and non-explosion conditions, the solution of (A.12) can be written as

$$
u(t, x) = \mathbb{E}^x\Big[\prod_{i=1}^{N_t} u_0\big(X_t^{(i)}\big)\Big],
\tag{A.13}
$$

where $(X_t^{(i)})_{1 \leq i \leq N_t}$ are the positions of particles in a branching diffusion system with spatial motion governed by $\mathcal{L}$ and binary or $p$-ary branching at rate $\lambda$ (see, e.g., Dynkin, 1991). More general semilinear PDEs with polynomial nonlinearities in $u$ and $\nabla u$ admit representations in terms of marked branching diffusions (Henry-Labordère et al., 2019; Agarwal & Claisse, 2020).

Monte Carlo evaluation of (A.13) proceeds by simulating the associated random branching tree and averaging the weight functional, provided the variance is controlled (Henry-Labordère et al., 2019).

### A.5. Hamilton–Jacobi–Bellman equations from stochastic control

Let $A$ be a compact metric control set. For an $A$-valued progressively measurable control process $\alpha$, consider the controlled diffusion

$$dX_s^{t,x,\alpha} = b\big(X_s^{t,x,\alpha}, \alpha_s\big)\, ds + \sigma\big(X_s^{t,x,\alpha}, \alpha_s\big)\, dB_s, \qquad X_t^{t,x,\alpha} = x, \tag{A.14}$$

with running cost $\ell : [0,T] \times \mathbb{R}^d \times A \to \mathbb{R}$ and terminal cost $g : \mathbb{R}^d \to \mathbb{R}$. The value function is

$$u(t,x) := \sup_\alpha \mathbb{E}^{t,x,\alpha}\Big[\int_t^T \ell\big(s, X_s^{t,x,\alpha}, \alpha_s\big)\, ds + g\big(X_T^{t,x,\alpha}\big)\Big], \tag{A.15}$$

where the supremum ranges over all admissible controls. Writing $\mathcal{L}^a$ for the diffusion generator associated with the frozen control $a \in A$, the corresponding Hamilton–Jacobi–Bellman equation is

$$\begin{cases} \partial_t u(t,x) + \sup_{a \in A}\big\{(\mathcal{L}^a u(t,\cdot))(x) + \ell(t,x,a)\big\} = 0, & (t,x) \in [0,T) \times \mathbb{R}^d, \\ u(T,x) = g(x), & x \in \mathbb{R}^d. \end{cases} \tag{A.16}$$

Under standard compactness, measurability, and regularity assumptions, $u$ is the unique bounded viscosity solution of (A.16) and coincides with (A.15) (see Fleming & Soner, 2006).

Stochastic representations of the form (A.15) can be approximated numerically by simulation-based dynamic programming and related Monte Carlo methods for controlled diffusions; see (Fleming & Soner, 2006) for an overview. For high-dimensional parabolic PDEs and BSDEs, including HJB-type equations from stochastic control, deep learning-based methods can approximate the gradient of the value function (interpreted as a feedback control or policy) by a neural network and train it via Monte Carlo simulation of the underlying diffusion; see, for example, (E et al., 2017). While FK-PINNs in the present work are restricted to linear operators, the HJB representation suggests a possible extension in which the neural network is supervised by Monte Carlo estimates of value functions under sampled controls.

### A.6. Fully nonlinear parabolic equations and second-order BSDEs

Consider a fully nonlinear parabolic PDE of the form

$$-\partial_t u(t,x) + F\big(t,x,u(t,x), Du(t,x), D^2 u(t,x)\big) = 0, \qquad u(T,x) = g(x), \tag{A.17}$$

where $F$ is nonlinear in the Hessian $D^2 u$ and satisfies suitable ellipticity, monotonicity, and regularity conditions ensuring a comparison principle and well-posedness in the viscosity sense. Second-order backward SDEs (2BSDEs) provide pathwise representations of solutions to (A.17) (Cheridito et al., 2007). Roughly speaking, given a forward diffusion $X$, one considers an adapted quadruple $(Y, Z, \Gamma, A)$ solving a 2BSDE whose first component satisfies

$$Y_t^{t,x} = u(t, X_t^{t,x}), \tag{A.18}$$

and conversely any such 2BSDE solution identifies the unique viscosity solution $u$ of (A.17) (Cheridito et al., 2007). Numerical schemes discretize both the forward diffusion and the 2BSDE and approximate conditional expectations by Monte Carlo in an enlarged state space, providing a probabilistic route to numerical solutions of fully nonlinear equations (Cheridito et al., 2007). Extending FK-PINNs to this 2BSDE setting is an interesting direction for future research but lies beyond the scope of this paper.

### A.7. Summary table

For convenience, Table 2 summarizes the PDE classes discussed above, the corresponding operators, the probabilistic objects, and some standard references.

| PDE class | Schematic form | Probabilistic object | Representation / references |
|---|---|---|---|
| Linear parabolic / elliptic (diffusion) | $\partial_t u + \mathcal{L}u - Vu + f = 0$, or $\mathcal{L}u - Vu + f = 0$ in $D$ | Diffusion $X$ with generator $\mathcal{L}$, exit time $\tau_D$ | Feynman–Kac expectations such as (A.4), (A.6); see (Dynkin, 1965; Karatzas & Shreve, 2014) |
| Linear nonlocal (Lévy / jump–diffusion) | $\partial_t u + \mathcal{L}u - Vu + f = 0$ with $\mathcal{L}$ as in (A.7) | Lévy or Lévy-type process $X$ | Same Feynman–Kac structure as in the diffusion case with nonlocal generator (Dynkin, 1965; Applebaum, 2009) |
| Semilinear parabolic (BSDE) | $\partial_t u + \mathcal{L}u + F(t, x, u, \sigma^\top \nabla u) = 0$ | Forward diffusion $X$ + BSDE $(Y, Z)$ | Nonlinear Feynman–Kac: $u(t,x) = Y_t^{t,x}$ as in (A.11) (Pardoux & Peng, 1990; El Karoui et al., 1997) |
| Semilinear (polynomial) | $\partial_t u + \mathcal{L}u = \Phi(t, x, u, \nabla u)$, $\Phi$ polynomial | (Marked) branching diffusion system | Branching representations such as (A.13) (Dynkin, 1991; Henry-Labordère et al., 2019; Agarwal & Claisse, 2020) |
| HJB (stochastic control) | $\partial_t u + \sup_{a \in A}\{\mathcal{L}^a u + \ell(t, x, a)\} = 0$ | Controlled diffusion $X^\alpha$ | Value function representation (A.15) (Fleming & Soner, 2006) |
| Fully nonlinear parabolic | $-\partial_t u + F(t, x, u, Du, D^2 u) = 0$ | Forward diffusion $X$ + 2BSDE $(Y, Z, \Gamma, A)$ | Fully nonlinear Feynman–Kac via 2BSDE, $u(t,x) = Y_t^{t,x}$ as in (A.18) (Cheridito et al., 2007) |

*Table 2.* PDE classes covered by standard probabilistic representations. Here $\mathcal{L}$ denotes a (possibly nonlocal) Markov generator, $X$ the corresponding Markov or controlled process, and expectations $\mathbb{E}^{t,x}[\cdot]$ are taken with respect to the law of $X$ started from $(t, x)$. The FK-PINNs studied in the main text use the first (linear diffusion) row, but the same Monte Carlo supervision principle could in principle be adapted to the other classes.

# B. Preconditioning Effect of the FK data term

We now prove the conditioning result stated informally as Theorem 5.4. The idea of the proof is to compare the empirical Gauss–Newton matrix of the PDE-only objective with that of the FK-augmented objective, and to show that the supervised FK term prevents the smallest nonzero eigenvalues from collapsing as the residual mesh is refined.

We write $L_{\mathrm{PDE+BC}}(\theta)$ for the empirical PDE residual plus Dirichlet boundary loss and $L_{\mathrm{data}}(\theta)$ for a supervised data loss, and we consider

$$L_\lambda(\theta) := L_{\mathrm{PDE+BC}}(\theta) + \lambda\, L_{\mathrm{data}}(\theta).$$

This is the same structure as the fixed-weight FK-PINN objective in Section 5, with $L_{\mathrm{data}}$ playing the role of the FK supervision term and $\lambda$ corresponding to $\lambda_{\mathrm{FK}}$.

The proof combines the residual ill-conditioning mechanism in (De Ryck et al., 2024; Rathore et al., 2024) described by Proposition B.2, with the local PL$^*$ control for supervised least squares from (Liu et al., 2022). In an overparameterized regime, the data Gauss–Newton matrix is typically low rank, so adding it does not automatically bound the smallest eigenvalue of the sum. We therefore impose a compatibility condition, Assumption B.1, which ensures that every direction relevant to the PDE and boundary terms is also seen by the data term. Under this condition, the smallest nonzero eigenvalues of the combined Gauss–Newton matrix are uniformly bounded below, which yields the PL$^*$ constant and the condition-number bound in Theorem B.9.

## B.1. PINN objective with noisy data

We consider a linear PDE with Dirichlet boundary conditions on a bounded domain $\Omega \subset \mathbb{R}^d$,

$$\begin{aligned}
\mathcal{L}[u](x) &= f(x), & x &\in \Omega, \\
u(x) &= g(x), & x &\in \partial\Omega,
\end{aligned} \tag{B.1}$$

where $\mathcal{L}$ is a linear differential operator, and $f$ and $g$ are given functions. We assume that (B.1) is well-posed in the sense that there exists a unique solution $u^\star$ in an appropriate function space.

Let $\theta \in \mathbb{R}^p$ denote the PINN parameter vector, and let $u(\cdot; \theta)$ be a neural network parameterization of $u$. We distinguish three types of training points:

- *Residual points* $\{x_i^{\mathrm{res}}\}_{i=1}^{n_{\mathrm{res}}} \subset \Omega$.

- *Boundary points* $\{x_j^{\mathrm{bc}}\}_{j=1}^{n_{\mathrm{bc}}} \subset \partial\Omega$.

- *Data points* $\{x_i^{\mathrm{d}}\}_{i=1}^{n_{\mathrm{d}}} \subset \Omega$.

RESIDUAL AND BOUNDARY TERMS

The empirical residual and boundary losses are defined by

$$L_{\mathrm{PDE+BC}}(\theta) := \frac{1}{2n_{\mathrm{res}}} \sum_{i=1}^{n_{\mathrm{res}}} \big(\mathcal{L}[u(x_i^{\mathrm{res}}; \theta)] - f(x_i^{\mathrm{res}})\big)^2 + \frac{1}{2n_{\mathrm{bc}}} \sum_{j=1}^{n_{\mathrm{bc}}} \big(u(x_j^{\mathrm{bc}}; \theta) - g(x_j^{\mathrm{bc}})\big)^2. \tag{B.2}$$

NOISY DATA TERM

We additionally observe (possibly noisy) function values at the data points,

$$y_i = u^\star(x_i^{\mathrm{d}}) + \xi_i, \quad i = 1, \ldots, n_{\mathrm{d}},$$

where the noise variables $\xi_i$ are independent across $i$ and satisfy $\mathbb{E}[\xi_i] = 0$.

The data-supervision term with weight $\lambda \geq 0$ is

$$L_{\mathrm{data}}(\theta) := \frac{1}{2n_{\mathrm{d}}} \sum_{i=1}^{n_{\mathrm{d}}} \big(u(x_i^{\mathrm{d}}; \theta) - y_i\big)^2. \tag{B.3}$$

The full PINN objective is then

$$L_\lambda(\theta) := L_{\mathrm{PDE+BC}}(\theta) + \lambda\, L_{\mathrm{data}}(\theta). \tag{B.4}$$

We denote by $\Theta_\lambda^\star$ the set of global minimizers of $L_\lambda$:

$$\Theta_\lambda^\star := \arg\min_{\theta \in \mathbb{R}^p} L_\lambda(\theta).$$

STACKED FEATURE MAP AND GAUSS–NEWTON MATRIX.

To streamline notation, define the residual, boundary, and data feature maps

$$F_{\mathrm{res}}(\theta) := \frac{1}{\sqrt{n_{\mathrm{res}}}}\big(\mathcal{L}[u(x_1^{\mathrm{res}};\theta)] - f(x_1^{\mathrm{res}}), \ldots, \mathcal{L}[u(x_{n_{\mathrm{res}}}^{\mathrm{res}};\theta)] - f(x_{n_{\mathrm{res}}}^{\mathrm{res}})\big)^\top,$$

$$F_{\mathrm{bc}}(\theta) := \frac{1}{\sqrt{n_{\mathrm{bc}}}}\big(u(x_1^{\mathrm{bc}};\theta) - g(x_1^{\mathrm{bc}}), \ldots, u(x_{n_{\mathrm{bc}}}^{\mathrm{bc}};\theta) - g(x_{n_{\mathrm{bc}}}^{\mathrm{bc}})\big)^\top,$$

$$F_{\mathrm{data}}(\theta) := \frac{1}{\sqrt{n_{\mathrm{d}}}}\big(u(x_1^{\mathrm{d}};\theta) - y_1, \ldots, u(x_{n_{\mathrm{d}}}^{\mathrm{d}};\theta) - y_{n_{\mathrm{d}}}\big)^\top.$$

Then

$$L_{\mathrm{PDE+BC}}(\theta) = \frac{1}{2}\big\|F_{\mathrm{res}}(\theta)\big\|_2^2 + \frac{1}{2}\big\|F_{\mathrm{bc}}(\theta)\big\|_2^2, \quad L_{\mathrm{data}}(\theta) = \frac{1}{2}\big\|F_{\mathrm{data}}(\theta)\big\|_2^2.$$

We introduce the stacked feature map

$$F_\lambda(\theta) := \begin{bmatrix} F_{\mathrm{res}}(\theta) \\ F_{\mathrm{bc}}(\theta) \\ \sqrt{\lambda}\, F_{\mathrm{data}}(\theta) \end{bmatrix} \in \mathbb{R}^{n_\lambda}, \quad n_\lambda := n_{\mathrm{res}} + n_{\mathrm{bc}} + n_{\mathrm{d}}, \tag{B.5}$$

so that $L_\lambda(\theta) = \frac{1}{2}\|F_\lambda(\theta)\|_2^2$. Let $J_{F_\lambda}(\theta) \in \mathbb{R}^{n_\lambda \times p}$ denote the Jacobian of $F_\lambda$ at $\theta$. The associated empirical Gauss–Newton matrix is

$$G_\lambda(\theta) := J_{F_\lambda}(\theta)^\top J_{F_\lambda}(\theta) \in \mathbb{R}^{p \times p}.$$

Its eigenvalues describe the local curvature of $L_\lambda$ in parameter space, and will provide the link between operator conditioning and PL-based optimization guarantees. From this point on, the conditioning question is reduced to bounding the smallest and largest eigenvalues of $G_\lambda(\theta)$ in the region where the PL$^*$ argument is applied.

**Definition B.1.** Let $A \in \mathbb{R}^{p \times p}$ be symmetric and positive semidefinite. We write $\lambda_{\min}^+(A)$ for the smallest strictly positive eigenvalue of $A$.

## B.2. Tangent function space and population Gauss–Newton (PDE-only)

We now recall the operator-theoretic interpretation of the PDE-only population Gauss–Newton matrix from De Ryck et al. (2024). This section mostly summarizes their construction in the finite-width setting, to make our later use of their results self-contained. The only reason to introduce the tangent space $\mathcal{H}(\theta)$, the map $T(\theta)$, and the operator $\mathcal{K}_\infty(\theta)$ is to express Gauss–Newton curvature in a form where the differential operator $\mathcal{L}$ appears as $\mathcal{A} = \mathcal{L}^*\mathcal{L}$. This will allow us to directly see which "directions" are poorly controlled by residual sampling and how a supervised term changes those directions.

Let $\theta \in \mathbb{R}^p$ be a parameter vector. For $j = 1, \ldots, p$, we define the tangent functions

$$\phi_j(x;\theta) := \frac{\partial}{\partial\theta_j} u(x;\theta), \quad x \in \Omega.$$

The $\phi_j(\cdot;\theta)$'s span the corresponding tangent function space

$$\mathcal{H}(\theta) := \mathrm{span}\{\phi_1(\cdot;\theta), \ldots, \phi_p(\cdot;\theta)\} \subset L^2(\Omega, \mu),$$

where $\mu$ is the reference sampling measure on $\Omega$ (i.e., the distribution used to draw residual and data points).

We then introduce the linear map

$$T(\theta) : \mathbb{R}^p \to \mathcal{H}(\theta), \quad T(\theta)v := \sum_{j=1}^{p} v_j \phi_j(\cdot; \theta),$$

and its adjoint

$$T(\theta)^* : L^2(\Omega, \mu) \to \mathbb{R}^p, \quad \big(T(\theta)^* f\big)_j := \int_\Omega f(x)\, \phi_j(x; \theta)\, d\mu(x).$$

We then define the naturally associated (population) kernel integral operator as:

$$\mathcal{K}_\infty(\theta) := T(\theta)T(\theta)^* : L^2(\Omega, \mu) \to \mathcal{H}(\theta). \tag{B.6}$$

Here $\mathcal{H}(\theta)$ is the tangent function space spanned by $\{\phi_j(\cdot; \theta)\}$, and $\mathcal{K}_\infty(\theta) = T(\theta)T(\theta)^*$ is the corresponding (finite-width) neural tangent kernel operator in the sense of De Ryck et al. (2024); Rathore et al. (2024).

POPULATION PDE OBJECTIVE AND GAUSS–NEWTON MATRIX.

For simplicity, we ignore the boundary term in this subsection. It contributes an additional positive semidefinite Gauss–Newton term, so keeping it separate can only strengthen the spectral lower bounds used later. With this simplification, the population counterpart of the residual loss is given by:

$$L_{\infty,\text{PDE}}(\theta) := \frac{1}{2} \int_\Omega \big(\mathcal{L}[u(x;\theta)] - f(x)\big)^2 d\mu(x). \tag{B.7}$$

Define $\mathcal{A} := \mathcal{L}^*\mathcal{L}$, where $\mathcal{L}^*$ denotes the adjoint of $\mathcal{L}$ with respect to the $L^2(\Omega, \mu)$ inner product.

The population Gauss–Newton matrix for (B.7) is

$$G_{\infty,\text{PDE}}(\theta) := \int_\Omega \mathcal{L}[\nabla_\theta u(x;\theta)]\, \mathcal{L}[\nabla_\theta u(x;\theta)]^\top\, d\mu(x),$$

where $\nabla_\theta u(x;\theta) \in \mathbb{R}^p$ is the gradient of $u$ with respect to $\theta$. A direct computation shows that

$$G_{\infty,\text{PDE}}(\theta) = T(\theta)^* \mathcal{A} T(\theta). \tag{B.8}$$

By (De Ryck et al., 2024), the finite-width population Gauss–Newton matrix $G_{\infty,\text{PDE}}(\theta)$ has the same non-zero eigenvalues as the compact, self-adjoint operator

$$\mathcal{A} \circ \mathcal{K}_\infty(\theta) : \mathcal{H}(\theta) \to \mathcal{H}(\theta), \quad (\mathcal{A} \circ \mathcal{K}_\infty(\theta))f := \mathcal{A}\big(T(\theta)T(\theta)^* f\big),$$

and hence

$$\lambda_j\big(G_{\infty,\text{PDE}}(\theta)\big) = \lambda_j\big(\mathcal{A} \circ \mathcal{K}_\infty(\theta)\big), \quad \kappa\big(G_{\infty,\text{PDE}}(\theta)\big) = \kappa\big(\mathcal{A} \circ \mathcal{K}_\infty(\theta)\big), \tag{B.9}$$

for $j = 1, \ldots, p$. We refer to De Ryck et al. (2024, Thm. 2.4) for the derivation, which is a finite-dimensional analogue of the fact that $S^*S$ and $SS^*$ have the same non-zero eigenvalues for any bounded operator $S$ between Hilbert spaces.

### B.3. Population Gauss–Newton with FK data

We now incorporate the Feynman-Kac supervision term into the population analysis. For simplicity, we assume that residual and data points are sampled from the same measure $\mu$ on $\Omega$, but the more general case follows from the same argument.

The population counterpart of the full loss (B.4) is

$$L_{\infty,\lambda}(\theta) := \frac{1}{2} \int_\Omega \big(\mathcal{L}[u(x;\theta)] - f(x)\big)^2 d\mu(x) + \frac{\lambda}{2} \int_\Omega \big(u(x;\theta) - u^\star(x)\big)^2 d\mu(x). \tag{B.10}$$

The first term yields the Gauss–Newton matrix $T(\theta)^* \mathcal{A} T(\theta)$ as in (B.8). The data term contributes

$$G_{\infty,\text{data}}(\theta) := \lambda \int_\Omega \nabla_\theta u(x;\theta)\, \nabla_\theta u(x;\theta)^\top\, d\mu(x) = \lambda\, T(\theta)^* T(\theta),$$

since $T(\theta)v = \sum_{j=1}^p v_j \phi_j(\cdot; \theta)$ and $\nabla_\theta u(x; \theta) = (\phi_1(x; \theta), \ldots, \phi_p(x; \theta))^\top$.

Thus the combined population Gauss–Newton matrix for $L_{\infty,\lambda}$ is

$$G_{\infty,\lambda}(\theta) := G_{\infty,\mathrm{PDE}}(\theta) + G_{\infty,\mathrm{data}}(\theta) = T(\theta)^* (\mathcal{A} + \lambda I) \, T(\theta). \tag{B.11}$$

Exactly as in the PDE-only case, $G_{\infty,\lambda}(\theta)$ is the matrix representation (in the basis $\{\phi_j(\cdot; \theta)\}_{j=1}^p$) of the compact, self-adjoint operator

$$(\mathcal{A} + \lambda I) \circ \mathcal{K}_\infty(\theta) : \mathcal{H}(\theta) \to \mathcal{H}(\theta),$$

and we obtain from De Ryck et al. (2024, Thm. 2.4) that their non-zero spectra and condition numbers coincide:

$$\lambda_j\big(G_{\infty,\lambda}(\theta)\big) = \lambda_j\big((\mathcal{A} + \lambda I) \circ \mathcal{K}_\infty(\theta)\big), \quad \kappa\big(G_{\infty,\lambda}(\theta)\big) = \kappa\big((\mathcal{A} + \lambda I) \circ \mathcal{K}_\infty(\theta)\big). \tag{B.12}$$

Intuitively, adding the data term simply replaces $\mathcal{A}$ by $\mathcal{A} + \lambda I$ in the operator-theoretic framework of De Ryck et al. (2024), thereby lifting the small eigenvalues associated with ill-conditioned PDE directions. This is the population analogue of Tikhonov regularization or ridge regression, but applied at the level of the PDE operator $\mathcal{A}$ composed with the tangent kernel $\mathcal{K}_\infty(\theta)$. This population calculation motivates what we look for empirically: if the data term contributes a matrix with eigenvalues bounded below independently of the number of residual points, then the sum $G_\lambda = G_{\mathrm{res}} + G_{\mathrm{bc}} + \lambda G_{\mathrm{data}}$ should inherit a uniform lower bound once $\lambda > 0$ is fixed.

## B.4. Finite-sample Gauss–Newton matrices and effective conditioning

We now move from the population operators in (B.8) and (B.11) to their empirical counterparts, and study how the data term affects the spectrum of $G_\lambda$ built from finitely many residual, boundary, and data points.

### EMPIRICAL GAUSS–NEWTON MATRICES

The empirical residual and data Gauss–Newton matrices at a parameter $\theta$ are given by

$$G_{\mathrm{res}}(\theta) := \frac{1}{n_{\mathrm{res}}} \sum_{i=1}^{n_{\mathrm{res}}} \mathcal{L}[\nabla_\theta u(x_i^{\mathrm{res}}; \theta)] \, \mathcal{L}[\nabla_\theta u(x_i^{\mathrm{res}}; \theta)]^\top, \tag{B.13}$$

$$G_{\mathrm{data}}(\theta) := \frac{1}{n_{\mathrm{d}}} \sum_{i=1}^{n_{\mathrm{d}}} \nabla_\theta u(x_i^{\mathrm{d}}; \theta) \, \nabla_\theta u(x_i^{\mathrm{d}}; \theta)^\top. \tag{B.14}$$

The boundary Gauss–Newton term $G_{\mathrm{bc}}(\theta)$ is defined analogously to $G_{\mathrm{res}}(\theta)$, replacing $\mathcal{L}[u] - f$ by $u - g$ on $\partial\Omega$.

The full empirical Gauss–Newton matrix for $L_\lambda$ at $\theta$ is then

$$G_\lambda(\theta) = G_{\mathrm{res}}(\theta) + G_{\mathrm{bc}}(\theta) + \lambda \, G_{\mathrm{data}}(\theta). \tag{B.15}$$

In what follows, we focus on the behavior of $G_\lambda(\theta)$ at a global minimizer $\theta_\lambda^\star \in \Theta_\lambda^\star$, although the analysis applies more generally to any parameter $\theta$ in a region of interest.

### POLYNOMIAL SPECTRAL DECAY FOR THE PDE PART

We assume that at the parameter of interest $\theta_\lambda^\star$ the population operator $\mathcal{A} \circ \mathcal{K}_\infty(\theta_\lambda^\star)$ has polynomially decaying eigenvalues:

$$\lambda_j\big(\mathcal{A} \circ \mathcal{K}_\infty(\theta_\lambda^\star)\big) \leq C \, j^{-2\alpha}, \quad j = 1, 2, \ldots, \tag{B.16}$$

for some constants $C > 0$ and $\alpha > 1/2$. This type of decay is typical for elliptic operators on smooth domains, possibly combined with smoothness properties of the tangent kernel $\mathcal{K}_\infty(\theta_\lambda^\star)$ (see (De Ryck et al., 2024) for some examples).

Theorem 5.3 in the main text is proved in (Rathore et al., 2024, Theorem 8.4) in the same linear PDE and interpolation setting considered here. We will use the following finite-sample spectral scaling for the residual Gauss–Newton matrix at $\theta_\lambda^\star$, stated in the notation of Section B.4.

**Proposition B.2.** *There exist constants $c_{\text{res}} > 0$, $\Lambda_{\text{PDE}} < \infty$ and $\alpha > 1/2$ such that, for every $\delta \in (0,1)$, there is $n_{\text{res}}^{\min}(\delta) \in \mathbb{N}$ with the following property: for all $n_{\text{res}} \geq n_{\text{res}}^{\min}(\delta)$, with probability at least $1 - \delta$ over the sampling of the residual points,*

$$\lambda_{\min}\big(G_{\text{res}}(\theta_\lambda^\star)\big) \ \geq \ c_{\text{res}}\, n_{\text{res}}^{-\alpha}, \quad \lambda_{\max}\big(G_{\text{res}}(\theta_\lambda^\star)\big) \ \leq \ \Lambda_{\text{PDE}}.$$

*Proof.* This is a restatement of (Rathore et al., 2024, Theorem 8.4) in the notation of (B.13), with $N = n_{\text{res}}$. $\qquad\square$

Proposition B.2 captures the deterioration of the residual contribution under mesh refinement, as established in (Rathore et al., 2024). We now turn to the supervised term and its effect on the combined spectrum.

DATA PL* CONDITION AND FINITE-SAMPLE CONCENTRATION

The beneficial effect of the data term on conditioning requires that the tangent features at the data points are sufficiently rich and non-degenerate. Following the PL* viewpoint of (Liu et al., 2022), we encode this via a finite-width PL* result for the data term, Proposition B.3, which yields a uniform spectral bound on the data tangent kernel and its associated empirical Gauss–Newton matrix. Informally, this proposition states that for a sufficiently wide network, the empirical data tangent kernel stays uniformly well conditioned in a ball around initialization, so that the supervised data loss satisfies a uniform PL* inequality with constants independent of $n_{\text{d}}$.

**Proposition B.3.** *Consider the fully-connected network, initialization scheme, and fixed data points $\{x_i^{\text{d}}\}_{i=1}^{n_{\text{d}}} \subset \Omega$ in the setting of Liu et al. (2022, Section 4), and define the rescaled data feature map*

$$\mathcal{F}_{\text{data}}(\theta) := \frac{1}{\sqrt{n_{\text{d}}}}\big(u(x_1^{\text{d}};\theta) - y_1, \ldots, u(x_{n_{\text{d}}}^{\text{d}};\theta) - y_{n_{\text{d}}}\big) \in \mathbb{R}^{n_{\text{d}}}.$$

*Let $\theta_0$ be the random initialization and define the associated tangent kernel $K_{\text{data}}$ by*

$$K_{\text{data}}(\theta) := D\mathcal{F}_{\text{data}}(\theta)\, D\mathcal{F}_{\text{data}}(\theta)^\top \in \mathbb{R}^{n_{\text{d}} \times n_{\text{d}}},$$

*where $D\mathcal{F}_{\text{data}}(\theta) \in \mathbb{R}^{n_{\text{d}} \times p}$ is the Jacobian matrix, and denote $\lambda_0 := \lambda_{\min}\big(K_{\text{data}}(\theta_0)\big)$. Fix any radius $R > 0$, target $\mu_{\text{data}} \in (0, \lambda_0)$, and confidence level $\delta \in (0,1)$. Then there exists a universal constant $C > 0$ such that, if the neural network's width $m$ satisfies*

$$m \ \geq \ C\, \frac{n_{\text{d}}\, R^{6L+2}}{\big(\lambda_0 - \mu_{\text{data}}\big)^2}\, \log\!\Big(\frac{2n_{\text{d}}}{\delta}\Big), \tag{B.17}$$

*then the following holds with probability at least $1 - \delta$ over the random initialization, simultaneously for all $\theta \in B(\theta_0, R)$:*

$$\mu_{\text{data}} \ \leq \ \lambda_{\min}\big(K_{\text{data}}(\theta)\big) \ \leq \ \lambda_{\max}\big(K_{\text{data}}(\theta)\big) \ \leq \ M_{\text{data}}, \tag{B.18}$$

*for some $M_{\text{data}} < \infty$ independent of $n_{\text{d}}$. In particular, the empirical data loss $L_{\text{data}}(\theta) = \frac{1}{2}\big\|\mathcal{F}_{\text{data}}(\theta)\big\|_2^2$ satisfies a $\mu_{\text{data}}$–PL* inequality on $B(\theta_0, R)$. Moreover, $G_{\text{data}}(\theta) = D\mathcal{F}_{\text{data}}(\theta)^\top D\mathcal{F}_{\text{data}}(\theta)$ and $K_{\text{data}}(\theta)$ have the same nonzero eigenvalues, hence the nonzero eigenvalues of $G_{\text{data}}(\theta)$ lie in $[\mu_{\text{data}}, M_{\text{data}}]$. Equivalently, $\lambda_{\min}^+\big(G_{\text{data}}(\theta)\big) \geq \mu_{\text{data}}$ and $\lambda_{\max}\big(G_{\text{data}}(\theta)\big) \leq M_{\text{data}}$.*

*Proof sketch.* This is the specialization of Liu et al. (2022, Theorem 4) to $\mathcal{F} = \mathcal{F}_{\text{data}}$, together with the identification of the PL* constant with $\lambda_{\min}\big(K_{\text{data}}(\theta)\big)$ (Liu et al., 2022, Theorem 1) and the fact that $G_{\text{data}}(\theta)$ and $K_{\text{data}}(\theta)$ have the same non-zero eigenvalues. $\qquad\square$

Fix a confidence level $\delta_{\text{init}} \in (0,1)$ and choose the width $m$ so that (B.17) holds with $\delta = \delta_{\text{init}}$. Let $\mathcal{E}_{\text{init}}$ denote the corresponding high-probability initialization event. By Proposition B.3, on $\mathcal{E}_{\text{init}}$ the following uniform spectral bounds hold on $B(\theta_0, R)$:

$$\lambda_{\min}^+\big(G_{\text{data}}(\theta)\big) \geq \mu_{\text{data}} \text{ and } \lambda_{\max}\big(G_{\text{data}}(\theta)\big) \leq M_{\text{data}} \text{ for all } \theta \in B(\theta_0, R). \tag{B.19}$$

When $p > n_{\text{d}}$, one typically has $\lambda_{\min}\big(G_{\text{data}}(\theta)\big) = 0$, so the relevant notion is the conditioning of $G_{\text{data}}(\theta)$ restricted to its range, quantified by $\lambda_{\min}^+\big(G_{\text{data}}(\theta)\big)$.

At this stage we have a uniform lower bound on the nonzero curvature provided by the data term through (B.19). To ensure that this conditioning transfers to the combined Gauss–Newton matrix, we need a compatibility condition describing how the PDE term acts on directions that are poorly constrained (or unconstrained) by the data tangent features. In particular, we rule out nearly-flat PDE curvature on the data-orthogonal subspace and control the coupling between *data-visible* and *data-invisible* directions. This the object of Assumption B.1 that we now introduce.

In what follows, we will let, for any $\theta \in \mathbb{R}^p$, $P_\theta \in \mathbb{R}^{p \times p}$ denote the projection matrix onto the range of $G_{\text{data}}(\theta)$, and $I \in \mathbb{R}^{p \times p}$ the identity. For $x \in \mathbb{R}^p$ and $A \in \mathbb{R}^{p \times p}$, we will also slightly abuse notation and respectively denote by $\|x\| := \sqrt{x_1^2 + \ldots + x_p^2}$ and $\|A\| := \sup_{v \in \mathbb{R}^p : \|v\| = 1} \|Av\|$ their Euclidean and operator norm.

**Assumption B.1.** Fix a set $\mathcal{S} \subset B(\theta_0, R)$. There exist constants $\delta', \mu_\perp > 0$ and $\rho \geq 0$, such that with probability at least $1 - \delta'$ over the initialization $\theta_0$ and the random sampling of the collocation, boundary and supervised data points, the following holds for every $\theta \in \mathcal{S}$:

$$\lambda_{\min}^+\big((I - P_\theta)G_{\text{PDE}}(\theta)(I - P_\theta)\big) \geq \mu_\perp, \tag{B.20}$$

$$\|P_\theta G_{\text{PDE}}(\theta)(I - P_\theta)\| \leq \rho. \tag{B.21}$$

*Remark* B.4. If the supervised data is not random (e.g., coarse finite element grid), one can formulate Assumption B.1 in a completely analogous manner and reach the same conclusion.

Assumption B.1 allows to bound from below the smallest eigenvalue of $G_\lambda(\theta)$ on $\mathcal{S}$ by an explicit quantity $\gamma(\lambda)$, as we will now establish.

**Proposition B.5.** *Assume that the conditions of Proposition B.3 are satisfied with radius $R > 0$ and confidence level $\delta_{\text{init}}$, and let $\mathcal{S} \subset B(\theta_0, R)$ be a set on the corresponding initialization event. Assume also that Assumption B.1 holds on $\mathcal{S}$. Then, with probability $1 - \delta_{\text{init}} - \delta' \equiv 1 - \delta$ we have for every $\lambda > 0$ and $\theta \in \mathcal{S}$,*

$$\lambda_{\min}^+\big(G_\lambda(\theta)\big) \geq \gamma(\lambda), \tag{B.22}$$

*where*

$$\gamma(\lambda) := \frac{(\lambda\mu_{\text{data}} + \mu_\perp) - \sqrt{(\lambda\mu_{\text{data}} - \mu_\perp)^2 + 4\rho^2}}{2}.$$

*In particular, if $\rho^2 < \lambda\mu_{\text{data}}\mu_\perp$, then $\gamma(\lambda) > 0$ uniformly over $\theta \in \mathcal{S}$.*

*Proof.* Fix $\theta \in \mathcal{S}$. Decompose any $v \in \mathbb{R}^p$ as $v = v_1 + v_2$ with $v_1 = P_\theta v$ and $v_2 = (I - P_\theta)v$. Using $\text{Range}(P_\theta) = \text{Range}(G_{\text{data}}(\theta))$ and (B.19),

$$v^\top(\lambda G_{\text{data}}(\theta))v = \lambda v_1^\top G_{\text{data}}(\theta)v_1 \geq \lambda\mu_{\text{data}}\|v_1\|^2.$$

Similarly, by (B.20), for $v_2$ orthogonal to the kernel of $(I - P_\theta)G_{\text{PDE}}(\theta)(I - P_\theta)$, we have

$$v_2^\top G_{\text{PDE}}(\theta)v_2 = v^\top(I - P_\theta)G_{\text{PDE}}(\theta)(I - P_\theta)v \geq \mu_\perp\|v_2\|^2.$$

For the cross term, by (B.21),

$$2v_1^\top G_{\text{PDE}}(\theta)v_2 \geq -2\|P_\theta G_{\text{PDE}}(\theta)(I - P_\theta)\|\,\|v_1\|\,\|v_2\| \geq -2\rho\,\|v_1\|\,\|v_2\|.$$

Putting these together,

$$v^\top G_\lambda(\theta)v \geq \begin{bmatrix} \|v_1\| \\ \|v_2\| \end{bmatrix}^\top \begin{bmatrix} \lambda\mu_{\text{data}} & -\rho \\ -\rho & \mu_\perp \end{bmatrix} \begin{bmatrix} \|v_1\| \\ \|v_2\| \end{bmatrix}.$$

Minimizing over $\|v\| = 1$ yields the smallest eigenvalue of the displayed $2 \times 2$ matrix, which equals $\gamma(\lambda)$. By the variational formulation of the smallest non-zero eigenvalue, the claimed lower bound on $\lambda_{\min}^+(G_\lambda(\theta))$ follows. $\square$

*Remark* B.6. Note that although we did not include it for simplicity, the boundary loss term has the same least-squares form as a supervised data term, so its Gauss–Newton contribution $G_{\text{bc}}(\theta)$ is positive semidefinite and can be handled in an analogous manner. More specifically, in Proposition B.5 and Theorem B.9 we can include the boundary term while assuming that $G_{\text{bc}}(\theta)$ satisfies Assumption B.1 as well. The analogous conclusion then follows by mirroring the current argument.

## B.5. PL constant, condition number, and gradient descent

We now translate the spectral bounds from Proposition B.5 into a Polyak–Łojasiewicz (PL) constant and condition number for $L_\lambda$, and derive implications for gradient descent.

### PL INEQUALITY AND PL-BASED CONDITION NUMBER

Let $\mathcal{S} \subset \mathbb{R}^p$ be a set containing $\theta_\lambda^\star$. We say that $L_\lambda$ satisfies a $\text{PL}^*$ inequality with constant $\mu_\lambda > 0$ on $\mathcal{S}$ if

$$\frac{\left\|\nabla L_\lambda(\theta)\right\|_2^2}{2\mu_\lambda} \geq L_\lambda(\theta) - L_\lambda(\theta_\lambda^\star), \quad \forall\, \theta \in \mathcal{S}. \tag{B.23}$$

Let $\beta_{L_\lambda}$ denote a smoothness constant on $\mathcal{S}$, i.e. an upper bound on the operator norm of the Hessian,

$$\beta_{L_\lambda} \geq \sup_{\theta \in \mathcal{S}} \left\| H_{L_\lambda}(\theta) \right\|.$$

The PL-based condition number of $L_\lambda$ on $\mathcal{S}$ is then defined by

$$\kappa_{L_\lambda}(\mathcal{S}) := \frac{\beta_{L_\lambda}}{\mu_\lambda}. \tag{B.24}$$

We will use the following Lemma, which is nothing more than (Liu et al., 2022, Theorem 1) reformulated in our setup.

**Lemma B.7.** *Let $L(\theta) = \frac{1}{2}\|F(\theta)\|_2^2$, where $F : \mathbb{R}^p \to \mathbb{R}^n$ is differentiable on a set $\mathcal{S}$. Suppose there exists $\theta^\star \in \mathcal{S}$ with $F(\theta^\star) = 0$. Define the sample tangent kernel $K_F(\theta) := J_F(\theta)\, J_F(\theta)^\top$. If there exists $\mu > 0$ such that $\lambda_{\min}(K_F(\theta)) \geq \mu$ holds for every $\theta \in \mathcal{S}$, then $L$ satisfies the $\text{PL}^*$ inequality on $\mathcal{S}$ with constant $\mu$.*

*Proof.* For $\theta \in \mathcal{S}$, $\nabla L(\theta) = J_F(\theta)^\top F(\theta)$, hence $\|\nabla L(\theta)\|_2^2 = F(\theta)^\top K_F(\theta) F(\theta)$. The assumed bound gives $\|\nabla L(\theta)\|_2^2 \geq \mu\|F(\theta)\|_2^2 = 2\mu L(\theta)$. Since $L(\theta^\star) = 0$, this is the claimed $\text{PL}^*$ inequality. $\square$

From basic linear algebra, we have the following relationship between the non-zero spectrum of $K_F$ and that of $G_F$:

**Lemma B.8.** *Let $F : \mathbb{R}^p \to \mathbb{R}^n$ be differentiable at $\theta$, with Jacobian $J_F(\theta)$. Define $G_F(\theta) := J_F(\theta)^\top J_F(\theta)$ and $K_F(\theta) := J_F(\theta) J_F(\theta)^\top$. Then $G_F(\theta)$ and $K_F(\theta)$ have the same nonzero eigenvalues. If $J_F(\theta)$ has full row rank, then $K_F(\theta)$ is positive definite and $\lambda_{\min}\big(K_F(\theta)\big) = \lambda_{\min}^+\big(G_F(\theta)\big)$.*

### BOUNDING THE PL CONSTANT AND CONDITION NUMBER

Lemma B.7 applied to $F_\lambda$ shows that it suffices to control the smallest eigenvalue of the stacked sample kernel $K_{F_\lambda}(\theta)$ on $\mathcal{S}$ to obtain a $\text{PL}^*$ inequality. Lemma B.8 on the other hand shows that this smallest eigenvalue is equal to the non-zero such eigenvalue of $G_\lambda$. Combining these facts with Proposition B.5, which gives a lower bound on $\lambda_{\min}^+(G_\lambda)$ yields the following uniform bound on the PL-based condition number, which is our main result.

**Theorem B.9.** *Fix confidence levels $\delta_{\text{init}} \in (0,1)$ and $\delta' \in (0,1)$. Choose the width $m$ so that Proposition B.3 holds with $\delta = \delta_{\text{init}}$, and let $\mathcal{S} \subset B(\theta_0, R)$ be any set containing a global minimizer $\theta_\lambda^\star$ of $L_\lambda$ satisfying $F_\lambda(\theta_\lambda^\star) = 0$. Assume that Assumption B.1 holds on $\mathcal{S}$ with confidence parameter $\delta'$, and let $\gamma(\lambda)$ be as in Proposition B.5.*

*Assume that $J_{F_\lambda}$ is Lipschitz on $\mathcal{S}$ with constant $L_J > 0$. Assume also that $J_{F_\lambda}(\theta_\lambda^\star)$ has full row rank, and define*

$$s_\lambda := \sigma_{\min}\big(J_{F_\lambda}(\theta_\lambda^\star)\big), \quad r_\lambda := \frac{s_\lambda}{2L_J}, \quad \text{and } \mathcal{S}_\lambda := \mathcal{S} \cap B(\theta_\lambda^\star, r_\lambda).$$

*Lastly, assume that there exist finite constants $B_F, B_J > 0$ such that $\sup_{\theta \in \mathcal{S}_\lambda} \|F_\lambda(\theta)\|_2 \leq B_F$ and $\sup_{\theta \in \mathcal{S}_\lambda} \|J_{F_\lambda}(\theta)\| \leq B_J$.*

*Then, with probability at least $1 - \delta_{\text{init}} - \delta' \equiv 1 - \delta$, the loss $L_\lambda$ satisfies a $\text{PL}^*$ inequality on $\mathcal{S}_\lambda$ with constant $\mu_\lambda$ obeying*

$$\mu_\lambda \geq \gamma(\lambda). \tag{B.25}$$

*Moreover, $L_\lambda$ is $\beta_{L_\lambda}$-smooth on $\mathcal{S}_\lambda$ with $\beta_{L_\lambda} \leq B_J^2 + L_J B_F$, and its PL-based condition number on $\mathcal{S}_\lambda$ satisfies*

$$\kappa_{L_\lambda}(\mathcal{S}_\lambda) \leq \frac{B_J^2 + L_J B_F}{\gamma(\lambda)}. \tag{B.26}$$

*Proof.* Work on the intersection of the initialization event from Proposition B.3 with $\delta = \delta_{\text{init}}$ and the event in Assumption B.1 on $\mathcal{S}$. This intersection event has probability at least $1 - \delta_{\text{init}} - \delta'$, and on it Proposition B.5 gives $\lambda^+_{\min}\big(G_\lambda(\theta)\big) \geq \gamma(\lambda)$ for every $\theta \in \mathcal{S}$.

We next propagate full row rank from $\theta^\star_\lambda$ to $\mathcal{S}_\lambda$. For any $\theta \in \mathcal{S}_\lambda$, Weyl's inequalities for singular value perturbations (Horn & Johnson, 2012) yields

$$\sigma_{\min}\big(J_{F_\lambda}(\theta)\big) \geq \sigma_{\min}\big(J_{F_\lambda}(\theta^\star_\lambda)\big) - \big\|J_{F_\lambda}(\theta) - J_{F_\lambda}(\theta^\star_\lambda)\big\| \geq s_\lambda - L_J\|\theta - \theta^\star_\lambda\| \geq \frac{s_\lambda}{2}.$$

Therefore $J_{F_\lambda}(\theta)$ has full row rank for every $\theta \in \mathcal{S}_\lambda$. Lemma B.8 then implies that $\lambda_{\min}\big(K_{F_\lambda}(\theta)\big) = \lambda^+_{\min}\big(G_\lambda(\theta)\big)$ holds for every $\theta \in \mathcal{S}_\lambda$, hence $\lambda_{\min}\big(K_{F_\lambda}(\theta)\big) \geq \gamma(\lambda)$ on $\mathcal{S}_\lambda$. Applying Lemma B.7 to $L_\lambda(\theta) = \frac{1}{2}\|F_\lambda(\theta)\|_2^2$ and $F_\lambda(\theta^\star_\lambda) = 0$ yields the PL* inequality on $\mathcal{S}_\lambda$ with constant $\mu_\lambda \geq \gamma(\lambda)$, which is (B.25).

For smoothness on $\mathcal{S}_\lambda$, use $\nabla L_\lambda(\theta) = J_{F_\lambda}(\theta)^\top F_\lambda(\theta)$ to write, for $\theta, \theta' \in \mathcal{S}_\lambda$,

$$\nabla L_\lambda(\theta) - \nabla L_\lambda(\theta') = \big(J_{F_\lambda}(\theta) - J_{F_\lambda}(\theta')\big)^\top F_\lambda(\theta) + J_{F_\lambda}(\theta')^\top \big(F_\lambda(\theta) - F_\lambda(\theta')\big).$$

The first term is bounded by $L_J\|\theta - \theta'\|\,\|F_\lambda(\theta)\|_2 \leq L_J B_F\|\theta - \theta'\|$. For the second term, the mean value inequality gives $\|F_\lambda(\theta) - F_\lambda(\theta')\|_2 \leq \sup_{\xi \in \mathcal{S}_\lambda}\|J_{F_\lambda}(\xi)\|\,\|\theta - \theta'\| \leq B_J\|\theta - \theta'\|$, hence the second term is bounded by $B_J^2\|\theta - \theta'\|$. Altogether,

$$\|\nabla L_\lambda(\theta) - \nabla L_\lambda(\theta')\| \leq \big(B_J^2 + L_J B_F\big)\|\theta - \theta'\| \text{ for all } \theta, \theta' \in \mathcal{S}_\lambda.$$

Therefore $L_\lambda$ is $\beta_{L_\lambda}$-smooth on $\mathcal{S}_\lambda$ with $\beta_{L_\lambda} \leq B_J^2 + L_J B_F$. Combining this with (B.25) and the definition (B.24) yields (B.26). $\square$

In the PDE-only case $\lambda = 0$, Proposition B.2 gives $\mu_0 \asymp n_{\text{res}}^{-\alpha}$, so the PL based condition number $\kappa_{L_0}(\mathcal{S})$ grows like $n_{\text{res}}^\alpha$. In contrast, (B.25)–(B.26) show that if $\lambda > \rho^2/(\mu_{\text{data}}\mu_\perp)$ is fixed independently of $n_{\text{res}}$, then $\mu_\lambda$ is bounded away from zero and $\kappa_{L_\lambda}(\mathcal{S})$ remains bounded as more residual points are added.

GRADIENT DESCENT CONVERGENCE UNDER UNIFORM CONDITIONING

Consider gradient descent on $L_\lambda$,

$$\theta_{k+1} = \theta_k - \eta\,\nabla L_\lambda(\theta_k),$$

with a fixed step size $\eta \in (0, 1/\beta_{L_\lambda}]$, and assume that the iterates stay in $\mathcal{S}_\lambda$. Under the setting of Theorem B.9, the result of Liu et al. (2022, Theorem 6) gives

$$L_\lambda(\theta_{k+1}) - L_\lambda(\theta^\star_\lambda) \leq \big(1 - \eta\mu_\lambda\big)\big(L_\lambda(\theta_k) - L_\lambda(\theta^\star_\lambda)\big).$$

Choosing $\eta = 1/\beta_{L_\lambda}$ and using (B.24)–(B.26), we obtain

$$L_\lambda(\theta_{k+1}) - L_\lambda(\theta^\star_\lambda) \leq \left(1 - \frac{1}{\kappa_{L_\lambda}(\mathcal{S})}\right)\big(L_\lambda(\theta_k) - L_\lambda(\theta^\star_\lambda)\big).$$

Thus the number of iterations required to reach $L_\lambda(\theta_k) - L_\lambda(\theta^\star_\lambda) \leq \varepsilon$ is

$$\mathcal{O}\big(\kappa_{L_\lambda}(\mathcal{S})\log(1/\varepsilon)\big).$$

In particular, in virtue of Theorem B.9, we see that this iteration complexity is bounded uniformly in $n_{\text{res}}$ when $\lambda > 0$ is fixed.

# C. Monte Carlo Feynman–Kac label noise

This section provides the probabilistic error analysis for the Monte Carlo Feynman–Kac estimator used to construct the FK supervision labels in $\mathcal{R}_{\text{FK-PINN}}$. Our goal is to justify Proposition 6.1 in the main text, namely that each FK datum can be written as a noisy point evaluation of the true solution $u^\star$ with sub-exponential noise whose parameters do not depend on the number of FK points.

Throughout this section, we fix a spatial point $x \in \Omega$ and study the error of the Monte Carlo Feynman–Kac estimator at this point. All constants may depend on the model data (domain, coefficients, source and boundary terms) and, when a truncation horizon $T_{\max}$ is present, on $T_{\max}$. They do *not* depend on the Monte Carlo sample size $N_{\text{MC}}$ and they do not depend on the time step $\Delta t$ once $\Delta t$ is restricted to be smaller than a suitable threshold.

## C.1. Setting and numerical scheme

We recall the diffusion setting that underlies the Feynman–Kac representation.

Let $\Omega \subset \mathbb{R}^d$ be a bounded domain with $C^2$ boundary $\partial\Omega$. Let $b : \mathbb{R}^d \to \mathbb{R}^d$ and $\sigma : \mathbb{R}^d \to \mathbb{R}^{d \times m}$ be globally Lipschitz functions. We assume that $\sigma$ is uniformly elliptic, i.e. there exist constants $0 < \underline{\lambda} \leq \overline{\lambda} < \infty$ such that

$$\underline{\lambda} |v|^2 \ \leq \ v^\top (\sigma\sigma^\top)(x)v \ \leq \ \overline{\lambda} |v|^2, \quad \text{for all } x, v \in \mathbb{R}^d.$$

Let $(W_t)_{t \geq 0}$ be an $m$-dimensional Brownian motion on some filtered probability space $(\Omega, \mathcal{F}, (\mathcal{F}_t)_{t \geq 0}, \mathbb{P})$.

The diffusion process $X = (X_t)_{t \geq 0}$ is the unique strong solution of

$$dX_t = b(X_t)\, dt + \sigma(X_t)\, dW_t, \quad X_0 = x \in \Omega. \tag{C.1}$$

We denote by

$$\tau := \inf\{t \geq 0 : X_t \notin \Omega\}$$

the first exit time of the diffusion from $\Omega$, and assume $\tau < \infty$ almost surely for all starting points $x \in \Omega$.

Let $f : \overline{\Omega} \to \mathbb{R}$, $g : \partial\Omega \to \mathbb{R}$ and $V : \overline{\Omega} \to \mathbb{R}$ be bounded measurable functions. In the main text these correspond to the source term, boundary data and zeroth-order potential of the elliptic boundary value problem. The associated (continuous-time) Feynman–Kac payoff is

$$Y := \int_0^\tau \exp\left(- \int_0^t V(X_s)\, ds\right) f(X_t)\, dt \ + \ \exp\left(- \int_0^\tau V(X_s)\, ds\right) g(X_\tau). \tag{C.2}$$

Under standard regularity and well-posedness assumptions (Evans, 2022), the elliptic boundary value problem admits a unique solution $u^\star$, and one has the Feynman–Kac representation

$$u^\star(x) \ = \ \mathbb{E}^x[Y],$$

where $\mathbb{E}^x$ denotes expectation for the diffusion (C.1) started at $X_0 = x$.

For the numerical approximation we fix a time step $\Delta t > 0$ and define the Euler–Maruyama scheme $(X_k^{\Delta t})_{k \in \mathbb{N}_0}$ by

$$X_0^{\Delta t} = x, \quad X_{k+1}^{\Delta t} = X_k^{\Delta t} + b(X_k^{\Delta t})\, \Delta t + \sigma(X_k^{\Delta t})\, \Delta W_k, \tag{C.3}$$

where $\Delta W_k := W_{(k+1)\Delta t} - W_{k\Delta t}$. The corresponding discrete exit time is

$$\tau_{\Delta t} := \inf\{k\Delta t : X_k^{\Delta t} \notin \Omega\}, \tag{C.4}$$

with the convention $\inf \emptyset = \infty$. Writing

$$N_{\Delta t} := \inf\{k \in \mathbb{N}_0 : X_k^{\Delta t} \notin \Omega\},$$

we have $\tau_{\Delta t} = N_{\Delta t}\Delta t$.

The discretized payoff functional is

$$Y^{\Delta t} := \sum_{k=0}^{N_{\Delta t}-1} \exp\Big(-\Delta t \sum_{j=0}^{k-1} V(X_j^{\Delta t})\Big) f(X_k^{\Delta t})\, \Delta t \;+\; \exp\Big(-\Delta t \sum_{j=0}^{N_{\Delta t}-1} V(X_j^{\Delta t})\Big) g(X_{N_{\Delta t}}^{\Delta t}). \tag{C.5}$$

The associated time-discretized value at $x$ is

$$u_{\Delta t}(x) := \mathbb{E}^x[Y^{\Delta t}].$$

Given $N_{\mathrm{MC}} \in \mathbb{N}$, we consider i.i.d. copies $(Y^{\Delta t,(i)})_{i=1}^{N_{\mathrm{MC}}}$ of $Y^{\Delta t}$ and define the Monte Carlo estimator (for fixed $x$ and $\Delta t$)

$$\widehat{u}^{\mathrm{MC}}(x) := \frac{1}{N_{\mathrm{MC}}} \sum_{i=1}^{N_{\mathrm{MC}}} Y^{\Delta t,(i)}. \tag{C.6}$$

In practice Algorithm 1 simulates trajectories until they exit $\Omega$ or until a truncation horizon $T_{\max} \in (0,\infty]$ is reached. For $T_{\max} = \infty$ this reduces to the infinite horizon estimator $\widehat{u}^{\mathrm{MC}}(x)$ defined above. For a finite truncation horizon we set $N_T := \lfloor T_{\max}/\Delta t \rfloor$ and define the truncated discrete stopping index

$$\kappa_{\Delta t}^{T_{\max}} := \min\{N_{\Delta t}, N_T\},$$

together with the truncated payoff

$$Y^{\Delta t, T_{\max}} := \sum_{k=0}^{\kappa_{\Delta t}^{T_{\max}}-1} \exp\Big(-\Delta t \sum_{j=0}^{k-1} V(X_j^{\Delta t})\Big) f(X_k^{\Delta t})\, \Delta t \;+\; \exp\Big(-\Delta t \sum_{j=0}^{\kappa_{\Delta t}^{T_{\max}}-1} V(X_j^{\Delta t})\Big) g(X_{\kappa_{\Delta t}^{T_{\max}}}^{\Delta t}). \tag{C.7}$$

We write $u_{\Delta t}^{T_{\max}}(x) := \mathbb{E}^x[Y^{\Delta t, T_{\max}}]$ for the corresponding truncated value. For $T_{\max} = \infty$ we adopt the convention $Y^{\Delta t,\infty} := Y^{\Delta t}$ and $u_{\Delta t}^\infty(x) := u_{\Delta t}(x)$ so that the notation covers both finite and infinite horizons. Our aim is to control the tails of FK estimators of the form $\widehat{u}^{\mathrm{MC}}(x)$ and of their truncated counterparts, and hence show that the resulting FK labels behave like noisy point evaluations of $u^\star$ with sub-exponential noise.

## C.2. Exit time estimates and exponential moments

We first establish exponential moments for the continuous exit time $\tau$ and the discrete exit time $\tau_{\Delta t}$. These are the key inputs for controlling the tails of the payoffs $Y$ and $Y^{\Delta t}$ and hence the Monte Carlo error.

### C.2.1. CONTINUOUS EXIT TIME

**Theorem C.1** (Exponential moments of the continuous exit time). *Suppose that $\Omega \subset \mathbb{R}^d$ is a bounded $C^2$ domain and that the diffusion $X$ defined by (C.1) has globally Lipschitz coefficients and is uniformly elliptic. Then there exist constants $\lambda_c > 0$ and $C_c < \infty$ such that*

$$\sup_{y \in \overline{\Omega}} \mathbb{E}^y\big[e^{\lambda_c \tau}\big] \;\le\; C_c.$$

*In particular, the law of $\tau$ has exponential tails uniformly over starting points $y \in \overline{\Omega}$.*

*Proof.* Under these assumptions, the generator $\mathcal{L}$ of $X$ with homogeneous Dirichlet boundary conditions on $\partial\Omega$ admits a discrete spectrum with strictly positive first eigenvalue $\lambda_1 > 0$ (Pinsky, 1995). The killed semigroup $(P_t^\Omega)_{t\geq 0}$ therefore satisfies

$$\|P_t^\Omega\|_{L^\infty(\Omega)\to L^\infty(\Omega)} \le C e^{-\lambda_1 t}, \quad \text{for all } t \geq 0$$

for some constant $C < \infty$. It follows that

$$\sup_{y \in \overline{\Omega}} \mathbb{P}^y(\tau > t) = \sup_{y \in \overline{\Omega}} P_t^\Omega \mathbf{1}(y) \le C e^{-\lambda_1 t}, \quad \text{for all } t \geq 0.$$

Fix any $\lambda_c \in (0, \lambda_1)$. Then, integrating the tail bound,

$$\mathbb{E}^y\big[e^{\lambda_c \tau}\big] = 1 + \int_0^\infty \lambda_c e^{\lambda_c t} \mathbb{P}^y(\tau > t)\, dt \le 1 + C\lambda_c \int_0^\infty e^{-(\lambda_1 - \lambda_c)t}\, dt < \infty,$$

with the bound uniform in $y \in \overline{\Omega}$. This proves the claim for a suitable constant $C_c$. $\square$

C.2.2. DISCRETE EXIT TIME

We next recall moment estimates for the discrete exit time $\tau_{\Delta t}$. We rely on the results of Bouchard et al. (2017).

**Theorem C.2** (Moments of continuous and discrete exit times). *Assume that $\Omega \subset \mathbb{R}^d$ is a bounded $C^2$ domain with non-characteristic boundary, and that the coefficients $b, \sigma$ in (C.1) are globally Lipschitz and of linear growth. Let $\tau$ be as above and $\tau_{\Delta t}$ be defined by (C.4). Then, for each integer $p \geq 1$, there exist finite constants $C_p < \infty$ and $\Delta t_p > 0$ such that for all $0 < \Delta t \leq \Delta t_p$,*

$$\sup_{y \in \overline{\Omega}} \mathbb{E}^y[\tau^p] \leq C_p,$$

$$\sup_{y \in \overline{\Omega}} \mathbb{E}^y[\tau_{\Delta t}^p] \leq C_p,$$

$$\sup_{y \in \overline{\Omega}} \mathbb{E}^y\big[|\tau_{\Delta t} - \tau|^p\big] \leq C_p \, \Delta t^{p/2}.$$

*Proof.* This is a specialization of the main results in Bouchard et al. (2017). See in particular Theorems 3.1 there, specialized to globally Lipschitz coefficients and a bounded $C^2$ domain. The non-characteristic boundary condition is implied by uniform ellipticity. Collecting the corresponding bounds yields the stated result. □

Combining Theorems C.1 and C.2, one can show that $\tau_{\Delta t}$ admits exponential moments uniformly over $\Delta t$ small enough by a standard renewal argument. We sketch this in the next two lemmas.

**Lemma C.3** (Uniform exit in finite time). *Under the standing assumptions, there exist constants $T_* > 0$ and $p_0 \in (0,1)$ such that*

$$\inf_{y \in \overline{\Omega}} \mathbb{P}^y(\tau \leq T_*) \ \geq \ p_0.$$

*Moreover, there exist $\Delta t_0 > 0$ and $p \in (0,1)$ such that for all $0 < \Delta t \leq \Delta t_0$,*

$$\inf_{y \in \overline{\Omega}} \mathbb{P}^y(\tau_{\Delta t} \leq T_*) \ \geq \ p.$$

*Proof sketch.* For the continuous exit time, Theorem C.1 yields $\sup_{y \in \overline{\Omega}} \mathbb{P}^y(\tau > t) \leq C_c e^{-\lambda_c t}$, hence, choosing $T_*$ large enough that $C_c e^{-\lambda_c T_*} \leq 1/2$ gives

$$\inf_{y \in \overline{\Omega}} \mathbb{P}^y(\tau \leq T_*) \geq 1/2 =: p_0.$$

By Theorem C.2, for some $p \geq 1$ we have $\sup_{y \in \overline{\Omega}} \mathbb{E}^y\big[|\tau_{\Delta t} - \tau|^p\big] \to 0$ as $\Delta t \to 0$, hence, for any $\varepsilon > 0$,

$$\sup_{y \in \overline{\Omega}} \mathbb{P}^y\big(|\tau_{\Delta t} - \tau| > \varepsilon\big) \leq \varepsilon^{-p} \sup_y \mathbb{E}^y\big[|\tau_{\Delta t} - \tau|^p\big] \to 0,$$

so $\tau_{\Delta t} \to \tau$ in probability, uniformly in $y$. Since the law of $\tau$ has a density (see, e.g., Pinsky (1995)), we have $\mathbb{P}^y(\tau = T_*) = 0$ for all $y$, and thus $\mathbb{P}^y(\tau_{\Delta t} \leq T_*) \longrightarrow \mathbb{P}^y(\tau \leq T_*)$ uniformly in $y \in \overline{\Omega}$ as $\Delta t \to 0$. Hence for $\Delta t$ small enough we still have

$$\inf_{y \in \overline{\Omega}} \mathbb{P}^y(\tau_{\Delta t} \leq T_*) \geq p_0/2 =: p \in (0,1).$$

□

**Proposition C.4** (Exponential moments of the discrete exit time). *Under the standing assumptions, there exist constants $\lambda_d > 0$, $C_d < \infty$ and $\Delta t_0 > 0$ such that*

$$\sup_{y \in \overline{\Omega}, \, 0 < \Delta t \leq \Delta t_0} \mathbb{E}^y\big[e^{\lambda_d \tau_{\Delta t}}\big] \ \leq \ C_d.$$

*Proof sketch.* Let $T_* > 0$, $p \in (0,1)$ and $\Delta t_0 > 0$ be as in Lemma C.3. In particular, $\inf_y \mathbb{P}^y(\tau_{\Delta t} \leq T_*) \geq p$, so $\sup_y \mathbb{P}^y(\tau_{\Delta t} > T_*) \leq 1 - p$ for all $0 < \Delta t \leq \Delta t_0$.

Consider the continuous-time interpolation of the Euler scheme, which is a time-homogeneous Markov process. By the (strong) Markov property at times $kT_*$,

$$\mathbb{P}^y(\tau_{\Delta t} > (k+1)T_*) = \mathbb{E}^y\big[\mathbf{1}_{\{\tau_\Delta t > kT_*\}}\mathbb{P}^{X^\Delta t_{kT_*}}(\tau_\Delta t > T_*)\big] \le (1-p)\mathbb{P}^y(\tau_\Delta t > kT_*).$$

Iterating this inequality gives $\sup_{y\in\overline{\Omega}}\mathbb{P}^y(\tau_{\Delta t} > kT_*) \le (1-p)^k$ for all $k \in \mathbb{N}$, $0 < \Delta t \le \Delta t_0$. For any $t \ge 0$ with $kT_* \le t < (k+1)T_*$, we have $\{\tau_\Delta t > t\} \subset \{\tau_\Delta t > kT_*\}$, so the same bound holds with $t$ in place of $kT_*$.

Proceeding as in the proof of Theorem C.1, we use the representation $\mathbb{E}^y[e^{\lambda\tau_\Delta t}] = 1 + \int_0^\infty \lambda e^{\lambda t}\mathbb{P}^y(\tau_\Delta t > t)\,dt$, and split the integral over the intervals $[kT_*, (k+1)T_*)$, yielding a geometric series upper bound by the previous paragraph.

Choosing $\lambda_d > 0$ so small that $e^{\lambda_d T_*}(1-p) < 1$, the series in question converges and its sum is finite and independent of $y$ and $\Delta t \in (0, \Delta t_0]$. This gives the claimed bound with some finite constant $C_d$. $\qquad\square$

### C.3. Exponential moments and tails for the payoff

We now transfer the exponential-moment bounds for $\tau$ and $\tau_{\Delta t}$ to the payoffs $Y$ and $Y^{\Delta t}$.

**Lemma C.5** (Deterministic bound for $Y^{\Delta t}$). *Assume that $f$ and $g$ are bounded and that $V \ge 0$ on $\overline{\Omega}$. Then there exist constants $C_f, C_g \ge 0$ such that respectively for all $x \in \overline{\Omega}$, $z \in \partial\Omega$, we have $|f(x)| \le C_f$, $|g(z)| \le C_g$, and*

$$|Y^{\Delta t}| \le C_f\tau_{\Delta t} + C_g \text{ almost surely.}$$

*Proof.* By boundedness of $f$ and $g$ there exist $C_f, C_g \ge 0$ with the stated properties. Since $V \ge 0$, all exponential factors in (C.5) are bounded by one. Thus

$$|Y^{\Delta t}| \le \sum_{k=0}^{N_{\Delta t}-1} |f(X_k^{\Delta t})|\,\Delta t + |g(X_{N_{\Delta t}}^{\Delta t})|$$

$$\le C_f N_{\Delta t}\Delta t + C_g = C_f\tau_{\Delta t} + C_g.$$

$\qquad\square$

A similar argument yields a deterministic bound for $Y$ in terms of $\tau$:

**Lemma C.6** (Deterministic bound for $Y$). *Under the same assumptions as in Lemma C.5, one has*

$$|Y| \le C_f\tau + C_g \text{ almost surely.}$$

Combining these bounds with Theorem C.1 and Proposition C.4, we obtain exponential moments for the payoffs.

**Proposition C.7** (Exponential moments for $Y$ and $Y^{\Delta t}$). *Under the standing assumptions and assuming in addition that $V \ge 0$ on $\overline{\Omega}$, there exist constants $\lambda_Y > 0$, $C_Y < \infty$ and $\Delta t_0 > 0$ such that*

$$\sup_{y\in\overline{\Omega}}\mathbb{E}^y\big[e^{\lambda_Y|Y|}\big] \le C_Y,$$

*and*

$$\sup_{y\in\overline{\Omega},\,0<\Delta t\le\Delta t_0}\mathbb{E}^y\big[e^{\lambda_Y|Y^{\Delta t}|}\big] \le C_Y.$$

*Proof.* By Lemma C.5,

$$e^{\lambda|Y^{\Delta t}|} \le e^{\lambda C_g}e^{\lambda C_f\tau_\Delta t},$$

and by Lemma C.6,

$$e^{\lambda|Y|} \le e^{\lambda C_g}e^{\lambda C_f\tau}.$$

Using Theorem C.1 and Proposition C.4, choose $\lambda_Y > 0$ small enough that $\lambda_Y C_f < \lambda_c$ and $\lambda_Y C_f < \lambda_d$. Then the stated bounds follow immediately, with

$$C_Y := e^{\lambda_Y C_g}\max\{C_c, C_d\}.$$

$\qquad\square$

*Remark* C.8 (Sub-exponential character of $Y$ and $Y^{\Delta t}$). A real-valued random variable $Z$ is sub-exponential if there exists $\alpha > 0$ with $\mathbb{E}[e^{\lambda Z}] < \infty$ for all $|\lambda| < \alpha$. Proposition C.7 shows that $Y$ and $Y^{\Delta t}$ are sub-exponential, with sub-exponential norms bounded uniformly in $\Delta t$ for $\Delta t \leq \Delta t_0$.

### C.4. Concentration for the Monte Carlo estimator at fixed $\Delta t$

We now derive non-asymptotic tail bounds for the Monte Carlo estimator $\widehat{u}^{\mathrm{MC}}(x)$ at a fixed time step $\Delta t \leq \Delta t_0$. The key tool is a Bernstein-type inequality for sub-exponential random variables.

#### C.4.1. A BERNSTEIN INEQUALITY FOR SUB-EXPONENTIAL VARIABLES

**Theorem C.9** (Bernstein inequality for sub-exponential variables). *Let $Z_1, \ldots, Z_N$ be i.i.d. real-valued random variables with $\mathbb{E}[Z_i] = 0$ and suppose there exist constants $\nu^2, b > 0$ satisfying*

$$\mathbb{E}\big[e^{\lambda Z_i}\big] \leq \exp\Big(\frac{\nu^2 \lambda^2}{2}\Big) \text{ for all } |\lambda| < 1/b.$$

*Then there exists a universal constant $c > 0$ such that, for all $t > 0$,*

$$\mathbb{P}\Big(\Big|\frac{1}{N}\sum_{i=1}^{N} Z_i\Big| > t\Big) \leq 2\exp\Big(-cN\min\Big(\frac{t^2}{\nu^2}, \frac{t}{b}\Big)\Big).$$

*Proof.* See, for example, (Vershynin, 2018, Theorem 2.9.1). □

#### C.4.2. APPLICATION TO THE EULER PAYOFF

We apply Theorem C.9 to the centered Euler payoff $Y^{\Delta t} - u_{\Delta t}(x)$.

**Lemma C.10** (Sub-exponential parameters for $Y^{\Delta t} - u_{\Delta t}(x)$). *Under the standing assumptions, there exist constants $\nu > 0$, $b > 0$ and $\Delta t_0 > 0$ such that, for all $\Delta t \leq \Delta t_0$, the centered random variable $Z^{\Delta t} := Y^{\Delta t} - u_{\Delta t}(x)$ satisfies*

$$\mathbb{E}\big[e^{\lambda Z^{\Delta t}}\big] \leq \exp\Big(\frac{\nu^2 \lambda^2}{2}\Big) \text{ for all } |\lambda| < 1/b.$$

*The constants $\nu$ and $b$ depend only on the model data and can be chosen independent of $\Delta t \leq \Delta t_0$.*

*Proof.* Proposition C.7 guarantees that there exist $\lambda_Y > 0$ and $C_Y < \infty$ such that

$$\sup_{0 < \Delta t \leq \Delta t_0} \mathbb{E}\big[e^{\lambda_Y |Y^{\Delta t}|}\big] \leq C_Y.$$

This implies that the moment generating function $\mathbb{E}[e^{\lambda Y^{\Delta t}}]$ is finite on $(-\lambda_Y, \lambda_Y)$ uniformly in $\Delta t \leq \Delta t_0$. Since $Z^{\Delta t} = Y^{\Delta t} - u_{\Delta t}(x)$ differs from $Y^{\Delta t}$ by a constant shift, the same holds for $Z^{\Delta t}$.

A standard argument (see Vershynin (2018, Section 2.7)) shows that there exist parameters $(\nu, b)$ depending only on the exponential-moment bounds of $Y^{\Delta t}$ such that

$$\mathbb{E}\big[e^{\lambda Z^{\Delta t}}\big] \leq \exp\Big(\frac{\nu^2 \lambda^2}{2}\Big) \text{ for all } |\lambda| < 1/b,$$

uniformly over $\Delta t \leq \Delta t_0$. We refer the reader to said reference for details. □

Combining Lemma C.10 with Theorem C.9 yields the following concentration inequality.

**Proposition C.11** (Concentration for the Monte Carlo error at fixed $\Delta t$). *Under the standing assumptions, there exist constants $c > 0$, $\nu > 0$, $b > 0$ and $\Delta t_0 > 0$ such that for all $\Delta t \leq \Delta t_0$, all $N_{\mathrm{MC}} \in \mathbb{N}$ and all $t > 0$,*

$$\mathbb{P}\Big(|\widehat{u}^{\mathrm{MC}}(x) - u_{\Delta t}(x)| > t\Big) \leq 2\exp\Big(-cN_{\mathrm{MC}}\min\Big(\frac{t^2}{\nu^2}, \frac{t}{b}\Big)\Big).$$

*Proof.* For fixed $x$, $\Delta t$ and $N_{\text{MC}}$, define $Z_i^{\Delta t} := Y^{\Delta t,(i)} - u_{\Delta t}(x)$ for $i = 1, \ldots, N_{\text{MC}}$. These variables are i.i.d. with zero mean, and by Lemma C.10 they satisfy the moment generating function condition of Theorem C.9 with parameters $(\nu, b)$ independent of $\Delta t \le \Delta t_0$. Since

$$\widehat{u}^{\text{MC}}(x) - u_{\Delta t}(x) = \frac{1}{N_{\text{MC}}} \sum_{i=1}^{N_{\text{MC}}} Z_i^{\Delta t},$$

Theorem C.9 yields the stated inequality. □

*Remark* C.12 (sub-Gaussian vs exponential regimes). The bound in Proposition C.11 has two regimes. For deviations $0 < t \le \nu^2/b$, the quadratic term dominates and the tails behave essentially like $\exp(-cN_{\text{MC}}t^2/\nu^2)$. For larger deviations, the linear term dominates and the decay is of order $\exp(-cN_{\text{MC}}t/b)$. In both cases the constants are independent of $N_{\text{MC}}$ and of $\Delta t$ for $\Delta t \le \Delta t_0$.

## C.5. Discretization bias

We next bound the bias $u_{\Delta t}(x) - u^\star(x)$ of the time-discretized FK value.

**Theorem C.13** (Weak error for killed Euler schemes). *Assume that $\Omega \subset \mathbb{R}^d$ is a bounded $C^2$ domain with non-characteristic boundary, that the coefficients $b, \sigma$ in (C.1) are $C_b^4$ with bounded derivatives up to order $4$, and that $f, g, V$ are bounded with bounded derivatives up to the necessary order. Then there exist constants $C_{\text{bias}} < \infty$ and $\Delta t_0 > 0$ such that, for all $0 < \Delta t \le \Delta t_0$,*

$$|u_{\Delta t}(x) - u^\star(x)| \le C_{\text{bias}} \sqrt{\Delta t}.$$

*Proof.* This is a specialization of the weak error estimates for killed Euler schemes in (Gobet, 2000, Theorem 2.4) and (Gobet & Menozzi, 2010, Theorem 17). Our assumptions on the regularity of $b, \sigma, f, g, V$ and on the geometry of $\Omega$ are stronger than those in the cited papers, so their hypotheses are satisfied. The functional $Y^{\Delta t}$ in (C.5) matches the discretization considered there. The stated bound follows, with a constant $C_{\text{bias}}$ independent of $\Delta t$ for $\Delta t$ small enough. □

**Lemma C.14** (Bias due to time truncation). *Assume the standing assumptions of this appendix and those of Lemma C.5. Then there exist constants $\tilde{\lambda} > 0$, $C_T < \infty$ and $\Delta t_0 > 0$ such that, for all $x \in \Omega$, all $0 < \Delta t \le \Delta t_0$ and all $T_{\max} > 0$,*

$$|u_{\Delta t}^{T_{\max}}(x) - u_{\Delta t}(x)| \le C_T e^{-\tilde{\lambda} T_{\max}}.$$

*Proof.* On the event $\{\tau_{\Delta t} \le T_{\max}\}$ the discrete trajectory exits $\Omega$ before the truncation horizon and $\kappa_{\Delta t}^{T_{\max}} = N_{\Delta t}$, hence $Y^{\Delta t, T_{\max}} = Y^{\Delta t}$. On the complement $\{\tau_{\Delta t} > T_{\max}\}$ the payoffs $Y^{\Delta t}$ and $Y^{\Delta t, T_{\max}}$ differ only through the contribution of the running functional $f$ after time $T_{\max}$ and through the terminal term that involves $g$. Using the representation (C.5) and the definition (C.7), together with boundedness of $f$ and $g$ and the fact that all exponential factors are at most one, we obtain

$$|Y^{\Delta t} - Y^{\Delta t, T_{\max}}| \le C_f \tau_{\Delta t} \mathbf{1}_{\{\tau_{\Delta t} > T_{\max}\}} + 2C_g \mathbf{1}_{\{\tau_{\Delta t} > T_{\max}\}},$$

with the constants $C_f$ and $C_g$ from Lemma C.5. Taking expectations and using Proposition C.4, for any $\lambda \in (0, \lambda_d)$,

$$\mathbb{P}^x(\tau_{\Delta t} > t) \le e^{-\lambda t} \mathbb{E}^x[e^{\lambda \tau_{\Delta t}}] \le C_d e^{-\lambda t},$$

and therefore

$$\mathbb{E}^x[\tau_{\Delta t} \mathbf{1}_{\{\tau_{\Delta t} > T_{\max}\}}] = \int_{T_{\max}}^\infty \mathbb{P}^x(\tau_{\Delta t} > u)\, du \le \frac{C_d}{\lambda} e^{-\lambda T_{\max}},$$

as well as

$$\mathbb{P}^x(\tau_{\Delta t} > T_{\max}) \le C_d e^{-\lambda T_{\max}}.$$

Combining these bounds yields

$$|u_{\Delta t}^{T_{\max}}(x) - u_{\Delta t}(x)| = |\mathbb{E}^x[Y^{\Delta t, T_{\max}} - Y^{\Delta t}]| \le \left(\frac{C_f C_d}{\lambda} + 2C_g C_d\right) e^{-\lambda T_{\max}}.$$

The claim follows by setting $\tilde{\lambda} := \lambda$ and $C_T := \frac{C_f C_d}{\lambda} + 2C_g C_d$, and by choosing $\Delta t_0$ as in Proposition C.4. □

These bounds summarize the non-asymptotic behavior of the infinite horizon Monte Carlo FK estimator at a fixed point $x$. For the learning-theoretic analysis we only need that, for each fixed $x$, the FK label produced by Algorithm 1 behaves like a noisy point evaluation of $u^\star(x)$ with sub-exponential noise whose parameters do not depend on the number of FK labels or on the Monte Carlo budget. This is precisely the content of Proposition 6.1 in the main text, whose proof we now provide.

**Proof of Proposition 6.1.** Fix $x \in \Omega$ and consider the FK estimator produced at this spatial point by Algorithm 1, with time step $0 < \Delta t \leq \Delta t_0$, truncation horizon $T_{\max} \in (0, \infty]$ and Monte Carlo budget $N_{\mathrm{MC}}$. For each trajectory we simulate a payoff $Y^{\Delta t, T_{\max}}$ as in (C.7). The FK label at $x$ is the empirical average of $N_{\mathrm{MC}}$ independent copies of $Y^{\Delta t, T_{\max}}$, that is

$$\widehat{u}^{T_{\max}}(x) := \frac{1}{N_{\mathrm{MC}}} \sum_{i=1}^{N_{\mathrm{MC}}} Y^{\Delta t, T_{\max}, (i)}.$$

*Step 1: sub-exponential Monte Carlo fluctuations.*

We distinguish the cases $T_{\max} < \infty$ and $T_{\max} = \infty$.

If $T_{\max} < \infty$, then at most $N_T = \lfloor T_{\max}/\Delta t \rfloor$ time steps are accumulated before truncation. Boundedness of $f$ and $g$ and nonnegativity of $V$ imply (as in Lemma C.5) the deterministic bound

$$|Y^{\Delta t, T_{\max}}| \leq C_f T_{\max} + C_g \text{ almost surely,}$$

uniformly in $x$ and $0 < \Delta t \leq \Delta t_0$. Hence, for each fixed $x$, the centered variable

$$Z^{\Delta t, T_{\max}} := Y^{\Delta t, T_{\max}} - u_{\Delta t}^{T_{\max}}(x)$$

is bounded and therefore sub-exponential, with parameters that depend only on $C_f$, $C_g$ and $T_{\max}$ and do not depend on $N_{\mathrm{MC}}$ or on the number of FK points. Standard properties of sub-exponential variables (see, e.g., Vershynin (2018, Section 2.7)) imply that empirical averages of i.i.d. sub-exponential variables are again sub-exponential, and the sub-exponential parameters can be chosen independently of the sample size. Thus, conditional on $X_0 = x$,

$$\widehat{u}^{T_{\max}}(x) - u_{\Delta t}^{T_{\max}}(x) = \frac{1}{N_{\mathrm{MC}}} \sum_{i=1}^{N_{\mathrm{MC}}} Z^{\Delta t, T_{\max}, (i)}$$

is mean-zero and sub-exponential with parameters depending only on the problem data and on $T_{\max}$, and not on $N_{\mathrm{MC}}$.

If $T_{\max} = \infty$, then $Y^{\Delta t, \infty} = Y^{\Delta t}$ and $u_{\Delta t}^{\infty}(x) = u_{\Delta t}(x)$. Proposition C.7 yields constants $\lambda_Y > 0$, $C_Y < \infty$ and $\Delta t_0 > 0$ such that

$$\sup_{y \in \overline{\Omega}, \, 0 < \Delta t \leq \Delta t_0} \mathbb{E}^y \big[ e^{\lambda_Y |Y^{\Delta t}|} \big] \leq C_Y.$$

In particular, for each fixed $x$ and $0 < \Delta t \leq \Delta t_0$, the centered variable

$$Z^{\Delta t} := Y^{\Delta t} - u_{\Delta t}(x)$$

is sub-exponential with parameters depending only on $C_Y$ and $\lambda_Y$ and independent of $\Delta t$, $N_{\mathrm{MC}}$ and the number of FK points, as per Lemma C.10. As in the finite-horizon case, empirical averages of i.i.d. copies of $Z^{\Delta t}$ are again sub-exponential, with parameters that can be chosen independent of $N_{\mathrm{MC}}$. Thus $\widehat{u}^{\infty}(x) - u_{\Delta t}(x)$ is mean-zero and sub-exponential with parameters depending only on the model data.

In both cases we may therefore write

$$\widehat{u}^{T_{\max}}(x) = u_{\Delta t}^{T_{\max}}(x) + \zeta(x),$$

where, conditional on $X_0 = x$, the fluctuation $\zeta(x)$ is mean-zero and sub-exponential with parameters that do not depend on $N_{\mathrm{MC}}$ or on the number of FK points (and are uniform over $0 < \Delta t \leq \Delta t_0$).

*Step 2: discretization and truncation bias.*

We now decompose the bias of the (possibly truncated) FK value. Adding and subtracting $u_{\Delta t}(x)$ and $u^{\star}(x)$ yields

$$\widehat{u}^{T_{\max}}(x) = u^{\star}(x) + \big( u_{\Delta t}(x) - u^{\star}(x) \big) + \big( u_{\Delta t}^{T_{\max}}(x) - u_{\Delta t}(x) \big) + \zeta(x).$$

By Theorem C.13 the discretization bias satisfies

$$|u_{\Delta t}(x) - u^{\star}(x)| \leq C_{\mathrm{bias}} \sqrt{\Delta t}, \text{ for all } 0 < \Delta t \leq \Delta t_0,$$

while Lemma C.14 gives

$$|u_{\Delta t}^{T_{\max}}(x) - u_{\Delta t}(x)| \le C_T e^{-\tilde{\lambda} T_{\max}}, \text{ for all } T_{\max} > 0,$$

with the convention $e^{-\tilde{\lambda} T_{\max}} = 0$ when $T_{\max} = \infty$. Defining

$$b(x) := \left(u_{\Delta t}(x) - u^\star(x)\right) + \left(u_{\Delta t}^{T_{\max}}(x) - u_{\Delta t}(x)\right),$$

we obtain the decomposition

$$\hat{u}^{T_{\max}}(x) = u^\star(x) + b(x) + \zeta(x),$$

with

$$|b(x)| \le C_{\text{bias}} \sqrt{\Delta t} + C_T e^{-\tilde{\lambda} T_{\max}},$$

and with $\zeta(x)$ mean-zero and sub-exponential with parameters independent of $N_{\text{MC}}$ and of the number of FK points, uniformly over $0 < \Delta t \le \Delta t_0$.

*Step 3: random FK locations.* In the FK-PINN construction we draw i.i.d. supervision locations $X_i^{\text{FK}} \in \Omega$, and at each such point we compute an independent estimator $\hat{u}^{T_{\max}}(X_i^{\text{FK}})$ using a fresh set of $N_{\text{MC}}$ trajectories. Setting

$$Y_i^{\text{FK}} := \hat{u}^{T_{\max}}(X_i^{\text{FK}}), \quad b(X_i^{\text{FK}}) := b\left(X_i^{\text{FK}}\right), \quad \zeta_i := \zeta\left(X_i^{\text{FK}}\right),$$

the pairs $\{(X_i^{\text{FK}}, Y_i^{\text{FK}})\}_{i=1}^{N_{\text{FK}}}$ are i.i.d., and, conditional on $X_i^{\text{FK}} = x$, each $Y_i^{\text{FK}}$ admits the decomposition

$$Y_i^{\text{FK}} = u^\star(x) + b(x) + \zeta_i,$$

with $|b(x)| \le C_{\text{bias}} \sqrt{\Delta t} + C_T e^{-\tilde{\lambda} T_{\max}}$ and with $\zeta_i$ mean-zero and sub-exponential with parameters that do not depend on $N_{\text{FK}}$ or $N_{\text{MC}}$. This is exactly the statement in Proposition 6.1, thus we are done.

# D. Error Analysis for FK-PINNs

The goal of this section is to prove Theorem 6.2. To this end, we will develop a comprehensive error analysis, based on a decomposition of the $L^2(\Omega)$ error in three terms, which we will control separately through learning-theoretic arguments.

## D.1. Preliminaries

We begin by recalling the setting in which we will be working together with notations and technical assumptions that we will be working under for the remainder of this section.

### D.1.1. NOTATIONS

We work on a bounded Lipschitz domain $\Omega \subset [0,1]^d$ with boundary $\partial\Omega$ and outward unit normal $\nu$. Integration with respect to Lebesgue measure on $\Omega$ is written as $\int_\Omega \cdot \, dx$, while $\int_{\partial\Omega} \cdot \, ds$ denotes integration with respect to surface measure on $\partial\Omega$.

For $1 \le p \le \infty$ we write $L^p(\Omega)$ and $L^p(\partial\Omega)$ for the usual Lebesgue spaces, with norms

$$\|u\|_{L^p(\Omega)} := \begin{cases} \left(\int_\Omega |u(x)|^p \, dx\right)^{1/p}, & 1 \le p < \infty, \\ \operatorname{ess\,sup}_{x \in \Omega} |u(x)|, & p = \infty, \end{cases}$$

and similarly on $\partial\Omega$. We write $C(\overline{\Omega})$ for the space of continuous functions on the closure of $\Omega$ with norm $\|u\|_{C(\overline{\Omega})} := \sup_{x \in \overline{\Omega}} |u(x)|$.

For $s \in \mathbb{N}$ and $1 \le p \le \infty$, the Sobolev space $W^{s,p}(\Omega)$ consists of functions in $L^p(\Omega)$ whose weak derivatives up to order $s$ lie in $L^p(\Omega)$, endowed with norm

$$\|u\|_{W^{s,p}(\Omega)} := \left( \sum_{|\alpha| \le s} \|D^\alpha u\|_{L^p(\Omega)}^p \right)^{1/p} \quad \text{for } p < \infty,$$

and with the obvious modification for $p = \infty$. We use $H^s(\Omega) := W^{s,2}(\Omega)$. Fractional Sobolev spaces $H^s(\Omega)$ for $s \notin \mathbb{N}$ are defined in the standard way, for instance via real interpolation or via the Gagliardo seminorm, see Evans (2022). We denote the associated norms by $\|\cdot\|_{H^s(\Omega)}$.

Throughout, $\langle \cdot \rangle \cdot$ denotes the $L^2(\Omega)$ inner product, and $\|\cdot\|_{L^2(\Omega)}$ is abbreviated to $\|\cdot\|_2$ when the domain is clear from the context. For random variables we write $\mathbb{E}[\cdot]$ for expectation and $\mathbb{P}(\cdot)$ for probabilities.

### D.1.2. SETTING AND ASSUMPTIONS

We recall the PDE from Section 2.1. Throughout this appendix we work in the elliptic setting for clarity. All statements are formulated and proved for the second order elliptic problem in (2.1)–(2.2). The same learning theoretic arguments extend to parabolic problems that admit a Feynman–Kac representation, once the time variable has been discretized and the resulting time discretization error is controlled separately. Since this would require additional notation and is orthogonal to the main message of the paper, we do not pursue the parabolic case in detail here.

**Assumption D.1** (PDE well posedness and FK representation). The operator $\mathcal{L}$ in (2.1) is uniformly elliptic with bounded measurable coefficients and satisfies the structural conditions stated in Section 2.1. The data $f$ and $g$ are such that (2.1)–(2.2) admit a unique strong solution
$$u^\star \in H^2(\Omega) \cap C(\overline{\Omega}),$$
with $u^\star|_{\partial\Omega} = g$. Moreover, there exists a diffusion process $(X_t)_{t \ge 0}$ and an exit time $\tau$ such that $u^\star$ admits a Feynman–Kac representation of the form
$$u^\star(x) = \mathbb{E}_x\left[ \int_0^\tau r(X_t) \, dt + h(X_\tau) \right], \quad x \in \Omega,$$
for suitable running and terminal functionals $r$ and $h$, as described in Section 3 and Appendix A.

We follow the sampling scheme described in Algorithm 1 and Algorithm 2. Let
$$X^{\text{int}} \sim U(\Omega), \qquad X^{\partial\Omega} \sim U(\partial\Omega),$$

be interior and boundary points drawn independently from the uniform distributions on $\Omega$ and $\partial\Omega$. The corresponding population PDE and boundary risks are

$$\mathcal{R}_{\mathrm{PDE}}(u) := |\Omega|\,\mathbb{E}\big[\big|\mathcal{L}u(X^{\mathrm{int}}) - f(X^{\mathrm{int}})\big|^2\big], \tag{D.1}$$

$$\mathcal{R}_{\partial\Omega}(u) := |\partial\Omega|\,\mathbb{E}\big[\big|u(X^{\partial\Omega}) - g(X^{\partial\Omega})\big|^2\big]. \tag{D.2}$$

In the FK term we work in a data scarce regime. We sample FK points from the same underlying geometry as the interior points, but we keep their number small. Concretely, we assume

$$X^{\mathrm{FK}} \sim U(\Omega),$$

independently of $X^{\mathrm{int}}$ and $X^{\partial\Omega}$, while the sample size $N_{\mathrm{FK}}$ is allowed to remain bounded or grow much more slowly than $N_{\mathrm{int}}$ and $N_{\partial\Omega}$. At each such point we generate a Monte Carlo FK estimate $\widehat{u}^{\mathrm{MC}}(X^{\mathrm{FK}})$ as in Algorithm 1. Proposition 6.1 shows that these labels admit the decomposition

$$\widehat{u}^{\mathrm{MC}}(X^{\mathrm{FK}}) = u^\star(X^{\mathrm{FK}}) + b(X^{\mathrm{FK}}) + \zeta,$$

where the deterministic bias $b : \Omega \to \mathbb{R}$ satisfies

$$|b(x)| \le C_{\mathrm{bias}}\sqrt{\Delta t} + C_T e^{-\kappa T_{\max}} \quad \text{for all } x \in \Omega,$$

for constants $C_{\mathrm{bias}}, C_T, \kappa > 0$ independent of $N_{\mathrm{FK}}$ and of the number of FK trajectories, and where, conditional on $X^{\mathrm{FK}}$, the fluctuation $\zeta$ is mean-zero and sub-exponential with parameters that do not depend on $N_{\mathrm{FK}}$ or on the number of trajectories. The corresponding population FK risk is

$$\mathcal{R}_{\mathrm{FK}}(u) := \mathbb{E}\big[\big|u(X^{\mathrm{FK}}) - u^\star(X^{\mathrm{FK}})\big|^2\big]. \tag{D.3}$$

For fixed nonnegative weights $\lambda_{\mathrm{PDE}}$, $\lambda_{\partial\Omega}$ and $\lambda_{\mathrm{FK}}$ we define the population FK-PINN risk

$$\mathcal{R}(u) := \lambda_{\mathrm{PDE}}\mathcal{R}_{\mathrm{PDE}}(u) + \lambda_{\partial\Omega}\mathcal{R}_{\partial\Omega}(u) + \lambda_{\mathrm{FK}}\mathcal{R}_{\mathrm{FK}}(u). \tag{D.4}$$

In what follows we assume that the PDE and boundary weights $\lambda_{\mathrm{PDE}}, \lambda_{\partial\Omega}$, and that the FK weight is non-negative. For convenience, we will use the abbreviation

$$\lambda_{\min} := \min\{\lambda_{\mathrm{PDE}}, \lambda_{\partial\Omega}\}.$$

Note that rescaling the three weights only changes the constants in the bounds below, so this choice entails no loss of generality. The empirical FK-PINN risk is obtained by Monte Carlo approximation of the expectations in (D.1)–(D.3). Given i.i.d. samples

$$\{x_i^{\mathrm{int}}\}_{i=1}^{N_{\mathrm{int}}} \subset \Omega, \quad \{x_j^{\partial\Omega}\}_{j=1}^{N_{\partial\Omega}} \subset \partial\Omega, \quad \{x_k^{\mathrm{FK}}\}_{k=1}^{N_{\mathrm{FK}}} \subset \Omega,$$

and the corresponding FK labels $\{\widehat{u}^{\mathrm{MC}}(x_k^{\mathrm{FK}})\}_{k=1}^{N_{\mathrm{FK}}}$, we set

$$\widehat{\mathcal{R}}_{\mathrm{PDE}}(u) := |\Omega|\,\frac{1}{N_{\mathrm{int}}}\sum_{i=1}^{N_{\mathrm{int}}}\big|\mathcal{L}u(x_i^{\mathrm{int}}) - f(x_i^{\mathrm{int}})\big|^2, \tag{D.5}$$

$$\widehat{\mathcal{R}}_{\partial\Omega}(u) := |\partial\Omega|\,\frac{1}{N_{\partial\Omega}}\sum_{j=1}^{N_{\partial\Omega}}\big|u(x_j^{\partial\Omega}) - g(x_j^{\partial\Omega})\big|^2, \tag{D.6}$$

$$\widehat{\mathcal{R}}_{\mathrm{FK}}(u) := \frac{1}{N_{\mathrm{FK}}}\sum_{k=1}^{N_{\mathrm{FK}}}\big|u(x_k^{\mathrm{FK}}) - \widehat{u}^{\mathrm{MC}}(x_k^{\mathrm{FK}})\big|^2, \tag{D.7}$$

and

$$\widehat{\mathcal{R}}(u) := \lambda_{\mathrm{PDE}}\widehat{\mathcal{R}}_{\mathrm{PDE}}(u) + \lambda_{\partial\Omega}\widehat{\mathcal{R}}_{\partial\Omega}(u) + \lambda_{\mathrm{FK}}\widehat{\mathcal{R}}_{\mathrm{FK}}(u). \tag{D.8}$$

Up to the adaptive task weights used in Equation (4.3), the empirical loss $\widehat{\mathcal{R}}(u_\theta)$ coincides with $\mathcal{R}_{\mathrm{FK\text{-}PINN}}(\theta, \varphi)$.

We will make use of the following classical PDE estimate.

**Lemma D.1** (PDE stability). *Under Assumption D.1 there exists a constant $C_{\text{stab}} > 0$ such that for every $u \in H^2(\Omega) \cap C(\overline{\Omega})$ with $u|_{\partial\Omega} \in L^2(\partial\Omega)$,*

$$\|u - u^\star\|^2_{L^2(\Omega)} \leq C_{\text{stab}}\Big(\mathcal{R}_{\text{PDE}}(u) + \mathcal{R}_{\partial\Omega}(u)\Big). \tag{D.9}$$

Lemma D.1 is a direct consequence of the a priori boundary estimates of Agmon et al. (1959), combined with the embedding $L^2(\Omega) \hookrightarrow H^{-s}(\Omega)$ for suitable $s > 0$. One may also refer to Evans (2022, Chapter 6) and Jiao et al. (2022); Lei et al. (2025) for formulations tailored to PINNs.

We will work with a hypothesis class $\mathcal{U}$ of candidate solutions, specified in the next subsection. For any $u \in \mathcal{U}$ we define

$$\mathcal{E}_{\text{app}} := \inf_{v \in \mathcal{U}} \|v - u^\star\|_{L^2(\Omega)}, \qquad \mathcal{E}_{\text{stat}} := \sup_{v \in \mathcal{U}} \big|\mathcal{R}(v) - \widehat{\mathcal{R}}(v)\big|. \tag{D.10}$$

The following decomposition is standard.

**Lemma D.2** (Risk decomposition). *Let $\widehat{u} \in \arg\min_{u \in \mathcal{U}} \widehat{\mathcal{R}}(u)$ be any empirical minimizer. Then*

$$\mathcal{R}(\widehat{u}) - \mathcal{R}(u^\star) \leq 2\,\mathcal{E}_{\text{stat}} + \inf_{u \in \mathcal{U}}\big[\mathcal{R}(u) - \mathcal{R}(u^\star)\big]. \tag{D.11}$$

*Proof.* Fix any $u \in \mathcal{U}$. Then

$$\mathcal{R}(\widehat{u}) - \mathcal{R}(u^\star) = \big(\mathcal{R}(\widehat{u}) - \widehat{\mathcal{R}}(\widehat{u})\big) + \big(\widehat{\mathcal{R}}(\widehat{u}) - \widehat{\mathcal{R}}(u)\big)$$
$$+ \big(\widehat{\mathcal{R}}(u) - \mathcal{R}(u)\big) + \big(\mathcal{R}(u) - \mathcal{R}(u^\star)\big).$$

By optimality of $\widehat{u}$ for $\widehat{\mathcal{R}}$ we have $\widehat{\mathcal{R}}(\widehat{u}) \leq \widehat{\mathcal{R}}(u)$, so the middle term is nonpositive. Hence

$$\mathcal{R}(\widehat{u}) - \mathcal{R}(u^\star) \leq \big|\mathcal{R}(\widehat{u}) - \widehat{\mathcal{R}}(\widehat{u})\big| + \big|\widehat{\mathcal{R}}(u) - \mathcal{R}(u)\big| + \big(\mathcal{R}(u) - \mathcal{R}(u^\star)\big)$$
$$\leq 2\,\mathcal{E}_{\text{stat}} + \big(\mathcal{R}(u) - \mathcal{R}(u^\star)\big),$$

where we used the definition of $\mathcal{E}_{\text{stat}}$ in (D.10). Since this bound holds for every $u \in \mathcal{U}$, taking the infimum over $u$ yields

$$\mathcal{R}(\widehat{u}) - \mathcal{R}(u^\star) \leq 2\,\mathcal{E}_{\text{stat}} + \inf_{u \in \mathcal{U}}\big[\mathcal{R}(u) - \mathcal{R}(u^\star)\big]. \qquad \square$$

Later we also introduce an optimization error $\mathcal{E}_{\text{opt}}$ that measures the gap between the empirical minimizer and the iterate produced by gradient descent. These three terms will form the building blocks of the final non-asymptotic bound.

### D.1.3. NEURAL NETWORK ARCHITECTURE

We consider fully connected feedforward neural networks with $\tanh$ activations in the hidden layers and a linear output. For an integer depth $L \geq 1$ and a fixed width vector $(m_\ell)_{\ell=0}^L$ with $m_0 = d$ and $m_L = 1$, a parameter vector

$$\boldsymbol{\theta} := \big\{W_\ell, b_\ell\big\}_{\ell=1}^L, \quad W_\ell \in \mathbb{R}^{m_\ell \times m_{\ell-1}}, \ b_\ell \in \mathbb{R}^{m_\ell},$$

defines a function $u_{\boldsymbol{\theta}} : \Omega \to \mathbb{R}$ by

$$z_0(x) := x \in \mathbb{R}^d, \tag{D.12}$$
$$z_\ell(x) := \tanh\big(W_\ell z_{\ell-1}(x) + b_\ell\big), \quad \ell = 1, \ldots, L-1, \tag{D.13}$$
$$u_{\boldsymbol{\theta}}(x) := W_L z_{L-1}(x) + b_L \in \mathbb{R}. \tag{D.14}$$

We write $P := P(L, (m_\ell)_\ell)$ for the total number of free parameters in $\boldsymbol{\theta}$ and call $L$ the depth and $m := \max_\ell m_\ell$ the width of the network.

We will later use the following upper bound on the number of free parameters: by definition,

$$P = \sum_{\ell=1}^L \big(m_\ell m_{\ell-1} + m_\ell\big), \ m_0 = d, \ m_L = 1 \ \text{and} \ m_\ell \leq m \ \text{for all} \ 0 \leq \ell \leq L.$$

This implies that $P \leq C_{\text{arch}}(d) \, L \, m^2$ for a constant $C_{\text{arch}}(d) > 0$ depending only on the input dimension $d$. In all subsequent bounds we may therefore replace $P$ by $C_{\text{arch}}(d) \, L \, m^2$ if we wish to make the dependence on the architectural hyperparameters explicit.

For the learning-theoretic analysis we restrict attention to networks whose outputs and low order derivatives are uniformly bounded on $\overline{\Omega}$. For fixed $L$, $(m_\ell)_\ell$ and $B > 0$ we first define the parameter set

$$\Theta_{L,m} := \left\{ \boldsymbol{\theta} = (W_\ell, b_\ell)_{\ell=1}^L : W_\ell \in \mathbb{R}^{m_\ell \times m_{\ell-1}}, \ b_\ell \in \mathbb{R}^{m_\ell} \right\}.$$

The associated hypothesis class is

$$\mathcal{U}_{L,m,B} := \left\{ u_{\boldsymbol{\theta}} : \boldsymbol{\theta} \in \Theta_{L,m}, \ \|u_{\boldsymbol{\theta}}\|_{C^2(\overline{\Omega})} \leq B \right\}, \tag{D.15}$$

where $\|\cdot\|_{C^2(\overline{\Omega})}$ denotes the usual $C^2$ norm that controls the function and its first and second order partial derivatives on $\overline{\Omega}$. The class $\mathcal{U}_{L,m,B}$ is nonempty for every $B$ large enough, since constant functions belong to it.

To lighten notation we often write $\mathcal{U}$ for $\mathcal{U}_{L,m,B}$ and denote by $P$ the maximal number of parameters of networks in $\mathcal{U}$. All constants in the sequel may depend on $(L, m, B, d)$ but never on the sample sizes $(N_{\text{int}}, N_{\partial\Omega}, N_{\text{FK}})$.

### D.2. Approximation Error Bound

We first bound the approximation error in (D.10) in terms of the size of the network. The desired bounds will directly follow from the approximation error bounds of $\tanh$ networks due to De Ryck et al. (2021).

**Theorem D.3** (Approximation by two hidden layer $\tanh$ networks, simplified)**.** *Let $d \in \mathbb{N}$, let $s \in \mathbb{N}$ with $s \geq 3$ and let $f \in W^{s,\infty}([0,1]^d)$. There exist constants $C_{\text{approx}}, C_{\text{width}} > 0$, depending only on $(d, s, \|f\|_{W^{s,\infty}})$, such that for every integer $N$ large enough there is a $\tanh$ network $\widehat{f}_N$ with two hidden layers whose total number of parameters satisfies*

$$P_N \leq C_{\text{width}} N^d$$

*and such that, for all multi indices $\alpha$ with $|\alpha| \leq 2$,*

$$\|D^\alpha f - D^\alpha \widehat{f}_N\|_{L^\infty([0,1]^d)} \leq C_{\text{approx}} N^{-(s-|\alpha|)}. \tag{D.16}$$

*Remark* D.4. Theorem D.3 is a mild simplification of De Ryck et al. (2021, Theorem 5.1). The original statement specifies exact widths for the two hidden layers and tracks additional logarithmic factors in $N$, while we only retain the dominant polynomial dependence on $N$, which is all that enters our rate calculations.

We now apply this result to the hypothesis class $\mathcal{U}$ defined in (D.15) and to the true solution $u^\star$.

**Corollary D.5** (Approximation error for $\mathcal{U}_{L,m,B}$)**.** *Suppose that Assumption D.1 holds and that $u^\star \in W^{s,\infty}(\Omega)$ for some integer $s \geq 3$. Fix any depth $L \geq 3$. Then there exist constants $C_{\text{app},1}, C_{\text{app},2} > 0$ and $m_0 \in \mathbb{N}$ such that for every width $m \geq m_0$ the hypothesis class $\mathcal{U}_{L,m,B}$ contains a network $u^{\text{approx}}$ with*

$$\|u^\star - u^{\text{approx}}\|_{C^2(\overline{\Omega})} \leq C_{\text{app},1} \, P(L,m)^{-\gamma}, \qquad \gamma := \frac{s-2}{d}, \tag{D.17}$$

*where $P(L,m)$ denotes the total number of free parameters of networks in $\mathcal{U}_{L,m,B}$. In particular,*

$$\mathcal{E}_{\text{app}} = \inf_{u \in \mathcal{U}_{L,m,B}} \|u - u^\star\|_{L^2(\Omega)} \leq C_{\text{app},2} \, P(L,m)^{-\gamma}. \tag{D.18}$$

*Proof.* By the Sobolev extension theorem whose conditions are satisfied here (Evans, 2022), we can extend $u^\star$ to a function in $W^{s,\infty}([0,1]^d)$ with comparable norm. Now we apply Theorem D.3 with this extension : for each integer $N$ large enough, we obtain a two-hidden-layer $\tanh$ network $\widehat{u}_N$ with depth $L = 3$, widths of order $N^d$ in each hidden layer, total number of parameters $P_N \leq C_{\text{width}} N^d$, and

$$\|D^\alpha u^\star - D^\alpha \widehat{u}_N\|_{L^\infty([0,1]^d)} \leq C_{\text{approx}} N^{-(s-|\alpha|)}, \qquad |\alpha| \leq 2.$$

In particular,

$$\|u^\star - \widehat{u}_N\|_{C^2(\overline{\Omega})} \leq C N^{-(s-2)}$$

for a constant $C$ depending on $(d, s, \|u^\star\|_{W^{s,\infty}})$.

Any such two-hidden-layer network can be realized as a special case of our architecture with depth $L \geq 3$ and width $m$ as soon as $m$ is larger than the maximal width of $\widehat{u}_N$, since the extra layers and neurons can be made inactive by setting their weights to zero or to identities. Thus, for every $L \geq 3$ there exists $m_0$ such that for all $m \geq m_0$ the network $\widehat{u}_N$ belongs to $\mathcal{U}_{L,m,B}$ provided $B$ is chosen large enough.

Let $N_{\min}$ be the (fixed) threshold required in Theorem D.3 and choose

$$N := \max\Big\{N_{\min}, \ \big\lfloor (P(L,m)/C_{\text{width}})^{1/d} \big\rfloor \Big\}.$$

Then $P_N \leq C_{\text{width}} N^d \leq P(L,m)$ and therefore

$$\|u^\star - \widehat{u}_N\|_{C^2(\overline{\Omega})} \leq C N^{-(s-2)} \leq C' P(L,m)^{-\gamma}, \qquad \gamma = \frac{s-2}{d},$$

for a constant $C' > 0$. Taking $u^{\text{approx}} = \widehat{u}_N$ yields (D.17) with $C_{\text{app},1} = CC'$. Since $\Omega$ is bounded, $\|v\|_{L^2(\Omega)} \leq C_\Omega \|v\|_{C(\overline{\Omega})}$ for every continuous $v$, and (D.18) follows from (D.17) by absorbing $C_\Omega$ into $C_{\text{app},2}$. $\qquad\square$

Corollary D.5 shows that $\mathcal{E}_{\text{app}}$ decays at least polynomially in the number of parameters whenever $u^\star$ has Sobolev regularity of order exceeding two. Stronger, essentially exponential, rates can be obtained under analyticity assumptions on $u^\star$ using De Ryck et al. (2021, Corollary 5.5), but we do not pursue this here.

**Lemma D.6** (Control of the population risk by $C^2$ approximation). *Under Assumption D.1 and with the weights $(\lambda_{\text{PDE}}, \lambda_{\partial\Omega}, \lambda_{\text{FK}})$ as in (D.4), there exists a constant $C_{\text{risk}} > 0$, depending only on the PDE coefficients, the domain, and the weights, such that for every $u \in C^2(\overline{\Omega})$ with*

$$\|u - u^\star\|_{C^2(\overline{\Omega})} \leq \varepsilon,$$

*one has*

$$\mathcal{R}(u) \leq C_{\text{risk}} \varepsilon^2. \tag{D.19}$$

*In particular, for the approximating network $u^{\text{approx}}$ from Corollary D.5 we have*

$$\mathcal{R}(u^{\text{approx}}) \leq \widetilde{C}_{\text{risk}} P^{-2\gamma}, \tag{D.20}$$

*for a constant $\widetilde{C}_{\text{risk}} > 0$ independent of $P$.*

*Proof.* Since $\mathcal{L}$ is a second order operator with bounded measurable coefficients, there exists a constant $C_{\mathcal{L}} > 0$ such that

$$\sup_{x\in\overline{\Omega}} \big|\mathcal{L}(u - u^\star)(x)\big| \leq C_{\mathcal{L}} \|u - u^\star\|_{C^2(\overline{\Omega})}.$$

Therefore

$$\mathcal{R}_{\text{PDE}}(u) = \int_\Omega \big(\mathcal{L}u(x) - f(x)\big)^2 dx = \int_\Omega \big(\mathcal{L}(u - u^\star)(x)\big)^2 dx \leq |\Omega| \, C_{\mathcal{L}}^2 \, \varepsilon^2.$$

On the boundary we have

$$\sup_{x\in\partial\Omega} \big|u(x) - g(x)\big| = \sup_{x\in\partial\Omega} \big|u(x) - u^\star(x)\big| \leq \|u - u^\star\|_{C(\overline{\Omega})} \leq \varepsilon,$$

which gives

$$\mathcal{R}_{\partial\Omega}(u) = \int_{\partial\Omega} \big(u(x) - g(x)\big)^2 ds \leq |\partial\Omega| \, \varepsilon^2.$$

Finally, for the FK term we use that $u^\star$ is itself the target in (D.3), hence

$$\mathcal{R}_{\text{FK}}(u) = \mathbb{E}\big[\big|u(X^{\text{FK}}) - u^\star(X^{\text{FK}})\big|^2\big] \leq \|u - u^\star\|_{C(\overline{\Omega})}^2 \leq \varepsilon^2.$$

Collecting these three bounds and inserting the weights gives (D.19) with a constant $C_{\text{risk}}$ that depends only on the coefficients of $\mathcal{L}$, the domain, and the weights. The bound (D.20) then follows from (D.17). $\qquad\square$

## D.3. Optimization Error Bound

We next control the optimization error incurred by terminating gradient descent after a finite number of iterations. Throughout this subsection we fix the sample set and view the empirical FK-PINN loss

$$\ell(\boldsymbol{\theta}) := \widehat{\mathcal{R}}(u_{\boldsymbol{\theta}})$$

as a deterministic function of the parameters.

We rely on the Polyak–Łojasiewicz framework of Karimi et al. (2016); Liu et al. (2022). A function $\ell : \mathbb{R}^P \to \mathbb{R}$ satisfies a local PL$^*$ condition on a set $S \subset \mathbb{R}^P$ with constant $\mu > 0$ if

$$\ell(\boldsymbol{\theta}) - \ell(\boldsymbol{\theta}^\star) \leq \frac{1}{2\mu} \|\nabla \ell(\boldsymbol{\theta})\|_2^2, \qquad \forall \boldsymbol{\theta} \in S, \tag{D.21}$$

for some global minimizer $\boldsymbol{\theta}^\star \in S$ of $\ell$. If, in addition, $\ell$ is $\beta$-smooth on $S$ (that is, $\|\nabla \ell(\boldsymbol{\theta}) - \nabla \ell(\boldsymbol{\theta}')\|_2 \leq \beta \|\boldsymbol{\theta} - \boldsymbol{\theta}'\|_2$ for all $\boldsymbol{\theta}, \boldsymbol{\theta}' \in S$), the ratio

$$\kappa_\ell(S) := \frac{\beta}{\mu}$$

plays the role of a condition number.

The following result is a special case of Liu et al. (2022, Theorem 6).

**Theorem D.7** (Gradient descent under a local PL$^*$ condition). *Let $\ell$ be differentiable and $\beta$-smooth on a convex set $S \subset \mathbb{R}^P$, and assume that $\ell$ satisfies the PL$^*$ condition (D.21) on $S$ with constant $\mu > 0$. Let $\boldsymbol{\theta}_0 \in S$ and consider gradient descent*

$$\boldsymbol{\theta}_{t+1} = \boldsymbol{\theta}_t - \eta \nabla \ell(\boldsymbol{\theta}_t),$$

*with a fixed step size $\eta \in (0, 2/\beta]$. Suppose that all iterates remain in $S$. Then for all $t \geq 0$,*

$$\ell(\boldsymbol{\theta}_t) - \ell(\boldsymbol{\theta}^\star) \leq \left(1 - \eta\mu\right)^t \left(\ell(\boldsymbol{\theta}_0) - \ell(\boldsymbol{\theta}^\star)\right). \tag{D.22}$$

In Theorem 5.4 we proved that the FK-augmented loss enjoys a PL$^*$ inequality with a condition number that stays uniformly bounded as the number of collocation points increases. We translate this consequence for the empirical loss.

**Corollary D.8** (Uniform PL$^*$ geometry for FK-PINNs). *Under the assumptions of Theorem 5.4 there exist a convex set $S \subset \mathbb{R}^P$, constants $\mu_{\mathrm{FK}} > 0$, $\beta_{\mathrm{FK}} < \infty$ and an initialization $\boldsymbol{\theta}_0 \in S$ such that the empirical loss $\ell(\boldsymbol{\theta}) = \widehat{\mathcal{R}}(u_{\boldsymbol{\theta}})$ satisfies the PL$^*$ condition (D.21) on $S$ with constant $\mu_{\mathrm{FK}}$, is $\beta_{\mathrm{FK}}$-smooth on $S$, and the associated condition number*

$$\kappa_{\mathrm{FK}} := \frac{\beta_{\mathrm{FK}}}{\mu_{\mathrm{FK}}}$$

*is bounded by a constant that does not depend on $(N_{\mathrm{int}}, N_{\partial\Omega}, N_{\mathrm{FK}})$.*

Under Theorem D.8, gradient descent with a suitable step size converges linearly to an empirical minimizer. We quantify the corresponding optimization error in terms of the number of iterations.

**Proposition D.9** (Optimization error bound). *Let $\{\boldsymbol{\theta}_t\}_{t \geq 0}$ be the gradient descent iterates for $\ell(\boldsymbol{\theta}) = \widehat{\mathcal{R}}(u_{\boldsymbol{\theta}})$ with step size $\eta \in (0, 2/\beta_{\mathrm{FK}}]$, initialized at $\boldsymbol{\theta}_0$ as in Theorem D.8. Let*

$$\widehat{u} \in \arg\min_{u \in \mathcal{U}} \widehat{\mathcal{R}}(u)$$

*be an empirical minimizer and write $\boldsymbol{\theta}^\star$ for a parameter vector with $u_{\boldsymbol{\theta}^\star} = \widehat{u}$. Define the optimization error*

$$\mathcal{E}_{\mathrm{opt}}(t) := \widehat{\mathcal{R}}(u_{\boldsymbol{\theta}_t}) - \widehat{\mathcal{R}}(\widehat{u}). \tag{D.23}$$

*Then, under Corollary D.8, there exist constants $C_{\mathrm{opt}}, c_{\mathrm{opt}} > 0$ independent of the sample sizes and of $t$ such that*

$$\mathcal{E}_{\mathrm{opt}}(t) \leq C_{\mathrm{opt}} \exp(-c_{\mathrm{opt}} t). \tag{D.24}$$

*In particular, for every $t \geq 0$ the network $u_{\boldsymbol{\theta}_t}$ is an $\mathcal{E}_{\mathrm{opt}}(t)$-approximate empirical minimizer of $\widehat{\mathcal{R}}$ in the sense that*

$$\widehat{\mathcal{R}}(u_{\boldsymbol{\theta}_t}) \leq \inf_{u \in \mathcal{U}} \widehat{\mathcal{R}}(u) + \mathcal{E}_{\mathrm{opt}}(t).$$

*Proof.* Inequality (D.24) follows directly from Theorem D.7 with $\mu = \mu_{\mathrm{FK}}$ and $\beta = \beta_{\mathrm{FK}}$, taking $\boldsymbol{\theta}^\star$ to be any parameter vector that realizes an empirical minimizer $\widehat{u}$ of $\widehat{\mathcal{R}}$. $\qquad\square$

## D.4. Statistical Error Bound

We now turn to the statistical error $\mathcal{E}_{\text{stat}}$ in (D.10), which captures the deviation between the population risk and its empirical approximation. Our goal is to obtain non-asymptotic high probability bounds on

$$\mathcal{E}_{\text{stat}} = \sup_{u \in \mathcal{U}} \left| \mathcal{R}(u) - \widehat{\mathcal{R}}(u) \right|$$

in terms of the number of collocation points and the complexity of the hypothesis class. The analysis follows the now standard Rademacher complexity approach in statistical learning theory, but it has two distinctive features.

First, we operate in a data scarce regime for the FK term: $N_{\text{FK}}$ is allowed to remain bounded while $N_{\text{int}}$ and $N_{\partial\Omega}$ grow. Second, we must control the complexity not only of the network outputs $u_{\boldsymbol{\theta}}$ but also of their first and second order derivatives, since the PDE residual involves $\partial^\alpha u_{\boldsymbol{\theta}}$ for $|\alpha| \le 2$. To the best of our knowledge, existing results of this kind only cover piecewise polynomial activations (see e.g. Jiao et al. (2022); Lu et al. (2022); Lei et al. (2025)), which does not include $\tanh$. We therefore derive new pseudo-dimension bounds for $\tanh$ networks and their derivatives, building on classical results of Karpinski & Macintyre (1997) and Anthony & Bartlett (2009). These bounds are of independent interest.

### D.4.1. PRELIMINARY RESULTS

We briefly recall the main tools from empirical process theory that we will use. Detailed expositions can be found, e.g., in Anthony & Bartlett (2009) and Wainwright (2019).

**Rademacher complexity.** Let $(Z_i)_{i=1}^n$ be i.i.d. random variables taking values in a space $\mathcal{Z}$ and let $\mathcal{F}$ be a class of measurable functions $f : \mathcal{Z} \to \mathbb{R}$. The empirical Rademacher complexity of $\mathcal{F}$ with respect to a fixed sample $Z_1, \ldots, Z_n$ is

$$\mathfrak{R}_n(\mathcal{F} \mid Z_{1:n}) := \mathbb{E}_\sigma \left[ \sup_{f \in \mathcal{F}} \frac{1}{n} \sum_{i=1}^n \sigma_i f(Z_i) \right],$$

where $(\sigma_i)_{i=1}^n$ are i.i.d. Rademacher variables ($\mathbb{P}(\sigma_i = 1) = \mathbb{P}(\sigma_i = -1) = 1/2$). The expected Rademacher complexity is $\mathfrak{R}_n(\mathcal{F}) := \mathbb{E}[\mathfrak{R}_n(\mathcal{F} \mid Z_{1:n})]$.

Rademacher complexity controls uniform deviations between empirical and population averages.

**Lemma D.10** (Symmetrization and contraction). *Let $\mathcal{F}$ be a class of functions $f : \mathcal{Z} \to \mathbb{R}$ with $|f(z)| \le B$ for all $f \in \mathcal{F}$ and $z \in \mathcal{Z}$. Then for any $n \in \mathbb{N}$ and any $\delta \in (0, 1)$, with probability at least $1 - \delta$,*

$$\sup_{f \in \mathcal{F}} \left| \mathbb{E}[f(Z)] - \frac{1}{n} \sum_{i=1}^n f(Z_i) \right| \le 2\,\mathfrak{R}_n(\mathcal{F}) + 3B \sqrt{\frac{\log(2/\delta)}{2n}}.$$

*Moreover, if $\phi : \mathbb{R} \to \mathbb{R}$ is $L$-Lipschitz and $\phi(0) = 0$, then*

$$\mathfrak{R}_n(\phi \circ \mathcal{F}) \le L\,\mathfrak{R}_n(\mathcal{F}),$$

*where $\phi \circ \mathcal{F} := \{\phi \circ f : f \in \mathcal{F}\}$.*

A proof of the deviation bound is given in (Wainwright, 2019, Chapter 3). The Lipschitz mapping bound follows from the contraction principle (Ledoux & Talagrand, 2013).

**Covering numbers and pseudo-dimension.** Let $A \subset \mathbb{R}^n$ and consider the metric

$$d_\infty(a, a') := \max_{1 \le i \le n} |a_i - a_i'|.$$

An $\varepsilon$-cover of $A$ with respect to $d_\infty$ is a finite subset $V \subset \mathbb{R}^n$ such that for every $a \in A$ there exists $v \in V$ with $d_\infty(a, v) \le \varepsilon$. The covering number $\mathcal{C}(\varepsilon, A, d_\infty)$ is the smallest cardinality of an $\varepsilon$-cover of $A$.

Given a function class $\mathcal{F}$ and a sample $Z_{1:n} = (Z_1, \ldots, Z_n)$, the evaluation set

$$\mathcal{F}|_{Z_{1:n}} := \left\{ (f(Z_1), \ldots, f(Z_n)) : f \in \mathcal{F} \right\} \subset \mathbb{R}^n$$

induces a covering number

$$\mathcal{C}_\infty(\varepsilon, \mathcal{F}, n) := \sup_{Z_{1:n}} \mathcal{C}\big(\varepsilon, \mathcal{F}|_{Z_{1:n}}, d_\infty\big).$$

The pseudo-dimension $\mathrm{Pdim}(\mathcal{F})$ is the standard extension of VC dimension to real-valued function classes (Anthony & Bartlett, 2009). The next two results relate Rademacher complexity, covering numbers and pseudo-dimension.

**Lemma D.11** (Dudley's entropy integral). *Let $\mathcal{F}$ be a class of functions $f : \mathcal{Z} \to \mathbb{R}$ with $|f| \le B$. Then for every $n \in \mathbb{N}$,*

$$\mathfrak{R}_n(\mathcal{F}) \le \inf_{0 < \delta < B} \left( 4\delta + \frac{12}{\sqrt{n}} \int_\delta^B \sqrt{\log\big(2\,\mathcal{C}_\infty(\varepsilon, \mathcal{F}, n)\big)}\, d\varepsilon \right).$$

**Proposition D.12** (Covering numbers from pseudo-dimension). *Let $\mathcal{F}$ be a class of functions from a domain $\mathcal{Z}$ to $[0, B]$ with pseudo-dimension $d_{\mathrm{p}} := \mathrm{Pdim}(\mathcal{F})$. Then for every $n \ge d_{\mathrm{p}}$ and every $\varepsilon \in (0, B)$,*

$$\mathcal{C}_\infty(\varepsilon, \mathcal{F}, n) \le \left( \frac{eBn}{\varepsilon d_{\mathrm{p}}} \right)^{d_{\mathrm{p}}}.$$

Proposition D.12 goes back to Anthony & Bartlett (2009). Combining Lemma D.11 with Proposition D.12 yields Rademacher complexity bounds of order $\sqrt{\mathrm{Pdim}(\mathcal{F})/n}$ up to logarithmic factors.

### D.4.2. PSEUDO-DIMENSION BOUNDS FOR DERIVATIVES OF $\tanh$ NETWORKS

We bound the pseudo-dimension of the classes generated by $u_{\boldsymbol{\theta}}$ and its first and second order derivatives by proceeding in two steps. First, each derivative $D^\alpha u_{\boldsymbol{\theta}}$ can be realized by an analytic network that reuses the original weights and biases, at the cost of a controlled increase in the number of computation nodes. Second, we invoke the VC-dimension bound for Pfaffian networks from (Karpinski & Macintyre, 1997), which applies to $\tanh$ since it is an analytic Pfaffian function.

**Lemma D.13** (Pseudo-dimension as VC dimension of subgraph thresholds). *Let $\mathcal{F} \subset \{f : X \to \mathbb{R}\}$ and define the associated threshold (subgraph) class*

$$\mathrm{thresh}(\mathcal{F}) := \Big\{ (x, t) \mapsto \mathbf{1}\{f(x) - t > 0\} \ : \ f \in \mathcal{F} \Big\} \subset \{0, 1\}^{X \times \mathbb{R}}.$$

*Then*

$$\mathrm{Pdim}(\mathcal{F}) = \mathrm{VCdim}\big(\mathrm{thresh}(\mathcal{F})\big).$$

*Proof.* The equivalence follows directly from the definitions of pseudo-shattering and shattering after mapping $f$ to the subgraph threshold map $(x, t) \mapsto \mathbf{1}\{f(x) > t\}$. A detailed argument is given in (Anthony & Bartlett, 2009, Chapter 11). $\square$

We will use the VC-dimension upper bound proved for Pfaffian networks[2] in (Karpinski & Macintyre, 1997, Section 4.2). We restate it in the parameterization needed below.

**Theorem D.14** (Pseudo-dimension bound via VC bounds for Pfaffian networks (Karpinski & Macintyre, 1997)). *Consider a feedforward* Pfaffian *network architecture A in the sense of Karpinski & Macintyre (1997, Section 4.2), with the following complexity parameters:*

- *$l \in \mathbb{N}$: number of trainable real parameters (weights/biases);*

- *$N \in \mathbb{N}$: number of computation nodes (non-input nodes);*

- *$d_{\mathrm{poly}} \in \mathbb{N}$: an upper bound on the degree of each node polynomial;*

- *$s \in \mathbb{N}$: an upper bound on the number of monomials in each node polynomial;*

- *$q \in \mathbb{N}$ and $D \in \mathbb{N}$: Pfaffian chain length and degree controlling all activation functions used by the architecture (as in Karpinski & Macintyre (1997, Section 4.2)).*

---

[2] The notions of Pfaffian networks and the associated complexity parameters $(l, N, q, D, d_{\mathrm{poly}}, s)$ are taken from (Karpinski & Macintyre, 1997). As the required background is quite involved, we refer the reader to said paper for detailed definitions and explanations.

*Let $\mathcal{F}(A) \subset \{f : \mathbb{R}^d \to \mathbb{R}\}$ be the resulting real-valued function class computed by $A$ (as the $l$ parameters vary freely in $\mathbb{R}^l$).*

*Define*

$$\Psi(l, N, q, D, d_{\text{poly}}, s) := lNq(lNq-1) + (16 + 2\log_2 d_{\text{poly}} + 2\log_2 s)\, l + 2(l + 2lNq)\log_2 l + 2(l + 2lNq)\log_2(d_{\text{poly}} + D).$$

*Then*

$$\text{Pdim}\big(\mathcal{F}(A)\big) \leq \Psi\big(l,\, N+1,\, q,\, D,\, d_{\text{poly}},\, s+1\big).$$

*Proof.* By Lemma D.13,

$$\text{Pdim}\big(\mathcal{F}(A)\big) = \text{VCdim}\big(\text{thresh}(\mathcal{F}(A))\big).$$

Now consider the augmented architecture $A^{\text{sub}}$ with one additional input $t \in \mathbb{R}$ and output gate computing the Boolean predicate

$$(x, t) \mapsto \mathbf{1}\{f(x) - t > 0\}, \quad \text{where } f \in \mathcal{F}(A).$$

Concretely, $A^{\text{sub}}$ is obtained by taking the original output polynomial and adding the extra term $(-t)$. This adds no trainable parameters, increases the number of computation nodes by at most 1 (if the subtraction is implemented as a separate final node), and increases the monomial count bound by at most 1. Thus $\text{thresh}(\mathcal{F}(A))$ is realized by a Pfaffian network with parameters bounded by $(l,\, N+1,\, d_{\text{poly}},\, s+1,\, q,\, D)$.

Applying the VC-dimension upper bound for Pfaffian networks from Karpinski & Macintyre (1997, Section 4.2, end of section) to $A^{\text{sub}}$ yields

$$\text{VCdim}\big(\text{thresh}(\mathcal{F}(A))\big) \leq \Psi\big(l,\, N+1,\, q,\, D,\, d_{\text{poly}},\, s+1\big),$$

and the result follows. $\qquad\square$

**Corollary D.15** (Implication of Theorem D.14 for fixed activations). *Assume $q, D, d_{\text{poly}}$ are absolute constants and $s \leq \text{poly}(N)$. Then there exists a constant $C > 0$ such that, for all $l, N \geq 2$,*

$$\text{Pdim}\big(\mathcal{F}(A)\big) \leq C\,(lN)^2 \log(lN).$$

We next need a lemma on the representation of products with analytic networks.

**Lemma D.16.** *Let $\varphi_{\text{sq}}(t) = t^2$. In a feedforward architecture allowing $\tanh$ and $\varphi_{\text{sq}}$ hidden units and a linear output, there exists a fixed constant-size subnetwork that maps two scalar inputs $(a, b) \in \mathbb{R}^2$ to the product $ab$ exactly.*

*Proof.* The identity $ab = \frac{1}{2}\big((a+b)^2 - a^2 - b^2\big)$ expresses multiplication using one addition, three squaring operations, and one final affine combination. These operations can be implemented by a constant number of units with fixed coefficients, independent of the surrounding network. $\qquad\square$

The next lemma shows that derivatives of $u_\theta$ with respect to the input variables can be represented by analytic networks with the same depth and with at most a constant factor more parameters.

**Lemma D.17.** *Fix an architecture $(m_\ell)_{\ell=0}^L$ with $m_0 = d$ and $m_L = 1$, and write*

$$P := \sum_{\ell=1}^{L}(m_\ell m_{\ell-1} + m_\ell), \quad N_{\text{hid}} := \sum_{\ell=1}^{L-1} m_\ell.$$

*Let $u_\theta$ be the corresponding fully connected $\tanh$ network (with linear output) on $\Omega$.*

*For every multi-index $\alpha \in \mathbb{N}_0^d$ with $1 \leq |\alpha| \leq 2$, there exists a feedforward Pfaffian network architecture $A_\alpha$ (in the sense of Karpinski & Macintyre (1997, Section 4.2)) with the following properties:*

1. *(Parameter sharing) The set of trainable parameters of $A_\alpha$ is exactly the original parameter vector $\theta \in \mathbb{R}^P$ (no additional free parameters).*

2. *(Exact derivative)* *For every $\theta$ and every $x \in \Omega$,*

$$A_\alpha(x; \theta) = D^\alpha u_\theta(x).$$

3. *(Computation nodes)* *The number $N_\alpha$ of computation nodes of $A_\alpha$ satisfies*

$$N_\alpha \le C_{d,|\alpha|} \, N_{\mathrm{hid}} + 3,$$

*where one may take*

$$C_{d,1} := 3 + 5d, \;\; C_{d,2} := 8 + 5d + \frac{13}{2} d(d+1).$$

*Proof.* Write the base $\tanh$ network as a feedforward computation graph with affine maps, $\tanh$ nonlinearities, and a final linear output. This fits the Pfaffian network framework of (Karpinski & Macintyre, 1997) because affine maps are polynomials and $\tanh$ is Pfaffian.

Fix a hidden neuron with pre-activation $a$ and activation $z = \tanh(a)$. Using only $\tanh$, squaring, affine combinations, and the exact multiplication subnetwork from Lemma D.16, we can compute $z^2$, $s := 1 - z^2$, and $t := -2z + 2z^3$. The identities $\tanh'(a) = s$ and $\tanh''(a) = t$ then hold exactly.

We construct $A_\alpha$ by propagating first and second input derivatives through the graph using the chain rule. For each input coordinate $i \in \{1, \ldots, d\}$, $\partial_i a$ is an affine combination of derivatives from the previous layer with coefficients given by the original weights, and $\partial_i z = s \, \partial_i a$ is obtained using one multiplication subnetwork. Similarly, for $1 \le i \le j \le d$, $\partial_{ij} a$ is affine in previous-layer derivatives and

$$\partial_{ij} z = t \, (\partial_i a)(\partial_j a) + s \, \partial_{ij} a$$

is implemented using three multiplications and one affine combination. All trainable coefficients are reused from $\theta$, while any additional coefficients introduced by squaring, addition, and the product construction are fixed constants. This gives parameter sharing and the exact derivative identity.

It remains to bound the number of computation nodes. Using the concrete multiplication construction of Lemma D.16, one multiplication can be implemented with a constant number of computation nodes, and we may upper bound this constant by 4. For each hidden neuron, computing $z$, $z^2$, and $s = 1 - z^2$ costs 3 nodes. For each $i$, computing $\partial_i a$ costs one affine node and computing $\partial_i z = s \, \partial_i a$ costs at most 4 nodes, for a total of $5d$ nodes. For second derivatives, computing $z^3$ and $t = -2z + 2z^3$ costs at most 5 additional nodes. For each pair $1 \le i \le j \le d$, the update for $\partial_{ij} z$ uses one affine node for $\partial_{ij} a$ and at most 3 multiplications, hence at most $1 + 3 \cdot 4 = 13$ nodes per pair. Since there are $d(d+1)/2$ such pairs, the per-neuron cost is bounded by $C_{d,1} = 3 + 5d$ for first derivatives and by $C_{d,2} = 8 + 5d + \frac{13}{2} d(d+1)$ for derivatives up to order 2. Multiplying by $N_{\mathrm{hid}}$ and adding a constant number of output nodes yields the claimed bound on $N_\alpha$. $\square$

We now define, for $k \in \{0, 1, 2\}$, the derivative classes

$$\mathcal{F}^{(k)} := \left\{ D^\alpha u_{\boldsymbol{\theta}} : u_{\boldsymbol{\theta}} \in \mathcal{U}_{L,m,B}, \; \alpha \in \mathbb{N}_0^d, \; |\alpha| = k \right\}, \tag{D.25}$$

with the convention that $D^0 u_{\boldsymbol{\theta}} = u_{\boldsymbol{\theta}}$. Lemma D.17 and Corollary D.15 together give the following quantitative bound.

**Proposition D.18** (Pseudo-dimension of $\tanh$ networks and their up-to-order-two derivatives). *Let $\mathcal{U}_{L,m,B}$ be as in (D.15) with architecture $(m_\ell)_{\ell=0}^L$, and define*

$$P := \sum_{\ell=1}^{L} (m_\ell m_{\ell-1} + m_\ell), \;\; N_{\mathrm{hid}} := \sum_{\ell=1}^{L-1} m_\ell, \;\; \text{and } N_k := C_{d,k} \, N_{\mathrm{hid}} + 3,$$

*with*

$$C_{d,0} := 1, \;\; C_{d,1} := 3 + 5d, \;\; \text{and } C_{d,2} := 8 + 5d + \frac{13}{2} d(d+1).$$

*Then there exists a constant $C_{\mathrm{pdim}} > 0$ depending only on $d$ and on the activation family (ultimately $\tanh$) such that for each $k \in \{0, 1, 2\}$,*

$$\mathrm{Pdim}\big(\mathcal{F}^{(k)}\big) \le C_{\mathrm{pdim}} \, (PN_k)^2 \log(PN_k).$$

*Proof.* Fix $k \in \{0, 1, 2\}$ and a multi-index $\alpha$ with $|\alpha| = k$. By Lemma D.17, the class $\mathcal{F}^{(\alpha)} := \{D^\alpha u_\theta : \theta \in \Theta_{L,m}\}$ is contained in the Pfaffian network class $\mathcal{F}(A_\alpha)$ of an architecture with $l = P$ trainable parameters and at most $N_k$ computation nodes. Hence

$$\mathrm{Pdim}(\mathcal{F}^{(\alpha)}) \le \mathrm{Pdim}(\mathcal{F}(A_\alpha)).$$

Applying Corollary D.15 (a consequence of Theorem D.14) gives

$$\mathrm{Pdim}(\mathcal{F}^{(\alpha)}) \le C(PN_k)^2 \log(PN_k)$$

for a constant $C$ depending only on the fixed activation family.

Finally, $\mathcal{F}^{(k)} = \bigcup_{|\alpha|=k} \mathcal{F}^{(\alpha)}$ is a finite union over $\binom{d+k-1}{k}$ multi-indices. Using the standard finite-union bound for pseudo-dimension (Anthony & Bartlett, 2009, Section 11.2) and absorbing $\binom{d+k-1}{k}$ into the constant yields the claim. $\square$

Proposition D.18 yields, to the best of our knowledge, the first rigorous bounds for first and second order derivatives of $\tanh$ networks. Together with Lemma D.11 and Proposition D.12, it yields the Rademacher complexity bounds for the PDE, boundary and FK losses stated later in this appendix.

### D.4.3. FINITE-SAMPLE BOUND ON THE STATISTICAL ERROR

We now bound $\mathcal{E}_{\mathrm{stat}}$ using the pseudo-dimension estimates from Proposition D.18. Recall that

$$\mathcal{R}(u) = \lambda_{\mathrm{PDE}} \mathcal{R}_{\mathrm{PDE}}(u) + \lambda_{\partial\Omega} \mathcal{R}_{\partial\Omega}(u) + \lambda_{\mathrm{FK}} \mathcal{R}_{\mathrm{FK}}(u),$$

and that $\widehat{\mathcal{R}}$ is defined in (D.8). We first separate the contributions of the three components.

**Lemma D.19** (Decomposition of the statistical error). *For every hypothesis class $\mathcal{U}$,*

$$\mathcal{E}_{\mathrm{stat}} \le \lambda_{\mathrm{PDE}}\, \mathcal{E}_{\mathrm{stat}}^{\mathrm{PDE}} + \lambda_{\partial\Omega}\, \mathcal{E}_{\mathrm{stat}}^{\partial\Omega} + \lambda_{\mathrm{FK}}\, \mathcal{E}_{\mathrm{stat}}^{\mathrm{FK}},$$

*where*

$$\mathcal{E}_{\mathrm{stat}}^{\mathrm{PDE}} := \sup_{u \in \mathcal{U}} \left| \mathcal{R}_{\mathrm{PDE}}(u) - \widehat{\mathcal{R}}_{\mathrm{PDE}}(u) \right|,$$

$$\mathcal{E}_{\mathrm{stat}}^{\partial\Omega} := \sup_{u \in \mathcal{U}} \left| \mathcal{R}_{\partial\Omega}(u) - \widehat{\mathcal{R}}_{\partial\Omega}(u) \right|,$$

$$\mathcal{E}_{\mathrm{stat}}^{\mathrm{FK}} := \sup_{u \in \mathcal{U}} \left| \mathcal{R}_{\mathrm{FK}}(u) - \widehat{\mathcal{R}}_{\mathrm{FK}}(u) \right|.$$

*Proof.* This is an immediate consequence of the triangle inequality and the linearity of $\mathcal{R}$ and $\widehat{\mathcal{R}}$ in their three components. $\square$

**Lemma D.20** (Pseudo-dimension and fixed multipliers / finite sums). *Let $\mathcal{F} \subset \{f : \mathcal{Z} \to \mathbb{R}\}$ be a function class with $\mathrm{Pdim}(\mathcal{F}) = d < \infty$ and let $g : \mathcal{Z} \to \mathbb{R}$ be a fixed (non-random) function. Then*

$$\mathrm{Pdim}(g\mathcal{F}) := \mathrm{Pdim}(\{z \mapsto g(z)f(z) : f \in \mathcal{F}\}) \le d.$$

*Moreover, if $\mathcal{F}_1, \ldots, \mathcal{F}_m$ are function classes with pseudo-dimensions $d_j := \mathrm{Pdim}(\mathcal{F}_j)$, and*

$$\mathcal{F}_{\mathrm{sum}} := \left\{ z \mapsto \sum_{j=1}^{m} f_j(z) \ : \ f_j \in \mathcal{F}_j \right\},$$

*then there exists a constant $C_m > 0$, depending only on $m$, such that*

$$\mathrm{Pdim}(\mathcal{F}_{\mathrm{sum}}) \le C_m \sum_{j=1}^{m} d_j.$$

*Proof.* For the first claim, fix a finite sample $z_1, \ldots, z_n$ and thresholds $r_1, \ldots, r_n \in \mathbb{R}$. If we replace the class $\mathcal{F}$ by $g\mathcal{F}$, the sign of $g(z_i)f(z_i) - r_i$ at each point $z_i$ is determined by the sign of $f(z_i) - r_i/g(z_i)$ whenever $g(z_i) \neq 0$, while points with $g(z_i) = 0$ contribute no additional sign variability. Thus the number of sign patterns achievable by $g\mathcal{F}$ on any finite sample is at most the number achievable by $\mathcal{F}$, so $\mathrm{Pdim}(g\mathcal{F}) \leq \mathrm{Pdim}(\mathcal{F})$.

The bound for finite sums is a special case of the general pseudo-dimension bound for linear combinations of real-valued function classes as shown by Anthony & Bartlett (2009, Chapter 11). Since the number $m$ of summands is fixed in our setting, its contribution can be absorbed into the constant $C_m$. □

We treat each term in Lemma D.19 with Lemma D.10 applied to an appropriate function class. We illustrate the argument for $\mathcal{E}_{\mathrm{stat}}^{\mathrm{PDE}}$; the remaining terms follow in the same way.

Define

$$\mathcal{F}_{\mathrm{PDE}} := \left\{ x \mapsto |\Omega| \left( \mathcal{L}u(x) - f(x) \right)^2 : u \in \mathcal{U} \right\}.$$

Define the (unsquared) residual class

$$\mathcal{G}_{\mathrm{PDE}} := \left\{ x \mapsto \mathcal{L}u(x) - f(x) : u \in \mathcal{U} \right\}.$$

By the bounds above we have

$$\sup_{g \in \mathcal{G}_{\mathrm{PDE}}} \sup_{x \in \Omega} |g(x)| \leq C_{\mathrm{PDE}}.$$

The mapping $u \mapsto \mathcal{L}u$ is linear in $u$ and involves only first and second order derivatives. In particular, every $g \in \mathcal{G}_{\mathrm{PDE}}$ can be written as a finite linear combination (with fixed coefficient functions given by the coefficients of $\mathcal{L}$ and by $-f$) of functions from the derivative classes $\mathcal{F}^{(0)}, \mathcal{F}^{(1)}, \mathcal{F}^{(2)}$ defined in (D.25). Note that for every $g \in \mathcal{G}_{\mathrm{PDE}}$ the function

$$x \mapsto |\Omega| \, g(x)^2$$

belongs to $\mathcal{F}_{\mathrm{PDE}}$, and the mapping $t \mapsto |\Omega| \, t^2$ is $2|\Omega|C_{\mathrm{PDE}}$-Lipschitz on $[-C_{\mathrm{PDE}}, C_{\mathrm{PDE}}]$. Hence, by the contraction inequality in Lemma D.10,

$$\mathfrak{R}_{N_{\mathrm{int}}}(\mathcal{F}_{\mathrm{PDE}}) \leq 2|\Omega|C_{\mathrm{PDE}} \, \mathfrak{R}_{N_{\mathrm{int}}}(\mathcal{G}_{\mathrm{PDE}}).$$

To bound $\mathfrak{R}_{N_{\mathrm{int}}}(\mathcal{G}_{\mathrm{PDE}})$ it suffices to control the Rademacher complexities of the derivative classes $\mathcal{F}^{(0)}, \mathcal{F}^{(1)}, \mathcal{F}^{(2)}$. Indeed, writing $\mathcal{L}$ in coordinates as

$$\mathcal{L}u(x) = \sum_{|\alpha| \leq 2} c_\alpha(x) \, D^\alpha u(x),$$

with bounded coefficient functions $c_\alpha$, we have for any fixed sample $x_1, \ldots, x_{N_{\mathrm{int}}}$,

$$\mathfrak{R}_{N_{\mathrm{int}}}(\mathcal{G}_{\mathrm{PDE}} \mid x_{1:N_{\mathrm{int}}}) \leq \sum_{|\alpha| \leq 2} \|c_\alpha\|_{L^\infty(\Omega)} \, \mathfrak{R}_{N_{\mathrm{int}}}(\mathcal{F}^{(|\alpha|)} \mid x_{1:N_{\mathrm{int}}}),$$

and the constant shift $-f$ does not affect Rademacher complexity. By Proposition D.18 (with $k \leq 2$) and the Dudley/covering-number bounds, this yields

$$\mathfrak{R}_{N_{\mathrm{int}}}(\mathcal{G}_{\mathrm{PDE}}) \lesssim (PN_2)\sqrt{\frac{\log(PN_2)}{N_{\mathrm{int}}}},$$

up to logarithmic factors and constants depending only on $(d, L)$ and the coefficients of $\mathcal{L}$.

Applying Lemma D.10, Lemma D.11 and Proposition D.12 to $\mathcal{F}_{\mathrm{PDE}}$, together with the bound $\mathfrak{R}_{N_{\mathrm{int}}}(\mathcal{G}_{\mathrm{PDE}}) \lesssim (PN_2)\sqrt{\log(PN_2)/N_{\mathrm{int}}}$, yields the following bound.

**Lemma D.21** (Statistical error for the PDE residual). *There exist constants $C_{\mathrm{PDE},1}, C_{\mathrm{PDE},2} > 0$, depending only on $(d, L, m, B)$ and on the coefficients of $\mathcal{L}$, such that for every $\delta \in (0, 1)$, with probability at least $1 - \delta$,*

$$\mathcal{E}_{\mathrm{stat}}^{\mathrm{PDE}} \leq C_{\mathrm{PDE},1}|\Omega|C_{\mathrm{PDE}}^2 \sqrt{\frac{(PN_2)^2 \log(PN_2) + \log(2/\delta)}{N_{\mathrm{int}}}} + C_{\mathrm{PDE},2}|\Omega|C_{\mathrm{PDE}}^2 \frac{(PN_2)^2 \log(PN_2) + \log(2/\delta)}{N_{\mathrm{int}}}.$$

The boundary term $\mathcal{E}_{\mathrm{stat}}^{\partial\Omega}$ is handled analogously, using that the boundary loss involves only $u_{\boldsymbol{\theta}}$ itself and no derivatives. In this case the relevant function class is generated by $\mathcal{F}^{(0)}$ and the same argument shows that

$$\mathcal{E}_{\mathrm{stat}}^{\partial\Omega} \lesssim |\partial\Omega| C_{\partial\Omega}^2 (PN_0)\sqrt{\frac{\log(PN_0) + \log(1/\delta)}{N_{\partial\Omega}}}. \tag{D.26}$$

for a suitable constant $C_{\partial\Omega}$.

The FK term requires a small modification. Recall that $\mathcal{R}_{\mathrm{FK}}(u)$ is defined using the true solution $u^\star$ in (D.3), while $\widehat{\mathcal{R}}_{\mathrm{FK}}(u)$ is defined with the noisy labels $\widehat{u}^{\mathrm{MC}}(x_k^{\mathrm{FK}})$. Writing

$$\widehat{u}^{\mathrm{MC}}(x_k^{\mathrm{FK}}) = u^\star(x_k^{\mathrm{FK}}) + b(x_k^{\mathrm{FK}}) + \zeta_k,$$

with $b$ and $\zeta_k$ as in Proposition 6.1, we deduce

$$\begin{aligned}
\mathcal{E}_{\mathrm{stat}}^{\mathrm{FK}} &= \sup_{u\in\mathcal{U}}\left|\mathbb{E}\big[(u(X^{\mathrm{FK}}) - u^\star(X^{\mathrm{FK}}))^2\big] - \frac{1}{N_{\mathrm{FK}}}\sum_{k=1}^{N_{\mathrm{FK}}}\big(u(x_k^{\mathrm{FK}}) - \widehat{u}^{\mathrm{MC}}(x_k^{\mathrm{FK}})\big)^2\right| \\
&\leq \sup_{u\in\mathcal{U}}\left|\mathbb{E}\big[(u(X^{\mathrm{FK}}) - u^\star(X^{\mathrm{FK}}))^2\big] - \frac{1}{N_{\mathrm{FK}}}\sum_{k=1}^{N_{\mathrm{FK}}}\big(u(x_k^{\mathrm{FK}}) - u^\star(x_k^{\mathrm{FK}})\big)^2\right| \\
&\quad + \sup_{u\in\mathcal{U}}\left|\frac{1}{N_{\mathrm{FK}}}\sum_{k=1}^{N_{\mathrm{FK}}}\big[(u(x_k^{\mathrm{FK}}) - u^\star(x_k^{\mathrm{FK}}))^2 - (u(x_k^{\mathrm{FK}}) - \widehat{u}^{\mathrm{MC}}(x_k^{\mathrm{FK}}))^2\big]\right| \\
&=: A_{\mathrm{FK}} + B_{\mathrm{FK}}.
\end{aligned}$$

The term $A_{\mathrm{FK}}$ can be bounded exactly as in Lemma D.21, now using only the class $\mathcal{F}^{(0)}$ on $\Omega$, since it is an empirical-process deviation for the squared error with respect to $u^\star$. For $B_{\mathrm{FK}}$ we expand the difference of squares and use that

$$(u - u^\star)^2 - (u - \widehat{u}^{\mathrm{MC}})^2 = 2\,(u - u^\star)\big(b(x_k^{\mathrm{FK}}) + \zeta_k\big) - \big(b(x_k^{\mathrm{FK}}) + \zeta_k\big)^2.$$

Using $\|u - u^\star\|_{C(\overline{\Omega})} \leq 2B$, the bias bound in Proposition 6.1, and the inequality $\big|\frac{1}{N_{\mathrm{FK}}}\sum_{k=1}^{N_{\mathrm{FK}}} a_k\big| \leq \sup_{1\leq k\leq N_{\mathrm{FK}}} |a_k|$, we obtain the deterministic estimate

$$\sup_{u\in\mathcal{U}}\left|\frac{1}{N_{\mathrm{FK}}}\sum_{k=1}^{N_{\mathrm{FK}}}\Big(2\,(u(x_k^{\mathrm{FK}}) - u^\star(x_k^{\mathrm{FK}}))\,b(x_k^{\mathrm{FK}}) - b(x_k^{\mathrm{FK}})^2\Big)\right| \leq 4B\,\|b\|_{L^\infty(\Omega)} + \|b\|_{L^\infty(\Omega)}^2.$$

Define the Monte Carlo bias level
$$\varepsilon_{\mathrm{bias}} := C_{\mathrm{bias}}\sqrt{\Delta t} + C_T e^{-\kappa T_{\max}},$$

so that $\|b\|_{L^\infty(\Omega)} \leq \varepsilon_{\mathrm{bias}}$. We keep the contribution $4B\,\varepsilon_{\mathrm{bias}} + \varepsilon_{\mathrm{bias}}^2$ explicit in the final bound.

The remaining stochastic contribution in $B_{\mathrm{FK}}$ involves the mean-zero sub-exponential fluctuations $\zeta_k$. A Bernstein-type bound yields a term of order $N_{\mathrm{FK}}^{-1/2}$ up to logarithmic factors, as shown in Appendix C. The genuinely stochastic part, involving the $\zeta_k$, is mean-zero and sub-exponential, and a standard Bernstein-type argument yields a contribution of order $\mathcal{O}(N_{\mathrm{FK}}^{-1/2})$ (up to logarithmic factors), as we show in Appendix C.

Combining these ingredients we obtain the following finite-sample bound on the full statistical error.

**Proposition D.22** (Finite-sample statistical error bound)**.** *Under Assumption D.1 and the standing assumptions on $\mathcal{U}$, there exist constants $C_{\mathrm{stat}}, C_{\mathrm{FK}} > 0$ such that for every $\delta \in (0,1)$, with probability at least $1 - \delta$,*

$$\begin{aligned}
\mathcal{E}_{\mathrm{stat}} \leq C_{\mathrm{stat}}&\left[(PN_2)\sqrt{\frac{\log(PN_2) + \log(6/\delta)}{N_{\mathrm{int}}}} + (PN_0)\sqrt{\frac{\log(PN_0) + \log(6/\delta)}{N_{\partial\Omega}}}\right] \\
&+ C_{\mathrm{FK}}\left[(PN_0)\sqrt{\frac{\log(PN_0) + \log(6/\delta)}{N_{\mathrm{FK}}}} + \sqrt{\frac{\log(6/\delta)}{N_{\mathrm{FK}}}} + \frac{\log(6/\delta)}{N_{\mathrm{FK}}}\right] + C_{\mathrm{bias}}\big(\varepsilon_{\mathrm{bias}} + \varepsilon_{\mathrm{bias}}^2\big),
\end{aligned} \tag{D.27}$$

where $\varepsilon_{\text{bias}} := C_{\text{bias}}\sqrt{\Delta t} + C_T e^{-\kappa T_{\max}}$ *is the FK Monte Carlo bias level from Proposition 6.1. In particular, for fixed network size $P$ and fixed confidence level, the first bracket in (D.27) tends to zero as $N_{\text{int}}, N_{\partial\Omega} \to \infty$, while the second bracket decays only as $N_{\text{FK}}^{-1/2}$ (up to logarithmic factors). Thus, when $N_{\text{FK}}$ is small or remains bounded, the FK contribution controls the asymptotic order of $\mathcal{E}_{\text{stat}}$ and induces a nonvanishing statistical floor of order*

$$\mathcal{E}_{\text{stat}} \gtrsim (PN_2)\sqrt{\frac{\log(PN_2)}{N_{\text{FK}}}}.$$

The dependence on $\log(6/\delta)$ in (D.27) comes from a simple union bound. We first apply Lemma D.21 to control $\mathcal{E}_{\text{stat}}^{\text{PDE}}$ with failure probability at most $\delta/3$ by replacing $\log(2/\delta)$ there with $\log(6/\delta)$. The same argument applied to the boundary term $\mathcal{E}_{\text{stat}}^{\partial\Omega}$ yields an analogous bound, again with failure probability at most $\delta/3$. Finally, the FK contribution $\mathcal{E}_{\text{stat}}^{\text{FK}}$ is controlled in the same way, with its own failure probability bounded by $\delta/3$ and with the same replacement of $\log(2/\delta)$ by $\log(6/\delta)$. A union bound over these three events shows that, with probability at least $1 - \delta$, all three component bounds hold simultaneously, which implies (D.27) via Lemma D.19.

## D.5. Putting it all together: non-asymptotic error bounds for FK-PINNs

We are now ready to combine the approximation, statistical and optimization bounds to prove the non-asymptotic error estimate announced in Theorem 6.2.

Let $\widehat{u} \in \mathcal{U}$ be an empirical minimizer of $\widehat{\mathcal{R}}$ and let $u_{\boldsymbol{\theta}_T}$ be the output of $T$ steps of gradient descent on $\ell(\boldsymbol{\theta}) = \widehat{\mathcal{R}}(u_{\boldsymbol{\theta}})$ starting from $\boldsymbol{\theta}_0$ as in Theorem D.8. We write

$$\mathcal{E}_{\text{tot}}(T) := \|u_{\boldsymbol{\theta}_T} - u^\star\|_{L^2(\Omega)}.$$

**Theorem D.23** (Non-asymptotic error bound for FK-PINNs). *Suppose that Assumption D.1 holds and that the assumptions of Corollary D.8 are satisfied, and that $u^\star \in W^{s,\infty}(\Omega)$ for some integer $s \geq 3$. Fix confidence level $\delta \in (0,1)$. Then there exist constants $C_{\text{app}}, C_{\text{stat}}, C_{\text{FK}}, C_{\text{opt}}, c_{\text{opt}} > 0$, independent of $(N_{\text{int}}, N_{\partial\Omega}, N_{\text{FK}})$ and of $T$, such that with probability at least $1 - \delta$ over the sampling of the collocation and FK points,*

$$\mathcal{E}_{\text{tot}}(T) \leq C_{\text{app}}P^{-\gamma} + C_{\text{stat}}\left[(PN_2)^{1/2}\left(\frac{\log(PN_2) + \log(6/\delta)}{N_{\text{int}}}\right)^{1/4} + (PN_0)^{1/2}\left(\frac{\log(PN_0) + \log(6/\delta)}{N_{\partial\Omega}}\right)^{1/4}\right]$$

$$+ C_{\text{FK}}\left[(PN_0)^{1/2}\left(\frac{\log(PN_0) + \log(6/\delta)}{N_{\text{FK}}}\right)^{1/4} + \left(\frac{\log(6/\delta)}{N_{\text{FK}}}\right)^{1/4} + \left(\frac{\log(6/\delta)}{N_{\text{FK}}}\right)^{1/2}\right]$$

$$+ C_{\text{bias}}\left(\varepsilon_{\text{bias}} + \varepsilon_{\text{bias}}^2\right)^{1/2} + C_{\text{opt}}\exp(-c_{\text{opt}}T/2).$$

$$\text{(D.28)}$$

*where $\gamma = (s-2)/d$ and $P$ is the maximal number of parameters of networks in $\mathcal{U}$. The constants $C_{\text{app}}, C_{\text{stat}}, C_{\text{FK}}, C_{\text{opt}}, c_{\text{opt}}$ may depend on the task weights $(\lambda_{\text{PDE}}, \lambda_{\partial\Omega}, \lambda_{\text{FK}})$ and on the PDE data, but not on the sample sizes $(N_{\text{int}}, N_{\partial\Omega}, N_{\text{FK}})$ nor on $T$.*

*Proof.* Fix $\delta \in (0,1)$ and work on the event where the statistical bound (D.27) of Proposition D.22 holds. Set

$$u_T := u_{\boldsymbol{\theta}_T},$$

let $\widehat{u}$ be an empirical minimizer of $\widehat{\mathcal{R}}$ as in Lemma D.2, and let $u^{\text{approx}} \in \mathcal{U}$ be the approximating network from Corollary D.5.

First we control the population risk of $u_T$. Adding and subtracting empirical risks gives

$$\mathcal{R}(u_T) - \mathcal{R}(u^\star) = \left[\mathcal{R}(u_T) - \widehat{\mathcal{R}}(u_T)\right] + \left[\widehat{\mathcal{R}}(u_T) - \widehat{\mathcal{R}}(\widehat{u})\right]$$
$$+ \left[\widehat{\mathcal{R}}(\widehat{u}) - \mathcal{R}(\widehat{u})\right] + \left[\mathcal{R}(\widehat{u}) - \mathcal{R}(u^\star)\right].$$

On the high probability event, the first and third brackets are bounded in absolute value by $\mathcal{E}_{\text{stat}}$. The second bracket is exactly $\mathcal{E}_{\text{opt}}(T)$ by (D.23). For the last bracket we apply Lemma D.2 with comparison function $u^{\text{approx}}$, which yields

$$\mathcal{R}(\widehat{u}) - \mathcal{R}(u^\star) \leq 2\,\mathcal{E}_{\text{stat}} + 2\left[\mathcal{R}(u^{\text{approx}}) - \mathcal{R}(u^\star)\right].$$

Since $\mathcal{R}(u^\star) = 0$ and $\mathcal{R}$ is nonnegative, this last term is at most $2\mathcal{R}(u^{\text{approx}})$. Combining these bounds we arrive at

$$\mathcal{R}(u_T) - \mathcal{R}(u^\star) \le 4\,\mathcal{E}_{\text{stat}} + \mathcal{E}_{\text{opt}}(T) + 2\,\mathcal{R}(u^{\text{approx}}). \tag{D.29}$$

By Lemma D.6 we have $\mathcal{R}(u^{\text{approx}}) \le \widetilde{C}_{\text{risk}} P^{-2\gamma}$, so

$$\mathcal{R}(u_T) - \mathcal{R}(u^\star) \le 4\,\mathcal{E}_{\text{stat}} + \mathcal{E}_{\text{opt}}(T) + C_1 P^{-2\gamma} \tag{D.30}$$

for a constant $C_1 > 0$ independent of the sample sizes and of $P$.

Next we use PDE stability. From (D.4) and the nonnegativity of $\mathcal{R}_{\text{FK}}$ we have, for every $u$,

$$\lambda_{\min}\big(\mathcal{R}_{\text{PDE}}(u) + \mathcal{R}_{\partial\Omega}(u)\big) \le \mathcal{R}(u) - \mathcal{R}(u^\star),$$

with $\lambda_{\min} = \min\{\lambda_{\text{PDE}}, \lambda_{\partial\Omega}\}$. Applying this with $u = u_T$ in (D.9) gives

$$\|u_T - u^\star\|^2_{L^2(\Omega)} \le \frac{C_{\text{stab}}}{\lambda_{\min}}\big(\mathcal{R}(u_T) - \mathcal{R}(u^\star)\big). \tag{D.31}$$

Substituting (D.30) into (D.31) we obtain

$$\|u_T - u^\star\|^2_{L^2(\Omega)} \le C_2\big(\mathcal{E}_{\text{stat}} + P^{-2\gamma} + \mathcal{E}_{\text{opt}}(T)\big),$$

for a constant $C_2 > 0$ independent of $(N_{\text{int}}, N_{\partial\Omega}, N_{\text{FK}})$ and of $T$. Taking square roots and using $\sqrt{a+b+c} \le \sqrt{a} + \sqrt{b} + \sqrt{c}$ gives

$$\mathcal{E}_{\text{tot}}(T) \le C_3\Big(\sqrt{\mathcal{E}_{\text{stat}}} + P^{-\gamma} + \sqrt{\mathcal{E}_{\text{opt}}(T)}\Big).$$

We now insert the statistical bound (D.27) and the optimization bound (D.24), which yields $\sqrt{\mathcal{E}_{\text{opt}}(T)} \le \sqrt{C_{\text{opt}}}\,\exp(-c_{\text{opt}}T/2)$, and then absorb constants into $C_{\text{app}}, C_{\text{stat}}, C_{\text{FK}}, C_{\text{opt}}, c_{\text{opt}}$. $\qquad\square$

*Remark* D.24 (Error bound in terms of depth and width). Recall that for networks in $\mathcal{U}_{L,m,B}$ the number of parameters satisfies

$$P(L,m) \le C_{\text{arch}}(d)\,L\,m^2.$$

Writing $\widetilde{P} := C_{\text{arch}}(d)\,L\,m^2$ and using $\log P \lesssim \log \widetilde{P} \lesssim \log(Lm)$, the bound (D.28) can be rewritten (up to a change of the constants) as

$$\mathcal{E}_{\text{tot}}(T) \;\lesssim\; (Lm^2)^{-\gamma} + L^2m^3\sqrt{\frac{\log(Lm) + \log(6/\delta)}{N_{\text{int}}}} + L^2m^3\sqrt{\frac{\log(Lm) + \log(6/\delta)}{N_{\partial\Omega}}}$$

$$+ L^2m^3\sqrt{\frac{\log(Lm) + \log(6/\delta)}{N_{\text{FK}}}} + \exp(-c_{\text{opt}}T),$$

with $\gamma = (s-2)/d$. Moreover $N_{\text{hid}} = \sum_{\ell=1}^{L-1} m_\ell \le (L-1)m$, hence $N_2 \le (C_{d,2}(L-1)m+3) \lesssim_d Lm$. Therefore $(PN_2) \lesssim_d L^2m^3$.

**Corollary D.25** (Depth-3 FK-PINN: optimal width and rate). *Assume the setting of Theorem D.23 and Corollary D.5, with $s \ge 3$ and $\gamma = (s-2)/d$. Fix the depth to $L = 3$. Using $P(L,m) \lesssim m^2$ in the master bound and suppressing logarithmic factors in the sample sizes and in $m$, we may write, for any training time $T \ge 0$,*

$$\mathcal{E}_{\text{tot}}(T) \;\lesssim\; m^{-2\gamma} + m^{3/2}N_{\text{int}}^{-1/4} + m\,N_{\partial\Omega}^{-1/4} + m\,N_{\text{FK}}^{-1/4} + \exp(-c_{\text{opt}}T/2) + \big(\varepsilon_{\text{bias}} + \varepsilon_{\text{bias}}^2\big)^{1/2},$$

*where $\varepsilon_{\text{bias}} := C_{\text{bias}}\sqrt{\Delta t} + C_T e^{-\kappa T_{\max}}$. In the data-scarce FK regime (with $N_{\text{int}}$ and $N_{\partial\Omega}$ large compared to $N_{\text{FK}}$), balance the approximation term $m^{-2\gamma}$ with the FK term $m\,N_{\text{FK}}^{-1/4}$. This gives the width*

$$m \;\asymp\; N_{\text{FK}}^{\frac{1}{4(1+2\gamma)}} = N_{\text{FK}}^{\frac{d}{4d+8(s-2)}}.$$

*For this architecture, apart from logarithmic factors and the optimization and bias terms, the bound yields the rate*

$$\mathcal{E}_{\text{tot}}(T) = \mathcal{O}\Big(N_{\text{FK}}^{-\frac{\gamma}{2+4\gamma}}\Big) = \mathcal{O}\Big(N_{\text{FK}}^{-\frac{s-2}{2d+4(s-2)}}\Big)\ (\text{up to logs}).$$

*once $N_{\text{int}}$ and $N_{\partial\Omega}$ are large enough for the terms $m^{3/2}N_{\text{int}}^{-1/4}$ and $mN_{\partial\Omega}^{-1/4}$ to be negligible.*

Corollary D.25 shows that, *if* the number of FK trajectories $N_{\mathrm{FK}}$ were allowed to grow, a depth-3 FK-PINN with optimally chosen width would achieve the rate $N_{\mathrm{FK}}^{-(s-2)/(2d+4(s-2))}$ up to logarithmic factors. This matches the asymptotic behaviour proved by Doumèche et al. (2025) for PINNs with dense physics sampling: in that ideal regime, FK supervision does not worsen the leading statistical exponent.

Our focus, however, is the genuinely sparse FK regime, where $N_{\mathrm{FK}}$ is fixed while $N_{\mathrm{int}}$ and $N_{\partial\Omega}$ may grow. In this setting the FK term behaves as an irreducible noise floor: even as $N_{\mathrm{int}}, N_{\partial\Omega} \to \infty$ and $T \to \infty$, the excess risk cannot be made uniformly small, because it is pinned down by the few noisy FK labels. This loss of uniform guarantees for the excess risk is the real "price" of FK supervision. We accept it in exchange for the PL$^*$ geometry and uniform conditioning provided by the FK term (Corollary D.8), which yield fast optimization (Proposition D.9). How to exploit a fixed FK budget (placement of FK points, choice of $\lambda_{\mathrm{FK}}$, etc.) so as to mitigate this statistical saturation while retaining the optimization benefits is an interesting avenue for future work.

# E. Numerical Experiments

We now illustrate our theoretical results by a series of numerical experiments.

## E.1. Equations Considered

We begin by specifying the concrete equations used to benchmark the proposed FK-PINNs framework. As discussed earlier, our focus is on regimes where traditional PINNs often become inaccurate or unstable, due to difficulties such multiscale coupling, singular perturbations, high frequencies etc. in the PDE data, leading to extremely stiff loss landscapes. Besides the Schrödinger equation introduced in the main text, we also consider the three following representative classes of problems:

### E.1.1. POISSON EQUATION

We consider a prototypical elliptic bounded value problem on a bounded domain $\Omega \subset \mathbb{R}^d$ with boundary $\partial\Omega$:

$$-\Delta u(x) = f(x) \qquad x \in \Omega, \tag{E.1}$$

with Dirichlet boundary conditions

$$u(x) = g(x) \qquad x \in \partial\Omega.$$

Here $u(x)$ denotes the stationary potential or steady-state field generated by a source term $f(x)$. In multiscale settings, $f(x)$ and/or the solution $u(x)$ can exhibit sharp layers and small-scale structures, which pose challenges for standard PINNs due to spectral bias.

### E.1.2. MEAN ESCAPE TIME

Consider an overdamped diffusion process in $\mathbb{R}^d$,

$$\mathrm{d}X_t = b(X_t)\,\mathrm{d}t + \sigma(X_t)\,\mathrm{d}W_t, \tag{E.2}$$

and a bounded domain $\Omega \subset \mathbb{R}^d$ with boundary $\partial\Omega$. The mean first exit time (or mean escape time) $\tau(x)$ from $\Omega$ solves the elliptic PDE

$$\mathcal{L}\,\tau(x) = -1, \qquad x \in \Omega, \tag{E.3}$$

with boundary condition

$$\tau(x) = 0, \qquad x \in \partial\Omega,$$

where $\mathcal{L}$ is the infinitesimal generator

$$\mathcal{L}\phi = -\nabla V \cdot \nabla\phi + \beta^{-1}\Delta\phi \tag{E.4}$$

where $\beta$ is inverse temperature. The function $\tau(x)$ quantifies how long, on average, trajectories starting from $x$ remain inside $\Omega$. In metastable or small-noise regimes, $\tau(x)$ can vary over exponentially large scales, leading to stiff elliptic problems characterized by sharp boundary layers and high-contrast magnitudes that are challenging for standard PINNs.

### E.1.3. COMMITTOR FUNCTION

For the same diffusion process

$$\mathrm{d}X_t = b(X_t)\,\mathrm{d}t + \sigma(X_t)\,\mathrm{d}W_t, \tag{E.5}$$

let $A, B \subset \mathbb{R}^d$ be two disjoint sets, and define $\Omega = \mathbb{R}^d \setminus (A \cup B)$. The committor function $q(x)$ is the probability that a trajectory starting at $x$ reaches $B$ before $A$. It is characterized as the solution of the elliptic boundary value problem

$$\mathcal{L}\,q(x) = 0 \quad x \in \Omega, \tag{E.6}$$

with boundary conditions

$$q(x) = 0 \quad x \in A, \qquad q(x) = 1 \quad x \in B,$$

where $\mathcal{L}$ is the same generator as above. The committor encodes the effective reaction coordinate in transition path theory and large deviation theory, typically exhibiting sharp interfaces between metastable basins. Such boundary layers and rare-event structures are difficult to resolve with conventional PINNs, especially under small-noise conditions.

## E.2. Evaluation of FK-PINNs against baseline PINNs

In this section, we evaluate our proposed FK-PINNs by testing them on the aforementioned PDEs, and comparing how they perform against standard PINNs as baseline. This choice is to isolate the effect of the Feynman-Kac supervision terms on the training dynamics.

### E.2.1. IMPLEMENTATION DETAILS

**Neural Network Architectures.** All neural networks (baseline and FK-PINNs) used for these experiments are fully connected $\tanh$ networks with 4 hidden layers, and 128 neurons per layer.

**Training Schedule.** All neural networks (baseline and FK-PINNs) used for these experiments are trained using the Adam optimizer for $30,000$ iterations, followed by $15,000$ iterations of L-BFGS to fine-tune the solution. For the Adam phase, we implement a step-based exponential decay schedule, where the initial learning rate of $10^{-3}$ is decayed every $k$ iterations by a factor of $\gamma = \exp(\ln(10^{-3})/(N/k))$, reaching a final learning rate of $10^{-6}$.

**Feynman-Kac data supervision.** To generate the FK "data points", we use Algorithm 1 and set $\Delta t = 10^{-3}$, $N_{\mathrm{MC}} = 500$, and $T_{\max} = \infty$ (we simulate the trajectories until escape). As for the number of FK points, we first choose a number $N_{\mathrm{coll}}$ of collocation points, and fix a proportion $p_{\mathrm{data}} \in (0, 1)$ of randomly selected points for which we compute the FK label. We thus define

$$N_{\mathrm{FK}} := \lfloor p_{\mathrm{data}} N_{\mathrm{coll}} \rfloor, \quad N_{\mathrm{int}} := N_{\mathrm{coll}} - N_{\mathrm{FK}}. \tag{E.7}$$

We fix $p_{\mathrm{data}} = 0.02$ for all experiments in E.2. We will further explore the effects of different $N_{\mathrm{MC}}$, $\Delta t$, and $p_{\mathrm{data}}$ on the experimental results in E.5.

**Metrics.** To evaluate the performance of the models, we use the $L^2$ and $H^1$ error metrics in both absolute and relative forms.

$$L^2 \text{ absolute error:} \quad L^2_{\mathrm{abs}}(f_\theta) = \sqrt{\frac{1}{N} \sum_{n=1}^{N} |f_\theta(x_n) - u_{\mathrm{ref}}(x_n)|^2} \tag{E.8}$$

$$L^2 \text{ relative error:} \quad L^2_{\mathrm{rel}}(f_\theta) = \frac{\sqrt{\frac{1}{N} \sum_{n=1}^{N} |f_\theta(x_n) - u_{\mathrm{ref}}(x_n)|^2}}{\sqrt{\frac{1}{N} \sum_{n=1}^{N} |u_{\mathrm{ref}}(x_n)|^2}} \tag{E.9}$$

$$H^1 \text{ absolute error:} \quad H^1_{\mathrm{abs}}(f_\theta) = \sqrt{\frac{1}{N} \sum_{n=1}^{N} |f_\theta(x_n) - u_{\mathrm{ref}}(x_n)|^2 + \|\nabla f_\theta(x) - \nabla u_{\mathrm{ref}}(x)\|^2} \tag{E.10}$$

$$H^1 \text{ relative error:} \quad H^1_{\mathrm{rel}}(f_\theta) = \frac{\sqrt{\frac{1}{N} \sum_{n=1}^{N} |f_\theta(x_n) - u_{\mathrm{ref}}(x_n)|^2 + \|\nabla f_\theta(x) - \nabla u_{\mathrm{ref}}(x)\|^2}}{\sqrt{\frac{1}{N} \sum_{n=1}^{N} |u_{\mathrm{ref}}(x_n)|^2 + \|\nabla u_{\mathrm{ref}}(x)\|^2}} \tag{E.11}$$

### E.2.2. POISSON EQUATION

In this example, we solve the Poisson equation on the square $\Omega = [0, 1] \times [0, 1]$, with source term given by

$$f(x_1, x_2) = \pi^2 [51 \sin(\pi x_1) \sin(4\pi x_2) + 156 \sin(5\pi x_1) \sin(\pi x_2) + 24 \sin(2\pi x_1) \sin(2\pi x_2) + 244 \sin(5\pi x_1) \sin(6\pi x_2)],$$

And exact solution $u^*$:

$$u^*(x_1, x_2) = 3 \sin(\pi x_1) \sin(4\pi x_2) + 6 \sin(5\pi x_1) \sin(\pi x_2) + 3 \sin(2\pi x_1) \sin(2\pi x_2) + 4 \sin(5\pi x_1) \sin(6\pi x_2). \tag{E.12}$$

We set $N_{\mathrm{coll}} = 10000$ interior collocation points, and ($N_{\mathrm{bc}} = 400$) boundary collocation points.

Figure 4 compares the performance of PINNs and FK-PINNs on this Poisson problem with high-frequency oscillations. While both models capture the global solution profile, PINNs exhibit significant absolute errors (up to 9.0), particularly near the oscillatory peaks and boundaries. In contrast, by leveraging the "extra data", FK-PINNs effectively mitigate spectral bias, achieving a much finer reconstruction with absolute errors nearly an order of magnitude lower.

Figure 5 likewise demonstrates that FK-PINNs achieve superior convergence stability and a significantly lower PDE loss compared to standard PINNs, effectively mitigating the optimization stagnation observed in the vanilla baseline.

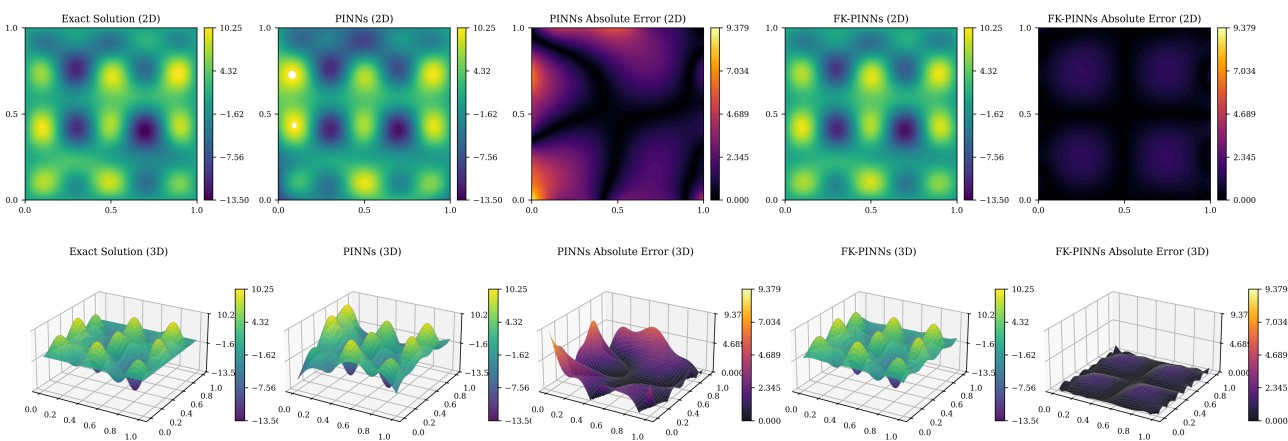

*Figure 4.* The ground truth solution ($Col.1$), predicted 2D,3D results by PINNs ($Col.2$), 2D,3D absolute error by PINNs ($Col.3$), predicted 2D,3D results by FK-PINNs ($Col.4$), 2D,3D absolute error by FK-PINNs ($Col.5$) on Poisson equations.

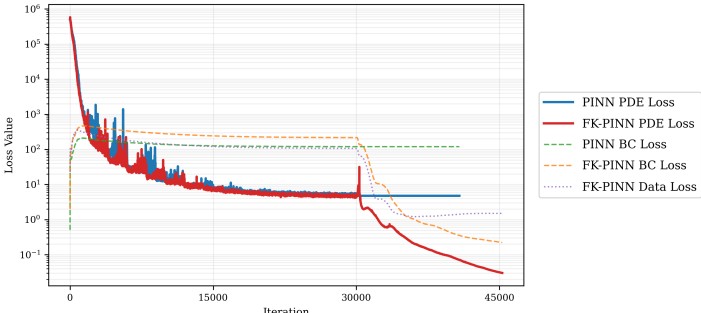

*Figure 5.* Comparison of PDE loss, BC loss of PINNs versus PDE loss, BC loss, Data loss of FK-PINNs for Poisson equation

### E.2.3. MEAN ESCAPE TIME

We set the domain $\Omega$ as the regular hexagon centered at the origin with circumradius $R = 2$. We set $V$ as a double-well potential function:

$$V\left(x_1, x_2\right) = \frac{1}{4}(x_1^2 - 1)^2 + \frac{\alpha}{2}x_2^2, \tag{E.13}$$

where $\alpha = 1$. The corresponding Mean Escape Time PDE is then given by:

$$-\nabla V \cdot \nabla \tau + \beta^{-1} \Delta \tau = -1, \tag{E.14}$$

with inverse temperature $\beta = 5$ and zero Dirichlet boundary condition.

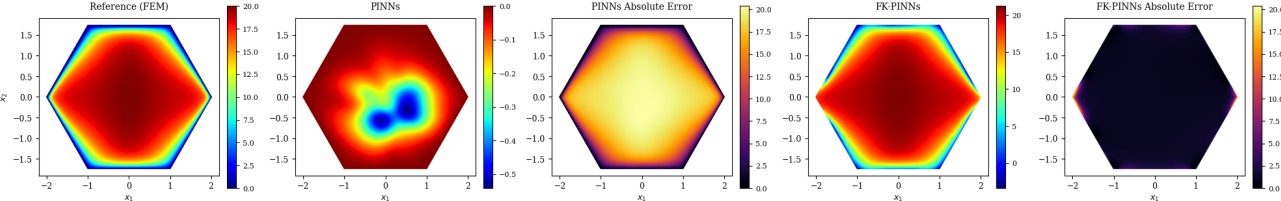

*Figure 6.* Comparison of Mean Escape Time PDE solutions learned by PINNs (left) and FK-PINNs(right)

Figure 6 shows the solutions learned both by PINNs and FK-PINNs. We see that PINNs (Cols. 2–3) exhibit *catastrophic failure* on this task, and completely fail to produce a meaningful solution. FK-PINNs (Cols. 4–5), on the other hand, successfully learn the solution and manage to capture the transition layers near the boundary, despite the corner singularities.

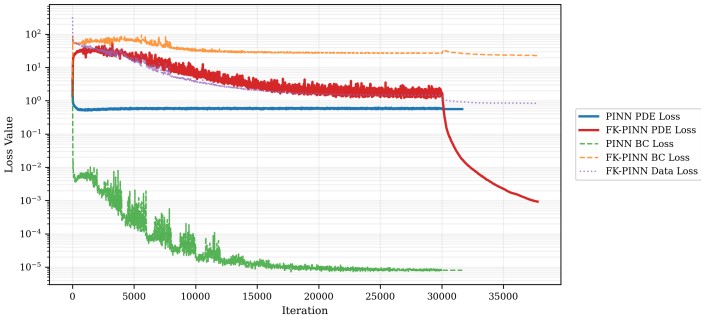

*Figure 7.* Comparing the evolution of loss components for solving the Mean Escape Time problem between PINNs and FK-PINNs

The training loss evolutions in Figure 7 reveal that standard PINNs encounter significant optimization stagnation after the initial phase, seemingly getting stuck around a spurious local minimum for the PDE loss. We see on the other hand that FK-PINNs manage to break through the poorly conditioned loss landscape and converge to the true solution.

### E.2.4. COMMITTOR FUNCTION

We now report the performance of FK-PINNs on the committor function problem on the domain $\Omega = [-1.5, 1.2] \times [-0.2, 2]$ with Müller-Brown potential, which is a standard benchmark in transition path studies (E & Vanden-Eijnden, 2006; Vanden-Eijnden et al., 2010). It is defined as the sum of four exponentials:

$$V(x,y) = \sum_{k=1}^{4} A_k \exp \left[ a_k(x - x_k^0)^2 + b_k(x - x_k^0)(y - y_k^0) + c_k(y - y_k^0)^2 \right] \tag{E.15}$$

The specific coefficients used to define the potential energy surface in this work are provided in Table 3. These parameters ensure the presence of the characteristic "three-well" structure often used to benchmark transition path sampling and committor function approximations.

We further define two metastable states, $A$ and $B$, which correspond to the reactant and product regions, respectively. These states are represented as circular boundary regions $\bar{\Omega}_A$ and $\bar{\Omega}_B$ with a fixed radius of $r = 0.2$. The centers of these disks, $\mathbf{c}_A$ and $\mathbf{c}_B$, are positioned at the primary local minima of the potential:

- **State $A$ (Reactant):** $\mathbf{c}_A = (-0.5582, 1.4417)$, satisfying the boundary condition $q(\mathbf{x}) = 0$.

- **State $B$ (Product):** $\mathbf{c}_B = (0.6235, 0.0281)$, satisfying the boundary condition $q(\mathbf{x}) = 1$.

The domain for the committor equation is thus defined as $\Omega' := \Omega \setminus (\bar{\Omega}_A \cup \bar{\Omega}_B)$

*Table 3.* Parameters for the Müller-Brown Potential Energy Surface.

| Index ($k$) | $A_k$ | $a_k$ | $b_k$ | $c_k$ | $x_k^0$ | $y_k^0$ |
|---|---|---|---|---|---|---|
| 1 | $-200$ | $-1$ | 0 | $-10$ | 1.0 | 0.0 |
| 2 | $-100$ | $-1$ | 0 | $-10$ | 0.0 | 0.5 |
| 3 | $-170$ | $-6.5$ | 11 | $-6.5$ | $-0.5$ | 1.5 |
| 4 | 15 | 0.7 | 0.6 | 0.7 | $-1.0$ | 1.0 |

As highlighted in Figure 8, the PINN solution fails to meaningfully capture the transition regions and the dynamics induced by the loss landscape, unlike FK-PINNs which recover a fully satisfactory solution. The improved training dynamics are also apparent in Figure 9.

### E.3. FK supervision improves the loss landscape

We now collect both qualitative and quantitative evidence supporting the improvement of the loss landscape as predicted by Theorem 5.4.

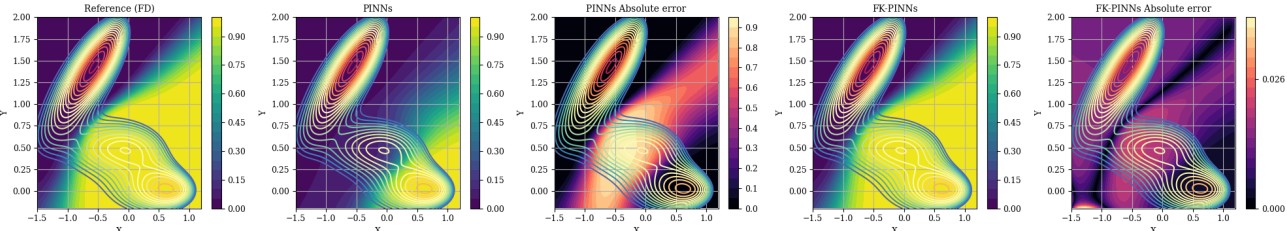

*Figure 8.* Comparison of Committor Function under Müller-Brown potential learned by PINNs (left) and FK-PINNs (right)

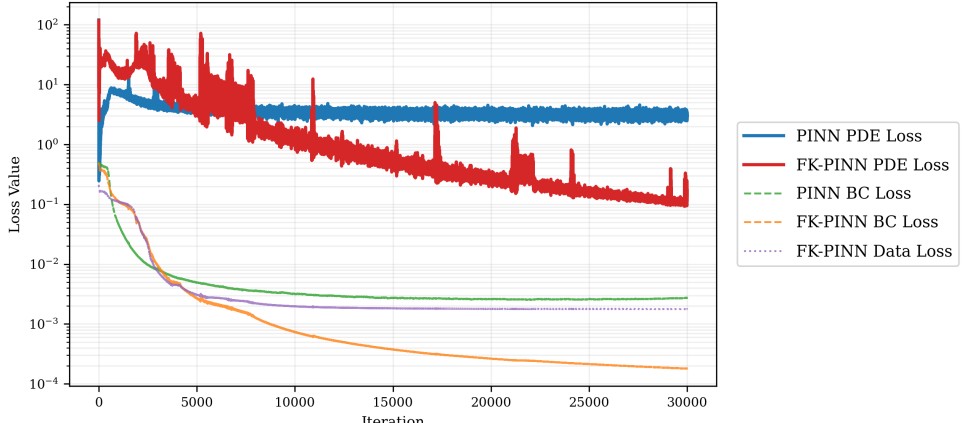

*Figure 9.* Comparison of the convergence behavior of individual loss components in PINNs and FK-PINNs trained with Adam optimizer only

| Model | Metric | Poisson (E.2.2) | Schrödinger-type (7) | Mean Escape Time (E.2.3) | Committor (E.2.4) |
|---|---|---|---|---|---|
| PINNs | $L^2$ Abs Err | $1.333 \pm 0.616$ | $0.475 \pm 0.148$ | $16.56 \pm 0.055$ | $1.028 \pm 0.283$ |
| | $L^2$ Rel Err | $0.322 \pm 0.149$ | $0.624 \pm 0.195$ | $1.007 \pm 0.003$ | $0.839 \pm 0.661$ |
| | $H^1$ Abs Err | $12.42 \pm 4.345$ | $2.893 \pm 0.415$ | $43.55 \pm 0.039$ | $3.154 \pm 0.794$ |
| | $H^1$ Rel Err | $0.171 \pm 0.061$ | $0.512 \pm 0.073$ | $1.002 \pm 0.001$ | $0.547 \pm 0.074$ |
| FK-PINNs | $L^2$ Abs Err | $0.761 \pm 0.011$ | $0.073 \pm 0.008$ | $1.755 \pm 0.093$ | $0.791 \pm 0.039$ |
| | $L^2$ Rel Err | $0.1184 \pm 0.003$ | $0.096 \pm 0.010$ | $0.107 \pm 0.006$ | $0.030 \pm 0.008$ |
| | $H^1$ Abs Err | $7.822 \pm 0.412$ | $1.043 \pm 0.083$ | $1.755 \pm 0.093$ | $1.495 \pm 0.105$ |
| | $H^1$ Rel Err | $0.108 \pm 0.006$ | $0.185 \pm 0.015$ | $0.395 \pm 0.051$ | $0.153 \pm 0.049$ |

*Table 4.* Performance comparison between standard PINNs and FK-PINNs on various problems. All results are averaged over 5 independent runs with different random seeds.

### E.3.1. HESSIAN CONDITION NUMBER

To provide quantitative justification for the superior convergence of FK-Enhanced PINNs, we analyze the local curvature of the loss landscape near a minimizer through the lens of the empirical Hessian matrix, $H = \nabla^2_\theta \mathcal{L}(\theta)$. As discussed in Section 5, the conditioning of the Hessian governs the stability and speed of gradient-based optimization.

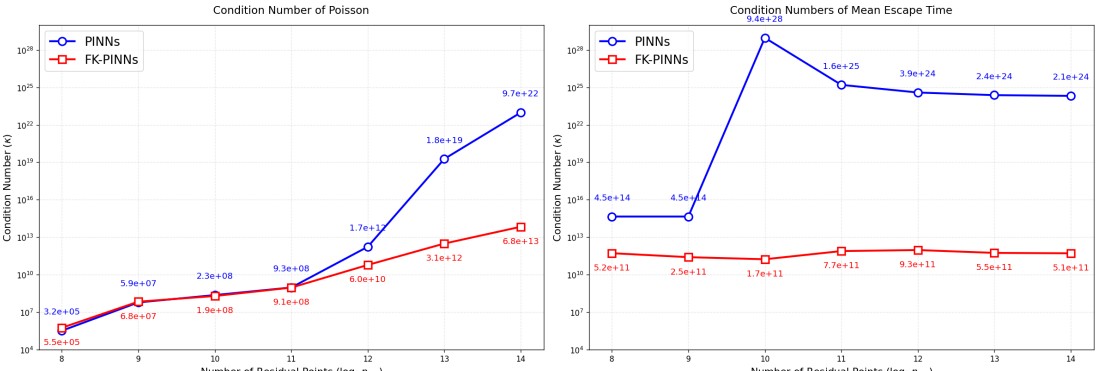

*Figure 10.* Condition numbers near a local minimum for PINNs and FK-PINNs for Poisson equation (left) and the Mean Escape Time Problem (right).

The evolution of the Hessian condition number as the number of collocation points increases plotted in Figure 10, suggests, as predicted by Theorem 5.3, that for the standard PINN, the condition number grows polynomially fast with sampling density. The condition number of FK-PINNs however, does not appear to follow such a growth rate, and remains for most sample sizes orders of magnitude below that of the standard PINN, in accordance with Theorem 5.4.

### E.3.2. VISUALIZING THE LOSS LANDSCAPE GEOMETRY

We now turn to a more qualitative understanding of the training difficulty for standard PINNs and FK-PINNs. To do so, we decide to directly visualize the loss landscape itself around a minimizer, by projecting the loss onto a 2D parameter subspace, and visually observe the result (Li et al., 2018; Krishnapriyan et al., 2021; Zhao et al., 2024). The resulting landscapes for the Mean Escape Time PDE, as seen in Figure 11, confirm what the theory and the other numerical results so far suggest: the PINN loss landscape exhibits a highly non-convex, jagged geometry, with many oscillations. This explains the failure of PINNs on this challenging PDE. On the other hand, the loss landscape of FK-PINNs appears much more regular and smooth, with a unique, well localized minimizer. This thus provides a vivid illustration of our theory.

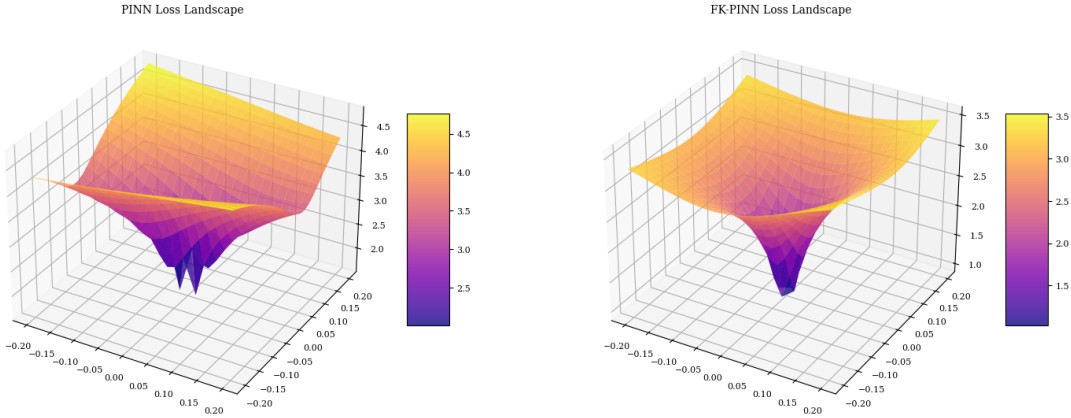

*Figure 11.* Loss landscape of Standard PINN (left) and FK-PINN (right) trained on the Mean Escape Time PDE, next to a minimizer.

## E.4. Comparison with other operator preconditioning techniques

Now that the improvements of FK-PINNs over the baseline PINNs is clear, we compare the FK-PINNs with two other "pre-conditioning inspired" approaches to train them. The first one we consider is the Energy Natural Gradient Descent algorithm (ENGD), proposed by Müller & Zeinhofer (2023), and which consists in performing gradient descent in the function space induced by the PDE. The second one is the NysNewtown-Conjugate Gradient Descent Algorithm (NNCG), proposed by Rathore et al. (2024), and which we run after Adam and L-BFGS, as the authors of said paper do to save computational cost.

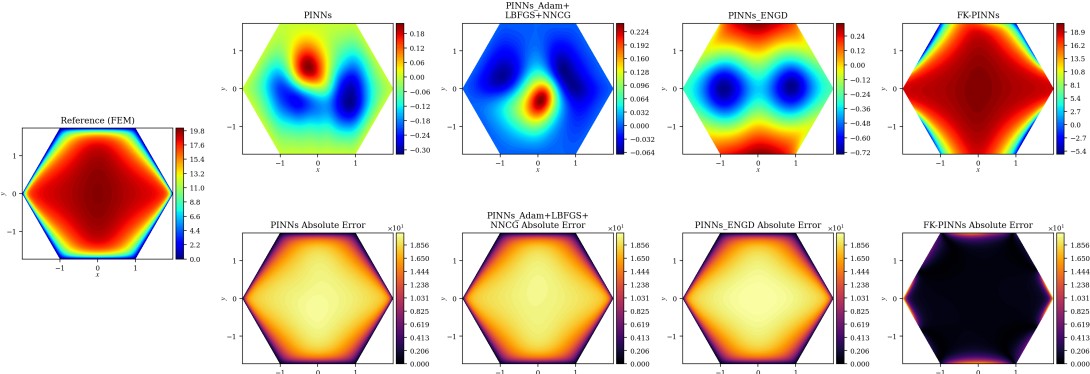

*Figure 12.* **Comparisons of different PINNs solving the Mean Escape Time PDE** trained with (from left to right) Standard PINN (Adam + L-BFGS), Adam + L-BFGS + NNCG, ENGD, FK-PINN method

| MODEL | L2 ERROR | H1 ERROR |
|---|---|---|
| PINNS (ADAM+L-BFGS) | $1.002 \pm 0.002$ | $1.001 \pm 0.001$ |
| PINNS (ADAM+L-BFGS+NNCG) | $0.986 \pm 0.011$ | $1.006 \pm 0.001$ |
| PINNS (ENGD) | $1.016 \pm 0.001$ | $1.006 \pm 0.014$ |
| FK-PINNS (ADAM+L-BFGS) | $\mathbf{0.174 \pm 0.004}$ | $\mathbf{0.699 \pm 0.008}$ |

*Table 5.* Relative $L^2$ and $H^1$ errors for each of the methods in Figure 12

As per the results reported in Figure 12 and Table 5, the other methods can not handle the stiff loss landscape of the Mean Escape Time PDE and fail just as much as the standard PINN does, despite their non-negligible computational overhead.

## E.5. Influence of the Monte Carlo budget

Finally, we report how the "Monte Carlo budget", i.e., the number $N_{\text{FK}}$ of FK data points we allow ourselves to use relative to the number of collocation points, the number $N_{\text{MC}}$ of paths we use to compute the Monte Carlo average, and the timestep $\Delta t$ we use to discretize the SDE influence the performance of FK-PINNs. We take $T_{\max} = \infty$ in all of the experiments presented below.

As shown in Figure 13, and as one could expect, too coarse discretization $\Delta t$ of the SDE leads to solutions which perform worse than the baseline PINN, which could be explained by the large amount of noise in the labels shifting the objective away from the true PDE solution. However we observe clear solution improvement for any discretization finer than $\Delta t = 10^{-3}$, which is well within the standard discretization sizes used to simulate SDE solutions. We likewise observe that for $\Delta t = 10^{-3}$, as even as little as $N_{\text{MC}} = 300$ provides clear improvement over the baseline, suggesting that even modest Monte Carlo computational budgets are already sufficient to "lift" the poorly conditioned PINN landscape with our method.

We finally show in Table 6 how the proportion $p_{\text{data}}$ of points used as FK supervision influences the solution quality. Perhaps contrary as to what one would expect, we find that, for the Mean Escape Time at least, there are diminishing returns when adding FK supervision data: while the solution quality improves as $p_{\text{data}}$ slowly increases from zero, we quickly find that there is a threshold where there is no clear improvement in the solution quality, while the time needed to simulate all the $N_{\text{MC}} \times N_{\text{FK}}$ paths until escape becomes prohibitively large, as the mean escape time is by design costly to estimate by

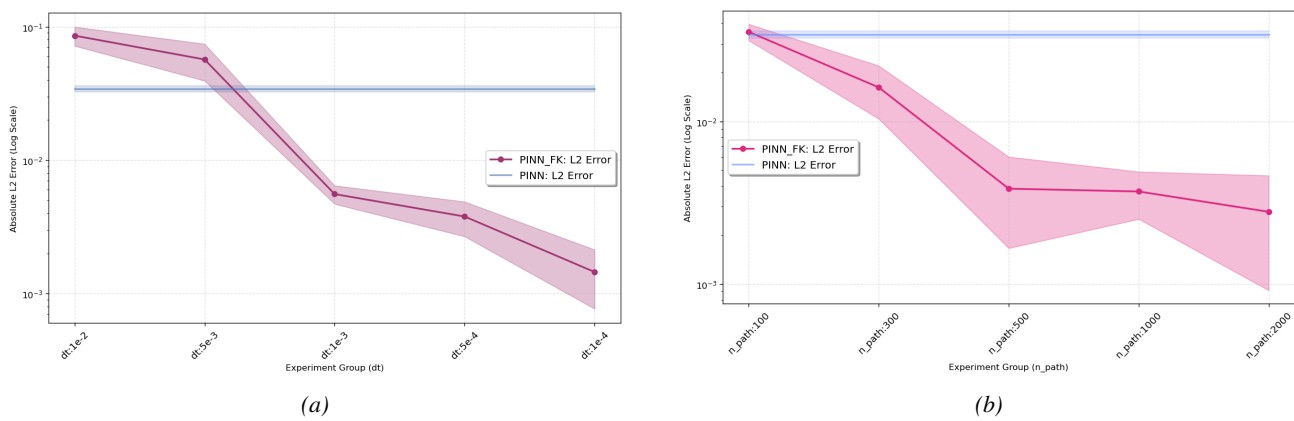

*(a)*                            *(b)*

*Figure 13.* Sensitivity Analysis of Key Parameters in Feynman-Kac Monte Carlo Supervision. Figure (a): Fixed $N_{\mathrm{MC}}$=500, influence of $\Delta t$ on solution accuracy. Figure (b): Fixed $\Delta t = 1e - 3$, influence of $N_{\mathrm{MC}}$ on solution accuracy

| P_DATA | L2 ERROR | H1 ERROR | TIME (S) |
|---|---|---|---|
| P_DATA=0 | $1.003 \pm 0.004$ | $1.001 \pm 0.001$ | $319.7 \pm 5.5$ |
| P_DATA=0.001 | $0.189 \pm 0.037$ | $0.594 \pm 0.069$ | $456.3 \pm 93.1$ |
| P_DATA=0.01 | $0.120 \pm 0.025$ | $0.431 \pm 0.143$ | $665.7 \pm 14.1$ |
| P_DATA=0.02 | $0.107 \pm 0.007$ | $0.353 \pm 0.086$ | $807.4 \pm 22.1$ |
| P_DATA=0.05 | $0.111 \pm 0.011$ | $\mathbf{0.224 \pm 0.072}$ | $1309.7 \pm 19.6$ |
| P_DATA=0.1 | $\mathbf{0.098 \pm 0.007}$ | $0.299 \pm 0.081$ | $2198.2 \pm 7.7$ |
| P_DATA=0.2 | $0.106 \pm 0.010$ | $0.382 \pm 0.081$ | $3968.9 \pm 25.1$ |

*Table 6.* Influence of the choice of $p_{\mathrm{data}}$ on the time overhead and the relative $L^2$ and $H^1$ error metrics. The PINNs here are trained to solve the Mean Escape Time PDE, and we've taken $\Delta t = 10^{-3}$, $N_{\mathrm{MC}} = 500$
.

Monte Carlo averages.

These observations suggest that it would be sensible, depending on the PDE being solved, to first fix $\Delta t$, and then preemptively simulate some Euler-Maruyama trajectories of the underlying stochastic process $X_t$ until escape, in order to estimate roughly the amount of time needed to fully simulate a path. One can then tune $p_{\mathrm{data}}$, $N_{\mathrm{MC}}$, and possibly $T_{\mathrm{max}}$ according to both the time and computational budget one has at disposition.

