# OpenReview forum: "Taming the Loss Landscape of PINNs with Noisy Feynman–Kac Supervision: Operator Preconditioning and Non-Asymptotic Error Bounds"
_ICML.cc/2026/Conference — ICML 2026 regular_

### Official Review · Reviewer_2PFp · 2026-03-04

**Soundness:** 2
**Presentation:** 2
**Significance:** 3
**Originality:** 3
**Overall Recommendation:** 5
**Confidence:** 4

**Summary:**

This paper introduces Feynman-Kac-Preconditioned Physics-Informed Neural Networks (FK-PINNs) to address the severe optimization pathologies and ill-conditioned loss landscapes that frequently cause standard PINNs to fail on complex PDEs.
To regularize the training dynamics without sacrificing the mesh-free advantage of PINNs, the authors propose augmenting the standard PDE residual and boundary losses with a sparse data-fidelity term. The target values for this data term are obtained by evaluating the Feynman-Kac (FK) formula via independent Monte Carlo (MC) simulations at a small number of interior spatial points
Theoretically, the authors prove that the introduction of a data-fidelity loss bounds the condition number, guaranteeing stable optimization. They also rigorously evaluate the Monte Carlo error of the FK supervision and derive non-asymptotic generalization bounds for FK-PINNs. Empirically, experiments on challenging PDEs confirm that FK-PINNs successfully smooth the loss landscape and converge accurately where baseline PINNs catastrophically fail.

**Compliance With Llm Reviewing Policy:**

Affirmed.

**Final Justification:**

Soundness: My main concern regarding the gap between the theoretical assumptions and the practical algorithm in Theorem 6.2 has been fully resolved by the authors' commitment to explicitly include the bias term in the error bound and move the current formulation to a corollary.
Presentation & Significance: The authors agreed to rewrite Section 5 to highlight the general applicability of the pre-conditioning theory (rather than limiting it to FK-supervision), which greatly enhances the paper's broader impact. Furthermore, incorporating a condensed version of the quantitative results (Table 3) into the main text provides the necessary empirical support that was previously missing.
Overall, the combination of operator pre-conditioning and mesh-free Feynman-Kac simulation is a highly original and mathematically robust contribution to the field.
The rebuttal addressed my concerns, so I have increased my score to 5.

**Key Questions For Authors:**

**1. Gap between theoretical assumptions and empirical setup in Theorem 6.2 (Critical for Soundness)**
In the derivation of Theorem 6.2 (specifically Appendix D.4.3), the simulation bias terms originating from the time discretization ($\Delta t$) and truncation horizon ($T_{max}$) are absorbed into the approximation error by assuming they are chosen as functions of $N_{FK}$. However, in the actual algorithm (Algorithm 2) and the sensitivity analysis (Appendix E.5, Figure 13), $\Delta t$ and $N_{FK}$ are treated and tuned as strictly independent hyperparameters.
*   **Question:** To better bridge this gap between the theoretical assumptions and the practical algorithm, would you consider modifying the bound in Theorem 6.2 to explicitly retain the bias term (e.g., $O(\sqrt{\Delta t}) + O(e^{-\kappa T_{max}})$), or perhaps adding a Corollary that accurately reflects the dependency on these practical hyperparameters? I believe doing so would significantly strengthen the theoretical soundness of the paper.


**2. Generalization of the pre-conditioning theory in Section 5 (Critical for Presentation/Significance)**
The current presentation of Theorem 5.4 gives the impression that the condition number bound (the eigenvalue lifting effect) is a unique property of the "Feynman-Kac (FK) supervision." However, as is evident from the proof (Appendix B.5) and Remark B.4, this pre-conditioning effect is a fundamental mathematical property of adding a general data term (squared error) under the interpolation regime (Assumption 5.1). Presenting this as an "FK-specific" effect undervalues your excellent general theoretical contribution and limits its reusability by other researchers.
*   **Question:** Would you be willing to rewrite Section 5 (specifically Theorem 5.4) to reflect its general applicability to standard data losses? You can then elegantly frame the FK supervision as a highly efficient, mesh-free realization that perfectly satisfies this general theorem. Addressing this will positively impact my evaluation of the paper's Presentation and Significance.

**3. Lack of quantitative evaluation in the main text (Critical for Presentation)**
Currently, the main text (Section 7) only contains qualitative visualizations for a single task. All quantitative evaluation results (such as the comprehensive error metrics in Table 3) are relegated to the appendix. This is structurally problematic as it forces readers to search the appendix to verify the primary claim that FK-PINNs significantly outperform baselines.
*   **Question:** Given space constraints, how do you plan to incorporate a summarized version of the quantitative comparisons (e.g., a condensed Table 3) into the main text? Including direct empirical evidence in the main body is essential for properly supporting your claims and will directly improve the Presentation score.

**Limitations:**

yes

**Strengths And Weaknesses:**

**Soundness**
*   **Strengths:** The paper is technically robust, and its claims are well-supported by theoretical analysis. In particular, the optimization theory proving that the addition of a data loss uniformly improves the condition number of finite-width PINNs (Theorem 5.4) and the derivation of non-asymptotic error bounds (Theorem 6.2) are excellent mathematical contributions.
*   **Weaknesses:**
    *   In the statement of Theorem 6.2, the simulation bias terms are absorbed into the approximation error, making it difficult to understand how the accuracy of the FK-supervision affects the final generalization performance. This creates a gap between the theoretical assumptions and the practical algorithm, where hyperparameters like $\Delta t$ and $N_{FK}$ are tuned independently. *(See Key Questions for details)*

**Presentation**
*   **Strengths:** The background of the study is very well explained. The context of why standard PINNs face optimization challenges and why pre-conditioning is necessary is clearly conveyed to the reader.
*   **Weaknesses:**
    *   The theoretical results in Section 5 (Theorem 5.4) are generally applicable to the pre-conditioning effect achieved by adding any data term, but the current presentation makes it seem as if this effect is unique to FK supervision. This undervalues the paper's broad theoretical contributions and limits its reusability. *(See Key Questions for details)*
    *   It is structurally problematic that all quantitative evaluation results (e.g., Table 3) are relegated to the appendix and completely absent from the main text, which weakens the direct empirical support for the claims. *(See Key Questions for details)*

**Significance**
*   **Strengths:** Addressing the critical issue of ill-conditioning that hinders PINN optimization, this paper proposes a "practical" pre-conditioning method using FK-supervision. The fact that this approach comes with solid theoretical guarantees makes it a highly significant contribution to the field.

**Originality**
*   **Strengths:**  The idea of realizing the pre-conditioning effect of data losses in finite-width networks (the mathematical contribution of Section 5) through "sparse self-supervision via the Feynman-Kac formula"—without sacrificing the mesh-free nature of PINNs—is highly novel.

---

> ### Author Rebuttal · Authors · 2026-03-31
>
> We thank the reviewer for their constructive suggestions, together with their positive assessment of our work and the appreciation of our theoretical results. Below we answer the questions raised by the referee.
>
>
> ## Weaknesses
>
>
> ### Soundness
>
>
> > “In the statement of Theorem 6.2, the simulation bias terms are absorbed into the approximation error, making it difficult to understand how the accuracy of the FK-supervision affects the final generalization performance. This creates a gap between the theoretical assumptions and the practical algorithm, where hyperparameters like \Delta t and N_{FK} are tuned independently.” (See Key Questions for details)
>
> We thank the reviewer for this opportunity to improve the presentation of our results. We initially opted to present the results in this manner to make the “main message” more apparent and streamlined, but we realize this choice might negatively impact their applicability and usefulness of our results. We will therefore make the dependence on the discretization parameter $\Delta t$ explicit in the revised version of the manuscript.
>
> ### Presentation
>
> > “The theoretical results in Section 5 (Theorem 5.4) are generally applicable to the pre-conditioning effect achieved by adding any data term, but the current presentation makes it seem as if this effect is unique to FK supervision. This undervalues the paper's broad theoretical contributions and limits its reusability.” (See Key Questions for details)
>
> We thank the reviewer for this excellent point that was also raised by other referees. The regularizing effect of adding supervised labels to the PINN objective is indeed agnostic to the source of this data and is thus applicable to a broader class of PDEs than those admitting a Feynman–Kac representation. We will revise the manuscript in a way that makes this message clear to the reader.
>
> > “It is structurally problematic that all quantitative evaluation results (e.g., Table 3) are relegated to the appendix and completely absent from the main text, which weakens the direct empirical support for the claims.” (See Key Questions for details)
>
> We understand and appreciate the reviewer's concern. The omission of further numerical evidence was mostly due to a lack of space in the main text, relative to the arguably large number of numerical experiments we've conducted to verify our theory. We intend to address this in the revised version of the manuscript by adding a short summary table in the last section of the main text (see our answer to your Key Questions below for details).
>
> ## Key Questions for Authors
>
> ### 1. Gap between theoretical assumptions and empirical setup in Theorem 6.2 (Critical for Soundness)
>
> We thank the reviewer for this astute suggestion. The only reason we “assumed” in Theorem 6.2 that $\Delta t$ and $T_{\max}$ are chosen as functions of $N_{FK}$ was to streamline the presentation. Including the explicit dependence of the error bound on the bias term is straightforward. In hindsight, we realize that this choice was not ideal, as it obscures the role that the discretization and truncation play in the theoretical analysis. To address this, we will, as per the referee’s suggestion, change the statement of Theorem 6.2 to explicitly include the bias term in the error bound and state the “$N_{FK}$ only” version of the theorem in a subsequent Corollary.
>
> ### 2. Generalization of the pre-conditioning theory in Section 5 (Critical for Presentation/Significance)
>
> We once again thank the referee for this thoughtful suggestion. We initially chose to focus on the effect of FK data terms on the loss landscape because we believed this to be a natural and realistic setting in which supervised data could be available. In light of your and other referees’ comments, we realize that this narrowing of the applicability of our results is counterproductive, as it limits the reusability of our conclusions in other settings where supervised data is available. We will therefore rewrite Section 5 to clearly highlight the broader applicability and validity of our results concerning the regularizing effects of data terms on the loss landscape and deduce the conclusion for FK-PINNs as a Corollary.
>
> ### 3. Lack of quantitative evaluation in the main text (Critical for Presentation)
>
> We thank the reviewer for this question. Because of space constraints, we initially omitted a detailed quantitative comparison from the main text. In the revision, we plan to include, as suggested by the reviewer, a condensed version of Table 3 in the main body, reporting only the $L^2$ absolute and relative errors of standard PINNs and FK-PINNs trained on the same PDEs. We believe the changes in writing and presentation we intend to make in Sections 5 and 6 will free enough space for such a table to be included.
>
> We hope that these replies address the referee’s concerns satisfactorily.

---

> > ### Author Rebuttal · Reviewer_2PFp · 2026-04-03
> >
> > Thank you for your thoughtful response. All of my concerns have been resolved. I will increase the score by one.

---

### Official Review · Reviewer_2yDz · 2026-03-10

**Soundness:** 3
**Presentation:** 3
**Significance:** 3
**Originality:** 4
**Overall Recommendation:** 4
**Confidence:** 2

**Summary:**

This paper introduces FK-PINNs, a method that enhances the training of Physics-Informed Neural Networks (PINNs). The key innovation of the proposed method is to add a Monte Carlo estimation term of the PDE solution derived from the Feynman-Kac (FK) probabilistic representation of the PDE into the PINN loss. Intuitively, the added FK data-fidelity term acts as an operator-level preconditioner, ensuring a bounded condition number even as the collocation point density increases, and eventually speeding up the gradient descent, as discussed in Section 5. The paper also provides theoretic analysis for FK-PINNs, including a non-asymptotic bound of the mean square error of FK-PINNs with tanh networks in Section 6. Numerical experiments on Poisson, Schrödinger, mean exit time, and committor problems demonstrate that FK-PINNs achieve more stable optimization, faster convergence, and lower solution error compared to standard PINNs, even with modest Monte Carlo sample budgets.

**Compliance With Llm Reviewing Policy:**

Affirmed.

**Final Justification:**

The authors' response has addressed most of my concern.

**Key Questions For Authors:**

1.When the problem domain exhibits a complex geometry, the accuracy and efficiency of the Monte Carlo procedure described in Section 3.2 may degrade. How does the proposed FK-PINN method address or mitigate this challenge?

2. How is the network trained in this work? In Assumption 5.1, it is assumed that the network is well-trained in an interpolation regime (i.e. the training loss is trained to zero) , while Theorem 6.2 employs a finite stopping time strategy ($T _ {max}$), which seems not consistent with the former assumption? How are these two aspects reconciled within the paper's theoretical framework?

3. Although the numerical simulations presented in Appendix E demonstrate the feasibility of narrow FK-PINNs (with width only $128$), the discussions in Section 6 are limited to the setting of sufficiently wide neural networks, which is inconsistent with the practical settings.

**Limitations:**

Discussed in the sections above.

**Strengths And Weaknesses:**

Strengths:

The paper introduces a novel and simple preconditioning technique to improve the performance of PINNs. It proposes a straightforward mesh-free modification to the standard PINN objective, adding a data-fidelity term corresponding with the Feynmann-Kac estimate of the solution into the PINN objective in order to address the root cause of PINN training failure---the ill-conditioned loss landscape inherited from the differential operator.

Besides, the paper offers both theoretical analysis and experimental results to support its claims. including a non-asymptotic error bound and a series of experimental comparisons between the performances of the proposed method and traditional PINNs on different settings of PDE problems.

Weaknesses:

1. Although the method's theoretical guarantees and practical implementation are developed for a ``broad class of linear second-order elliptic and parabolic PDEs’’, the paper simply claims that the discussions hold "under conditions ensuring that (2.1)–(2.2) admits a Feynmann-Kac representation" in Section 3.1, while lacking explicit discussions on when this Feynman-Kac representation can be admitted.

2. Likewise, some of the important concepts and terminologies in this paper lacks formal definitions or further clarifications. For example, the kernel integral operator $\mathcal{T} _ K$ and its corresponding kernel $K$ in Theorem 5.3, the Hessian-like operator $H$ and the mass operator $M$ in (2.6), have no formal definition in the main context.

---

> ### Author Rebuttal · Authors · 2026-03-31
>
> We thank the reviewer for the positive assessment of our work and the constructive suggestions. Below we address the concerns raised.
>
> ## Weaknesses
>
> ### 1. Conditions for admitting a Feynman–Kac representation
>
> We refer the reviewer to Appendix A (in particular Section A.1), where we provide a more exhaustive discussion of PDEs for which the Feynman–Kac representation formula applies. In the revised version, we will move a concise summary of these conditions into Section 3.1, so that the reader does not have to rely solely on the appendix.
>
> Furthermore, as was also highlighted in other reviews, the theoretical guarantees and benefits induced by the presence of “supervised data” in the loss are in fact agnostic to the source of these data points. In particular, our analysis applies in principle to any PDE for which such supervised interior data is available or can be computed at reasonable cost.
>
> ### 2. Undefined operators and terminology
>
> We appreciate this opportunity to clarify our terminology and make the paper more readable. Some of these objects are indeed not given fully explicit definitions in the main text because their precise formulations are technically involved and space in the main body is limited. In the revised manuscript, we will thus make an effort to avoid using undefined quantities in the main text, or at least describe in words, symbols whose precise mathematical definitions are left in the appendix.
>
> ## Key Questions for Authors
>
> ### 1. Complex geometries and Monte Carlo accuracy
>
> Our current theoretical analysis of the Monte Carlo error indeed relies on regularity assumptions on the domain boundary. This suggests that, for non-smooth or highly complex boundaries, one cannot expect the same accuracy guarantees as those derived under smoothness assumptions. This is corroborated by our numerical experiments for the mean exit time in polygonal domains (Appendix E.2.3), where the FK-PINN solution exhibits singular behavior near corners.
>
> At present, our FK-PINN implementation does not include a dedicated mechanism that completely eliminates this difficulty, and in challenging geometries it therefore inherits the limitations of the underlying Monte Carlo procedure. However, the framework naturally accommodates boundary-adapted sampling and local refinement strategies, for example placing more FK supervision points and allocating a larger Monte Carlo budget near corners in polygonal domains, which we expect to help mitigate the degradation induced by irregular boundaries in practice. Developing and rigorously analyzing such geometry-aware sampling and variance-reduction schemes is an important and non-trivial direction that we view as promising future work.
>
> ### 2. Interpolation regime vs finite stopping time
>
> We apologize for the confusing wording. In Assumption 5.1, by saying that the network is in an *interpolation regime*, we mean that the architecture is sufficiently overparameterized (i.e., wide enough) so that **there exists** at least one choice of parameters for which the network interpolates the training data, i.e., achieves zero training loss. The set $S$ of parameter values $\theta$ that yield zero empirical error is what we refer to as the “interpolation manifold.”
>
> Crucially, we do **not** assume that gradient descent (run for $T$ steps) actually finds a point in $S$. Theorem 6.2 explicitly analyzes the behavior of gradient descent with a finite training time, without requiring convergence to an interpolating solution. Thus, Assumption 5.1 is an architectural (existence) assumption about the capacity and landscape of the network, while Theorem 6.2 is a statement about the trajectory of a practical optimization algorithm stopped at a finite time. In the revision, we will clarify this distinction to avoid the impression that we are assuming exact interpolation is achieved in practice.
>
> ### 3. Width assumptions vs narrow architectures in experiments
>
> We appreciate the opportunity to clarify this important point: respectfully, we do not agree that the discussions in Section 6 regarding the width requirements are inconsistent with practical settings. Indeed, note that in Theorem 6.2 we only require the network width to grow polynomially with the number $N_{FK}$ of Feynman–Kac supervision points. In our intended regime of interest, $N_{FK}$ is small (i.e., we are in a “sparse supervision” regime), as also reflected in the numerical experiments in Appendix E. Consequently, the polynomial width requirement of the theorem is compatible with the relatively narrow architectures used in practice (e.g., width 128).
>
> We will make this dependence on $N_{\mathrm{FK}}$ more explicit in the discussion around Theorem 6.2 and better connect the theoretical scaling assumptions to the actual experimental setups reported in the appendix.
>
> We hope that these clarifications address the reviewer’s questions and concerns.

---

> > ### Author Rebuttal · Reviewer_2yDz · 2026-04-01
> >
> > Thank you for your response. It has addressed most of my concern.

---

### Official Review · Reviewer_h7Jj · 2026-03-12

**Soundness:** 3
**Presentation:** 3
**Significance:** 3
**Originality:** 2
**Overall Recommendation:** 4
**Confidence:** 3

**Summary:**

This paper addresses the well-known training difficulties of PINNs. The authors propose an extension of PINNs aimed particularly at enhancing convergence and providing an error analysis. This method is called "FK-PINNs" and is (currently) limited to 2nd order linear PDEs with Dirichlet BCs that admit a Feynman-Kac (FK) representation.

By means of the FK representation of the PDE solution, the authors compute additional solution values at interior points, which are approximate evaluations of the (weak) solution of the PDE. These additional data points form an extra objective, which is introduced as supervised loss term.

In doing so, the authors show that this extra data supervision loss term bounds the condition number uniformly with high probability, following the argumentation outlined in Rathore et al. (2024). Furthermore, the additional points, generated with FK, also allows the authors to derive non-asymptotic L2 error bounds.

**Compliance With Llm Reviewing Policy:**

Affirmed.

**Final Justification:**

The paper is solidly written with no major technical flaws observed. The theoretical contributions are genuine and interesting. I strongly appreciate the additional experiments provided during the rebuttal period, which addressed several of my concerns.

However, my main concern regarding the role and positioning of Feynman-Kac remains unresolved. The paper pursues two directions simultaneously without fully committing to either:

- As a new method (FK-PINNs), a comprehensive baseline comparison is missing -- the FEM supervision experiment added in rebuttal is a step in the right direction, but alternative pointwise methods and a comparison with more modern, more competitive PINN variants are not adequately discussed.
- As a theoretical contribution on PINNs with data loss, a discussion of when the theorem assumptions realistically hold and dedicated experiments verifying the theoretical predictions (e.g., the predicted scaling of condition numbers) are insufficiently developed.

Both directions individually yield interesting results worth publishing. However, by pursuing both simultaneously, the paper does not fully satisfy the standards of either. I therefore maintain my score of Weak Accept: the paper contains several valuable contributions that merit publication, but a higher rating would require the narrative to be sharpened and one of the two directions to be addressed more rigorously.

**Key Questions For Authors:**

(Q1) Have you compared FK supervision against FEM-based supervision empirically? If a coarse FEM solution (or even a few pointwise FEM evaluations) provides similar preconditioning benefits at lower cost, this would significantly impact the practical relevance of the FK-specific approach.

(Q2) When comparing FK-PINNs with standard PINNs (e.g., in section 7 or appendix E.2), do you also use adaptive weights (described in section 4.2) for the standard PINN? In particular, it is important to understand whether the difference in performance arises from the adaptive weighting or from the FK approach.

(Q3) How sensitive is the method to the choice of $\lambda_{FK}$? The adaptive weighting scheme (section 4.2) is mentioned, but ablation studies on this hyperparameter would be valuable.

**Limitations:**

yes

**Strengths And Weaknesses:**

# Strengths

The authors present an interesting method and substantiate their claims with a solid theoretical foundation, which is based on the Polyak-Lojasiewicz inequality and the work of Liu et al. (2022), De Ryck et al. (2024) and Rathore et al. (2024). Their theoretical analysis is on par with current research on PINNs. Particularly promising are the theoretical guarantees they provide, which may be extensible to general data-enhanced PINNs.
Furthermore, the illustrative example they provide is convincing. The introduction is particularly noteworthy for its clarity and comprehensive overview of the field. The detailed appendix is also commendable.


# Major Issues

## 1. Role of Feynman-Kac

In the proposed FK-PINNs, the FK approach is used to generate additional data points inside the domain, which are then incorporated into a supervised loss term. However, the stated theorems and theoretical considerations appear to be independent of the source of these data points. Hence, one could presumably use a different method to obtain interior solution estimates (e.g., FEM or other classical numerical methods) while achieving similar results.
However, the idea of including a supervised loss term with additional interior data points was already considered in the original work by Raissi et al. (2019), which significantly diminishes the novelty of the FK-specific training. The authors should clarify what advantages FK offers over simpler alternatives, particularly for low-dimensional problems where mesh-based methods are efficient. Also, an experimental comparison with simpler alternatives would be desirable.

If the core finding is "just" that additional data-fidelity loss terms can structurally enhance PINN training, this would still be a very valuable contribution. However, this would require a different narrative.

## 2. Readability aspects

While most parts of the article are clearly structured and pleasant to read, I strongly recommend clarifying and reworking the structure in Sections 4 and 5. In particular, the authors should avoid using symbols (e.g., $\phi$) before introducing them in a later section. Also, the nomenclature introduced in Section 4.2 seems partially unnecessary, which makes it harder to follow. Readability could be enhanced by relegating auxiliary information to footnotes (e.g., the definition of L-smooth), helping the reader focus on the main arguments. In addition, I would avoid using the same symbol (e.g., $\beta$) multiple times with different meanings.

Due to the format specifications of ICML, it is not always typographically clear when a definition environment ends. I would therefore reconsider the use of definition environments, particularly if the definitions are not referenced later.

This is rather a matter of taste, but in my opinion, readability can be greatly enhanced by placing a short motivation before stating theorems.

## 3. Assumptions and constants in the main theorems

The theoretical findings (mainly theorems 5.4 and 6.2) appear very interesting for practical applications. However, they depend on several assumptions that may be difficult to verify in practice. It would be helpful to discuss when these assumptions are expected to hold and whether they can be checked empirically.

On the other hand, the key statements include several undetermined constants. It would be interesting to see their values in practical examples. More experiments in this direction would therefore be helpful.

Although these additional experiments are not strictly necessary in my opinion, they could significantly strengthen the theoretical statements for practical applications by quantitatively connecting experiments to theoretical bounds.

# Minor Issues

1. **Abstract:** Although the abstract aligns with the suggested length (4–6 sentences), it could be shortened slightly to provide a more concise description.
2. **Wording:** What are "mixed Dirichlet boundary conditions"? Standard terminology includes Dirichlet BCs, Neumann BCs, or mixed BCs (combining Dirichlet and Neumann on different parts of the boundary). This non-standard term is introduced in lines 114–115 and should be clarified. Additionally, it is unclear to the reader what "Hessian-like" operators refer to (line 118, right column). A proper definition would improve understandability.
3. **Summary regarding standard PINNs**: Since their introduction, there have been several developments in standard PINNs, which should -- at least briefly -- be included in the background section. For instance, it is now widely recommended -- and considered standard practice -- not to use uniformly distributed collocation points (see e.g. Wu et al.: A comprehensive study of non-adaptive and residual-based adaptive sampling for physics-informed neural networks (2022)). Furthermore, second-order optimization methods are often preferred in the training of PINNs (see e.g. Urban et al.: Unveiling the optimization process of physics informed neural networks: How accurate and competitive can PINNs be? (2024)). A comparison with a state-of-the-art PINN baseline -- incorporating, e.g., adaptive sampling and second-order optimization -- would provide a fairer evaluation and strengthen the contribution.
4. **Related work:** Several approaches already combine PINNs and FK, and should be mentioned explicitely, e.g., Li, Liu: Feynman-Kac Operator Expectation Estimator (2024). In addition, there are other ML methods for solving PDE based on FK, that should potentially be mentioned, e.g., Zheng et al.: Deep Feynman-Kac Methods for High-dimensional Semilinear Parabolic Equations: Revisit (2025).
5. **Redundancy:** In my opinion, it is not necessary to re-introduce the nomenclature from section 2 in section 6.1.
6. **Related work for error bounds:** Although the nature of the statements differs, the authors should mention and compare their error bounds with the work of Hillebrecht, Unger: Certified machine learning: Rigorous a posteriori error bounds for PDE defined PINNs (2022).

# Editorial Comments

* The notation of $PL^\star$ and $PL_\star$ is used inconsistently.
* Hyphens are used very inconsistently and appear to be artifacts from copying.

---

> ### Author Rebuttal · Authors · 2026-03-31
>
> We thank the reviewer for their thoughtful comments and address each point below.
>
> ## Major Issues
>
> ### 1. Role of Feynman–Kac
>
> We use Feynman–Kac supervision because it provides a generic, mesh‑free way to generate interior data in high‑dimensional or irregular domains where mesh‑based solvers (e.g., FEM) are often infeasible; when a full FEM solution is available, a separate PINN surrogate is usually unnecessary. Our main message is that a small amount of accurate interior supervision can substantially improve the PINN loss landscape and training reliability. Our analysis (including new error bounds for tanh networks) studies how such labels shape PINN optimization, and the experiments are designed to validate these effects rather than to benchmark FK‑PINNs against classical solvers.
>
> ### 2. Readability aspects
>
> We thank the reviewer for the helpful suggestions, which we will implement in the revised version.
>
> ### 3. Assumptions and constants in the main theorems
>
> We thank the reviewer for highlighting the practical relevance of our results and for prompting a clearer summary of our assumptions, which we will make explicit in the revision. Regarding the constants in our bounds, we agree that a more systematic numerical study would be valuable. Here we focus on qualitative effects of FK‑based supervision, such as improved conditioning and non‑asymptotic $L^2(\Omega)$ error bounds, and use experiments mainly to confirm these trends. A detailed investigation of the constants and their scaling is left for future work.
>
> ## Minor Issues
>
> ### Abstract length
>
> We will revise the abstract to make it slightly more concise while retaining its message.
>
> ### Wording
>
> We will remove the term “mixed Dirichlet boundary conditions” and instead describe the boundary decomposition directly, and we will replace “Hessian‑like operator” with a direct reference to the NTK operator in (B.6).
>
> ### Comparison with standard PINNs
>
> We thank the reviewer for these pointers and agree that a plain “vanilla” PINN is not an adequate baseline. Our goal is to compare with reasonably strong PINN-training variants.
>
> **Residual-based sampling.**  We retrained all baseline PINNs with residual‑based adaptive sampling on each PDE in Appendix E. The results (https://ibb.co/fGSj5V5X) show that while this helps, it does not remove the failure modes. We will include this table and discussion in the revision.
>
> **Second-order optimization.**  All “standard PINN” baselines use the same optimization pipelines as FK‑PINNs (Adam + L‑BFGS, Adam + L‑BFGS + NNCG, or ENGD; see Appendix E, Table 4), so the performance gap is not due to weaker optimizers.ance gap does not arise from using weaker optimizers for the baselines.
>
> ### Related work on PINNs and FK
>
> We will cite and discuss these works which use FK in a way that is complementary to ours.
>
> ### Redundancy in section 6.1.
>
> We thank the reviewer for this suggestion and will follow through in our revision.
>
> ### Related work for error bounds
>
> We will cite this reference. Hillebrecht & Unger (2022) give a posteriori bounds based on observed residuals, whereas we provide a priori bounds explicit in the architecture, sample sizes, and optimization, giving sufficient conditions for a target $L^2(\Omega)$ error. The approaches are thus different but complementary.
>
> ## Editorial Comments
>
> Thank you for pointing out the inconsistencies for PL notation and hyphenation throughout the paper. Those will be fixed in the revised version.
>
> ## Key Questions For Authors
>
> ### Comparison with FEM-based supervision
>
> Using FEM solely to generate sparse supervision points is atypical: FEM already yields a global mesh solution, so if that solution is available, a separate PINN surrogate is usually unnecessary. One could in principle use a very coarse FEM grid, but the resulting label error would typically violate the supervision-quality assumptions in Proposition 6.1 and thus invalidate the guarantees of Theorem 6.2. Our focus is instead on high-dimensional or complex-geometry regimes where FEM is unavailable or intractable and Monte Carlo–based methods (including FK) are the practical way to obtain flexible, localized, mesh-free supervision.
>
> ### Use of adaptive weighting for both FK-PINNs and standard PINNs
>
> Yes. Both PINNs and FK-PINNs are trained under *exactly the same conditions*: same architectures, collocation and boundary sampling, training schedule, and importantly, *same adaptive weighting scheme* from Section 4.2.
>
> ### Sensitivity to the choice of $\lambda_{FK}$
>
> We agree that a clearer picture of how $\lambda_{FK}$ affects FK-PINN performance would be valuable, and have thus performed some ablation studies on this hyperparameter, which will be included in the revision. The preliminary results are available here: https://ibb.co/qTj7hcF.
>
> We hope that these answers adequately address the reviewer’s concerns and we thank them again for their constructive and detailed feedback.

---

> > ### Author Rebuttal · Reviewer_h7Jj · 2026-04-02
> >
> > Thank you for the detailed rebuttal. The confirmation that adaptive weighting is applied equally to both FK-PINNs and standard PINNs (Q2) is appreciated and resolves that concern. The residual-based adaptive sampling experiments are also a welcome addition. However, my main concern regarding the role and positioning of the Feynman-Kac component remains unresolved.
> >
> > Your rebuttal argues that FK is motivated by settings where FEM is infeasible. I appreciate that obtaining a full global FEM solution when only a few interior points are needed is indeed atypical. However, generating sparse pointwise supervision via a coarse FEM or FDM grid is a natural and cheap alternative. Notably, a deterministic bias from coarse FEM would not violate the assumptions of Proposition 6.1, which -- in my understanding -- only requires bounded bias and controlled noise. Furthermore, all numerical examples in the paper are low dimensional problems where such approaches are straightforwardly applicable.
> >
> > In addition, the overhead (Table 5 in the article) from FK supervision is of OOM larger than the cost of an FEM/FDM computation, which usually takes less than a second on these problems. This makes it difficult to justify the FK-specific approach on practical efficiency grounds for the problems studied.
> >
> > The rebuttal also implicitly positions FK as a high-dimensional tool, but the paper's own error rate $\beta = (s-2)/(2(d+s-2))$ decays rapidly with increasing $d$, reflecting a curse of dimensionality as well. Further, the cost of FK trajectory simulation could grow rapidly in high-dimensions.
> >
> > I agree, that FK is indeed inherently pointwise, which is a genuine advantage. However, there are several further point-wise methods (e.g., Walk on Spheres, BEM, Green's function) -- some of them even seem to avoid the time-discretization bias from FK. Also, the authors cite Sabelfeld (1991) -- a classical WoS reference -- but never discuss it as a natural alternative. A comparison or at least a discussion of WoS or other methods would significantly strengthen the positioning of the FK approach.
> >
> > To be constructive: the operator-preconditioning analysis and non-asymptotic error bounds are genuinely interesting and stand on their own. If the core message is that any accurate sparse interior supervision improves PINN conditioning -- with FK as one practical instance -- this would be a valuable and honest framing. As it stands, presenting FK as the essential ingredient risks overstating its specific role.
> >
> > I maintain my current score, but would be happy to revise upward if these points are addressed, ideally with a comparison against at least one alternative supervision strategy and a more careful discussion of high-dimensional scaling.

---

> > > ### Author Response · Authors · 2026-04-07
> > >
> > > We thank the referee for their comprehensive response, and we are glad to see that their concerns have been partially resolved. We address the remaining follow-up questions below:
> > >
> > > ## FEM vs FK supervision in low-dimensional settings
> > >
> > > While we initially did not consider using FEM as supervision data for reasons discussed above, we understand and agree with the referee's point that using a coarse FEM solution as supervision data (when available) could, on paper, be a competitive alternative to FK supervision. We have thus implemented this alternative approach on all the PDEs we considered in our initial numerical experiments: we computed an FEM solution on a coarse mesh with roughly as many points as the original FK-PINN used as supervision data, and used that to enhance the training loss (see here: https://ibb.co/XkZn3gW0). For completeness, we also report the number of supervision points used in each method in the below table.
> > >
> > > | Poisson                          | Schrödinger-type                 | MET                              | Committor                        |
> > > |----------------------------------|----------------------------------|----------------------------------|----------------------------------|
> > > | $N_{FK}= 256 $, $N_{FEM} = 256 $ | $N_{FK}= 464 $, $N_{FEM} = 441 $ | $N_{FK}= 208 $, $N_{FEM} = 225 $ | $N_{FK}= 256 $, $N_{FEM} = 256 $ |
> > >
> > > As supported by our theory, the FEM supervision term indeed helps regularize the loss landscape and prevents the PINN from getting trapped in spurious local minima. However, on such coarse meshes, the bias of the FEM solution is generally too strong and thus degrades the solution quality compared to (accurate) FK data. For illustration: here is the true solution of the Poisson PDE we consider (https://ibb.co/TByZx2dP), and the solution computed by coarse FEM (https://ibb.co/jkDk4Hjs).
> > >
> > > ## High-dimensional scaling
> > >
> > > The reason we "implicitly position FK as a high-dimensional tool" is simply because it is meant as a solution to overcome the well-known "failure modes" of PINNs. Because PINNs are meant as an alternative approach to numerically solve PDEs whenever classical methods are intractable (typically, in high dimensions and/or complex geometries), we believe that presenting our theoretical results for arbitrary dimensions is reasonable.
> > >
> > > Regarding the scaling of our rates with dimension, there are two things we would like to point out: first, we would like to highlight that the rates we obtain are of similar order to the minimax optimal rates obtained, e.g., in [1], thus this type of scaling with $d$ is typically unavoidable and is sometimes referred to as the *Curse of Dimensionality (CoD)*. Second, the appeal of our error analysis is not just the (close to optimal) rate, but rather the fact that unlike similar works, like [1, 2] which *assume* having access to a minimizer of the loss, our bound holds for a model trained with *finitely many steps of gradient descent*. In particular, **we do not claim that FK-PINNs can beat the CoD**, but rather that **FK supervision translates rates for the (intractable) loss minimizer into rates for a (realistic) model trained after finitely many steps of gradient descent**. We will clarify this point in the revised manuscript.
> > >
> > > To illustrate this scaling, we have trained standard PINNs and FK-PINNs on 10-dimensional Black-Scholes PDE (https://ibb.co/tp3SZxH4). Our (preliminary) results show that while performance degrades in higher dimension, the beneficial effect of FK supervision is still clear.
> > >
> > > ## Comparison with other methods
> > >
> > > As mentioned earlier, and as we will clarify in our manuscript, the regularizing of the loss landscape is agnostic to the source of supervision data, as long as the data is "accurate enough". We think that FK is a general and flexible way to acquire such data, especially for problems that PINNs were originally made for. Indeed, in comparison (as per our understanding):
> > >
> > > - WoS is an alternative method to compute the FK functional with better theoretical guarantees, and can be seen as a "special case" of our work (WoS is not available for all FK PDEs).
> > > - BEM requires (a) access to the fundamental solution (b) inverting a linear system in $d-1$ dimensions (still scales poorly for $d$ large) (c) sufficient boundary regularity.
> > > - Green's function methods (a) require an accurate Green's function approximation and (b) do not extend nicely to higher dimensional problems.
> > >
> > > Thus none of these alternatives appear realistic to us.
> > >
> > > We hope that this resolves the referee's concerns.
> > >
> > > [1] Lu, Y., Chen, H., Lu, J., Ying, L., & Blanchet, J. (2022). *Machine Learning For Elliptic PDEs: Fast Rate Generalization Bound, Neural Scaling Law and Minimax Optimality.* International Conference on Learning Representations.
> > >
> > > [2] Lei, G., Lei, Z., Shi, L., Zeng, C., & Zhou, D. X. (2025). *Solving PDEs on spheres with physics-informed convolutional neural networks.* Applied and Computational Harmonic Analysis, 74, 101714.

---

### Official Review · Reviewer_CKni · 2026-03-13

**Soundness:** 3
**Presentation:** 3
**Significance:** 2
**Originality:** 2
**Overall Recommendation:** 4
**Confidence:** 4

**Summary:**

This paper proposes a training framework for physics-informed neural networks that incorporates a Feynman–Kac based supervision signal into the PINN objective. The main idea is to use the stochastic representation of PDE solutions provided by the Feynman–Kac formula as an auxiliary training signal that acts as a form of preconditioning or regularization for the PINN objective. By approximating the corresponding stochastic operator with neural networks and Monte Carlo simulations, the method aims to guide the PINN optimization toward solutions that are consistent with the probabilistic representation of the PDE.

**Compliance With Llm Reviewing Policy:**

Affirmed.

**Final Justification:**

This paper proposes an interesting integration of the Feynman–Kac operator into the PINN framework with supporting theoretical analysis . The idea is conceptually meaningful and technically sound.

However, there are two main limitations. First, beyond direct performance comparisons, the paper does not clearly articulate compelling methodological advantages over existing Feynman–Kac-based approaches. Second, the theoretical analysis, while rigorous, provides limited practical insight into when the method significantly improves PDE solving.

Overall, despite these limitations, the paper does not contain clear factual errors or critical flaws that would warrant rejection. Given its conceptual merit and potential, I therefore lean toward a weak accept.

**Key Questions For Authors:**

1. How does the proposed method differ from approaches such as Deep Picard Iteration or DFLM? A clearer comparison would help clarify the novelty of the framework.

2. For the PDE classes considered in the paper (e.g., linear parabolic PDEs), classical numerical solvers are already highly developed. What concrete advantages does the proposed method offer compared to these established approaches?

3. How sensitive is the method to the variance introduced by Monte Carlo sampling?

**Limitations:**

Yes

**Strengths And Weaknesses:**

**Strengths**

1. Interesting integration of probabilistic representations with PINNs: The idea of incorporating the Feynman–Kac representation as a supervision signal in PINN training is conceptually interesting. Using probabilistic solution representations to guide neural PDE solvers is a meaningful direction, particularly for high-dimensional PDEs where classical grid-based solvers become infeasible.

2. Methodological effort in approximating the Feynman–Kac operator: The paper introduces a concrete procedure for approximating the Feynman–Kac operator using neural networks and stochastic simulation.

3. Overall, the paper is clearly written and logically structured.

4. The paper provides a reasonably rigorous theoretical analysis of the proposed framework. In particular, the analysis attempts to characterize the behavior of the modified objective and its relationship to the underlying PDE solution.

**Weaknesses**

1. The claimed “broad class” of PDEs is somewhat misleading: Throughout the paper, the method is described as applicable to a broad class of PDEs. However, PDEs that admit a Feynman–Kac representation are actually quite restricted.

2. Missing literature review on Feynman–Kac–based neural PDE solvers: The literature review on methods that leverage Feynman–Kac representations for neural PDE solvers is incomplete. For example, see

-  Han, Jiequn et al. Deep Picard Iteration for High-Dimensional Nonlinear PDEs, SIAM Journal on Scientific Computing (2026).

- Han, Jihun, Mihai Nica, and Adam Stinchcombe. A Derivative-Free Method for Solving Elliptic Partial Differential Equations with Deep Neural Networks, Journal of Computational Physics (2020).

These works explicitly exploit stochastic representations of PDEs and are closely related to the ideas explored in this paper. A clearer discussion of how the proposed method differs from these approaches would significantly improve the positioning of the contribution.

3. Limited novelty of the Feynman–Kac preconditioned PINN objective: The idea of using Feynman–Kac representations to supervise or guide neural PDE solvers has already appeared in several forms in the literature. Therefore, the novelty of introducing a Feynman–Kac–based PINN objective appears somewhat limited. But I appreciate that he paper does introduce certain methodological differences in how the Feynman–Kac operator is approximated and incorporated into the loss function, the conceptual contribution relative to the existing approaches.

4. Insufficient experimental validation: Since the method targets PDE classes that are already well studied, particularly linear parabolic PDEs, the empirical evaluation should provide strong evidence that the proposed approach offers clear advantages over existing numerical methods. Also, it is necessary to compare with the existing approaches based on Feyman-Kac representation based neural solvers.

5. The paper places some emphasis on theoretical analysis of the proposed objective. However, because the method only applies to PDEs that admit a Feynman–Kac representation, the broader impact of such analysis is somewhat limited

---

> ### Author Rebuttal · Authors · 2026-03-31
>
> We thank the reviewer for the careful reading of our manuscript and their thoughtful comments. We address the points below.
>
> ## Weaknesses
>
> ### 1. Claimed “broad class” of PDEs
>
> While our main results focus on linear parabolic and elliptic equations (classical FK setting), Appendix A (in particular A.7) shows that many PDEs (e.g., semilinear parabolic, HJB...) admit FK-type stochastic representations. Our Monte Carlo approximation of the FK term and the resulting noisy interior supervision can, in principle, be used for any PDE with such a representation. Moreover, Theorem 5.4 is agnostic to how labeled interior data are obtained, so our theory applies to any linear PDE with sufficiently accurate interior labels. We will clarify this broader scope in the revision.
>
> ### 2. Missing literature review on FK-based neural PDE solvers
>
> We thank the reviewer for pointing out these two relevant papers which we will cite and discuss (see also our response to Key Question 1).
>
> ### 3. Limited novelty of the FK-preconditioned PINN objective
>
> While adding supervised data to a PINN objective has been considered before, prior work, as far as we know, (i) does not use stochastic‑representation‑based supervision as we do (labels are usually assumed to come from surrogate models, FEM, or measurements) and (ii) treats such supervision almost purely empirically, without a rigorous explanation of when and why it helps. We instead use FK-based interior labels and provide what we believe is the first rigorous analysis of their effect on the PINN loss landscape, showing that under our assumptions the FK term improves conditioning and removes certain failure modes. We will make our contributions more explicit in the revision.
>
> ### 4. Insufficient experimental validation
>
> Our main goal is not to propose a PDE solver that outperforms classical numerical methods, but to analyze and mitigate failure modes of PINNs by augmenting them with labeled (FK) data. Accordingly, our experimental comparison is between standard PINNs and FK-PINNs, to show that the additional supervision regularizes the loss landscape and helps overcome these failure modes. DPI and DFLM instead use stochastic representations to define stand-alone solver objectives for nonlinear parabolic equations. The approaches therefore target different goals and are not directly comparable numerically. It would be interesting future work to combine such alternative objectives with PINN loss to further help training of PINNs solving nonlinear parabolic equations. Empirical results in this direction appear in [1, Supplementary Material SM2.5].
>
> ### 5. Limited broader impact of the theory
>
> As discussed above, many PDEs (including important nonlinear examples) admit stochastic representations that can provide supervision data. For the linear PDEs covered by our analysis, Theorems 5.4 and 6.2 show that such supervision provably improves the loss landscape and yields explicit a priori error bounds. This regularizing effect is not specific to Feynman–Kac: any sufficiently accurate interior supervision data has the same qualitative impact, and FK-based supervision is one natural and scalable way to generate such labels.
>
> ## Key Questions For Authors
>
> ### 1. Difference from Deep Picard Iteration and DFLM
>
> DPI and DFLM use FK-type representations to define stand-alone stochastic-solver objectives for nonlinear parabolic PDEs. In contrast, our approach remains within the PINN framework: we keep the residual-based objective enforcing the PDE and boundary conditions and augment it with an FK-based supervision term providing noisy interior labels. Thus FK is used to regularize and precondition standard PINN training rather than to replace the residual formulation, and we will clarify this distinction and position FK-PINNs as an enhancement of PINNs rather than a direct competitor to Deep Picard or DFLM.
>
> ### 2. Advantages over classical numerical solvers
>
> The purpose of our work is not to beat classical solvers, but to improve the reliability of PINNs in the regimes they target: high-dimensional, mesh-free, or irregular domains where classical methods fail.
>
> ### 3. Sensitivity to Monte Carlo variance
>
> The Monte Carlo estimator of the Feynman–Kac functional in Algorithm 1 has variance that scales like $1/N_{MC}$. In experiments with $N_{MC}$ between 100 and 2000, larger $N_{MC}$ improves the average accuracy of FK-PINNs by reducing the variance of the supervision signal, while for fixed $N_{MC}$ repeated runs yield similar errors with small standard deviations. Figure 13(b) in Appendix E.5 summarizes these results and indicates that FK-PINNs are not overly sensitive in practice to Monte Carlo randomness.
>
> We hope these points satisfactorily address the referee's concerns.
>
> [1] Bensoussan, A., Han, J., Yam, S. C. P., & Zhou, X. (2023). *Value-gradient based formulation of optimal control problem and machine learning algorithm.* SIAM Journal on Numerical Analysis, 61(2), 973-994.

---

> > ### Author Rebuttal · Reviewer_CKni · 2026-04-02
> >
> > Thank you for the detailed rebuttal.I appreciate the effort to broaden the discussion through Appendix A, where the authors list several PDE classes admitting FK-type stochastic representations. However, this does not fully resolve my main concern about scope and significance.
> >
> > The authors acknowledge that the goal is not empirical superiority over established solvers, and experiments are limited to comparisons with vanilla PINNs. As such, the empirical evidence does not convincingly demonstrate strong practical impact. In particular, stronger validation on genuinely challenging high-dimensional problems (e.g., HJB equations) would be necessary. At present, the claimed broader applicability appears more as a potential direction than a demonstrated strength, and the effective scope remains limited.
> >
> > While the paper makes a serious theoretical effort, the contribution appears limited. The analysis relies on a PL-type condition, which is relatively strong and restricts the generality of the conclusions. Moreover, although non-asymptotic error bounds are presented, the lack of explicit constants makes it difficult to interpret their practical implications. As a result, the overall impact of the theory remains somewhat limited.
> >
> > Nevertheless, the paper explores a relatively less-studied direction of leveraging Feynman–Kac representations within the PINN framework, which still provides sufficient novelty to merit acceptance. Therefore, I will maintain my original weak accept score.

---

> > > ### Author Response · Authors · 2026-04-07
> > >
> > > We thank the reviewer for their acknowledgement and for the appreciation of the novelty of our contributions. Below we answer the main points raised in the referee's response.
> > >
> > > ## Limited applicability and scope
> > >
> > > > The authors acknowledge that the goal is not empirical superiority over established solvers, and experiments are limited to comparisons with vanilla PINNs. As such, the empirical evidence does not convincingly demonstrate strong practical impact. In particular, stronger validation on genuinely challenging high-dimensional problems (e.g., HJB equations) would be necessary.
> > >
> > > We take this opportunity to reiterate our paper's main contribution: a **new method to alleviate the failure modes of PINNs**, backed by rigorous theoretical analysis . We are therefore *not* trying to introduce a *new method to numerically solve PDEs*, and the basis of comparison of our method is with *other methods designed to alleviate the failure modes of PINNs*. Regarding the points raised in this quote, we reply below.
> > >
> > > - *"[...] experiments are limited to comparisons with vanilla PINNs":* We respectfully point out that this claim is **not true**. Our FK-PINNs are compared with other solvers designed to alleviate the known issue with the loss landscape of PINNs. In particular, we compare our FK-PINNs with PINNs using (a) residual-based adaptive sampling, (b) Adam + L-BFGS, (c) Adam + L-BFGS + NN-CG and (d) ENGD (we refer the referee to Table 4 in our paper and the "comparison with standard PINNs" section of our reply to Reviewer h7Jj). All of these alternative methods were introduced to overcome the training difficulties of PINNs, and can therefore not be referred to as "vanilla PINNs".
> > >
> > > - *"[...] stronger validation on genuinely challenging high-dimensional problems (e.g., HJB equations) would be necessary":* As per our above discussion, the goal of our numerical experiments is to validate our theory and confirm that FK supervision can successfully overcome the PINNs failure modes, especially where other approaches designed for the same purpose fail. For our current purposes, we therefore believe that our numerical experiments are enough.
> > >
> > > ## Limited applicability of the theory
> > >
> > > >  The analysis relies on a PL-type condition, which is relatively strong and restricts the generality of the conclusions. Moreover, although non-asymptotic error bounds are presented, the lack of explicit constants makes it difficult to interpret their practical implications.
> > >
> > > Regarding the PL* condition we use, there is a growing body of theoretical evidence in the literature supporting the satisfaction of this condition for (sufficiently overparametrized) neural networks for general losses. A non exhaustive list of such works includes:
> > > - Liu, Chaoyue, Libin Zhu, and Mikhail Belkin. *"Loss landscapes and optimization in over-parameterized non-linear systems and neural networks."* ACHA (2022)
> > > - Taheri, Hossein, and Christos Thrampoulidis. *"Generalization and stability of interpolating neural networks with minimal width."* JMLR (2024)
> > > - Chatterjee, Sourav. *"Convergence of gradient descent for deep neural networks."* arXiv preprint arXiv:2203.16462 (2022).
> > > - Du, Simon S., et al. *"Gradient Descent Provably Optimizes Over-parameterized Neural Networks."* ICLR (2019)
> > >
> > > In light of these works (and references within), it seems to us that using this PL* assumption for our analysis, while constraining, is not overly restrictive.
> > >
> > > Regarding the lack of closed-form expressions for the multiplicative constants in our non-asymptotic error bounds, we would first like to mention that although these constants can be straightforwardly estimated by inspecting the proof, we chose not to in order to (a) save space and (b) avoid technicalities that would distract from the main point: the scaling laws. Indeed, learning-theoretic excess risk bounds of this type are routinely presented without the explicit multiplicative constants in the literature, as the focus is always on the dependence on the sample size $N$. See, for instance
> > > - Chen, Mo, et al. *"Convergence analysis of PINNs with over-parameterization."* Communications in Computational Physics (2025)
> > > - Ding, Zhao, et al. *"Convergence analysis of deep Ritz method with over-parameterization."* Neural Networks (2025).
> > > - Lu, Yiping et al. *"Machine Learning For Elliptic PDEs: Fast Rate Generalization Bound, Neural Scaling Law and Minimax Optimality."* ICLR (2022)
> > > - Lei, Guanhang. *"Solving PDEs on spheres with physics-informed convolutional neural networks."* ACHA (2025)
> > >
> > > Not including the constants in closed-form thus does not, we believe, make our bound any harder to interpret than any other learning-theoretic error bound in the existing literature on physics-informed machine learning.
> > >
> > > We hope that these clarifications satisfactorily address the referee's remaining concerns.

---

### Decision · Program_Chairs · 2026-04-30

**Decision:**

Accept (regular)

**Comment:**

This paper introduces a new alternative, Feynman-Kac supervision, to improve the conditioning of the PINN's loss, with theoretical guarantee.

All the reviewers agreed with the soundness of this new perspective for better conditioning the loss of PINNs, but raised two main concerns: 1) the utility of the proposed pre-conditioning approach by comparing more recent SOTA methods; 2) the clarity of the novelty claim. During the rebuttal period, the authors provided extensive comparison results as requested by the reviewers and clarified the main advantage of the Feynman-Kac supervision. From my point of view, I think most of the concerns have been addressed relatively well.

After pondering the greater value of this paper's methodological novelty than the detailed empirical incompleteness, I still recommend acceptance, but suggest the authors carefully add more empirical results and clarifications of the real punching point of the Feynman-Kac supervision in the final version.